# Federated Spectral Clustering via Secure Similarity Reconstruction

**Dong Qiao**[1,2]      **Chris Ding**[1]      **Jicong Fan** [1,2*]
[1]The Chinese University of Hong Kong, Shenzhen, China
[2]Shenzhen Research Institute of Big Data, Shenzhen, China
dongqiao@link.cuhk.edu.cn     {chrisding,fanjicong}@cuhk.edu.cn

## Abstract

Federated learning has a significant advantage in protecting data and information privacy. Many scholars proposed various secure learning methods within the framework of federated learning but the study on secure federated unsupervised learning especially clustering is limited. We in this work propose a secure kernelized factorization method for federated spectral clustering on distributed data. The method is non-trivial because the kernel or similarity matrix for spectral clustering is computed by data pairs, which violates the principle of privacy protection. Our method implicitly constructs an approximation for the kernel matrix on distributed data such that we can perform spectral clustering under the constraint of privacy protection. We provide a convergence guarantee of the optimization algorithm, reconstruction error bounds of the Gaussian kernel matrix, and the sufficient condition of correct clustering of our method. We also present guarantees of differential privacy. Numerical results on synthetic and real datasets demonstrate that the proposed method is efficient and accurate in comparison to baselines.

## 1   Introduction

In the era of big data, human beings can analyze massive data in various fields due to the improvement of storage and computational capabilities of computing devices [Li *et al.*, 2020b]. Some popular fields such as artificial intelligence, machine learning, internet of things (IoT), and cloud computing have seen explosive development over the past few years. Nevertheless, a side effect of this trend is that individuals and organizations have more and more concerns about potential violation of privacy [Kairouz *et al.*, 2021]. As a result, it has become a challenge to mine valuable information from user data but not to directly access it.

Federated learning [Kairouz *et al.*, 2021; McMahan *et al.*, 2017] can train a global model without retrieving dispersed data [Yang *et al.*, 2018]. This advantage has made it so popular that many scholars have put much effort into the study of federated learning. For example, Yang *et al.* [2019] presented the definitions of horizontal federated learning, vertical federated learning, and federated transfer learning. Some privacy-preserving machine learning models were also presented. For instance, He *et al.* [2020] developed a federated group knowledge transfer algorithm to train small CNNs on edge devices. Chen *et al.* [2018] proposed a protocol to conduct privacy-preserving ridge regression over high-dimensional data. Besides, Kim *et al.* [2018] proposed a block-chained federated learning architecture that enables on-device learning without any central coordination.

Regardless of the great progress of federated learning, it can be found that most of the existing studies are for supervised learning [Li *et al.*, 2020a; Ghosh *et al.*, 2020]. Note that collecting labeled data may deserve very high cost in real situations [Li *et al.*, 2020b] while unlabeled data are abundant. Thus, it

---

*Corresponding author

37th Conference on Neural Information Processing Systems (NeurIPS 2023).

is necessary and important to study federated learning for unsupervised learning [Zhang *et al.*, 2020; Tzinis *et al.*, 2021; Zhuang *et al.*, 2021; Dennis *et al.*, 2021] such as clustering [Ng *et al.*, 2001; Fan and Chow, 2017; Fan *et al.*, 2018, 2021; Fan, 2021; Cai *et al.*, 2022; Fan *et al.*, 2022]. For example, Li *et al.* [2021] proposed a federated matrix factorization with a privacy guarantee for recommendation systems. Wang and Chang [2022] proposed two federated matrix factorization algorithms that can be used for federated clustering. Besides, there are some studies on federated spectral clustering. For instance, Wang *et al.* [2020] presented a federated multi-view spectral clustering method under the assumption that the data of each view are in one client. Hernández-Pereira *et al.* [2021] developed a cooperative spectral clustering model to deal with distributed data but the model is linear. However, the study on federated spectral clustering is still very limited and deserves more attention and effort.

In this paper, we propose a federated kernelized factorization method to reconstruct a similarity matrix for secure spectral clustering on distributed data. Our contributions are as follows.

- We propose a federated spectral clustering algorithm and provide convergence guarantee for the optimization.
- We further propose to add noise to the data or the learned factors to enhance the security of clustering and provide guarantees of differential privacy.
- We provide upper bounds for the reconstruction error of the true similarity matrix and theoretical guarantees for correct clustering.

We test our method on both synthetic data and real datasets in comparison to baselines, which verify the effectiveness of our method.

**Notations**    We use $y$, $\boldsymbol{y}$, and $\boldsymbol{Y}$ to denote scalar, vector, and matrix, respectively. The element of $\boldsymbol{Y}$ at row $i$ and column $j$ is denoted by $y_{ij}$. We use $\|\cdot\|_2$ to denote the $\ell_2$ norm of a vector and use $\mathrm{Tr}(\cdot)$, $\|\cdot\|_F$, and $\|\cdot\|_{sp}$ to denote the trace, Frobenius norm, and spectral norm of a matrix respectively. The $\ell_\infty$ norm and $\ell_{2,\infty}$ norm of a matrix $\boldsymbol{Y}$ are defined as $\|\boldsymbol{Y}\|_\infty = \max_{ij} |y_{ij}|$ and $\|\boldsymbol{Y}\|_{2,\infty} = \max_j \sqrt{\sum_i y_{ij}^2}$ respectively. $\boldsymbol{K}$, $\mathcal{K}$, $\mathcal{K}$, and $k$ denote the kernel matrix, kernel function, number of clusters, and the number $k$ in KNN, respectively. $\phi$ denotes the feature map induced by $\mathcal{K}$.

## 2    Federated Spectral Clustering (FedSC)

Suppose we have $n$ data points of dimension $m$ distributing in $P$ clients. For convenience, we denote by $\boldsymbol{X} \in \mathbb{R}^{m \times n}$ the matrix composed of all the $n$ data points and denote by $\boldsymbol{X}_p \in \mathbb{R}^{m \times N_p}$ the matrix composed of the $N_p$ data points in client $c_p$, where $N_p \geq 1$, $p = 1, \ldots, P$, and $\sum_{p=1}^P N_p = n$. Without loss of generality, we let $\boldsymbol{X} = [\boldsymbol{X}_1, \boldsymbol{X}_2, \ldots, \boldsymbol{X}_P]$, which means $\{\boldsymbol{X}_p\}_{p=1}^P$ are submatrices of $\boldsymbol{X}$. Our goal is to perform spectral clustering on these data to partition them into $\mathcal{K}$ groups, under the constraint that the data in each client cannot leave the client itself and the privacy of the data should be protected as much as possible, though there could be a central server conducting clustering.

The aforementioned task is non-trivial because in spectral clustering, the first step is constructing an adjacency matrix $\mathbf{A} \in \mathbb{R}^{n \times n}$, which has to evaluate the similarity between every data pair $(\boldsymbol{x}_i, \boldsymbol{x}_j)$ using a metric function $\mathcal{M}(\cdot, \cdot)$ and hence violates the privacy constraint in the task. To solve the problem, we present a federated spectral clustering model in this section.

### 2.1    Similarity Reconstruction via Feature Space Factorization

In spectral clustering, for $\mathcal{M}(\cdot, \cdot)$, there are many choices such as $k$ nearest neighbor similarity and various kernel functions. Let $\mathcal{K}(\cdot, \cdot)$ be a kernel function and we have

$$\mathcal{K}(\boldsymbol{x}_i, \boldsymbol{x}_j) = \phi(\boldsymbol{x}_i)^T \phi(\boldsymbol{x}_j), \tag{1}$$

where $\phi : \mathbb{R}^m \to \mathbb{R}^{m'}$ is a feature map induced by the kernel function[2] and does not need to be carried out explicitly. When it comes to federated spectral clustering, the central server has no access

---

[2]The most widely-used kernel is the Gaussian kernel $\mathcal{K}(\boldsymbol{x}_i, \boldsymbol{x}_j) = \exp\left(-\frac{1}{2r^2} \|\boldsymbol{x}_i - \boldsymbol{x}_j\|_2^2\right)$, of which the feature map $\phi$ is an infinite-order polynomial feature map and $r$ is a hyperparameter controlling the smoothness.

to the raw data distributed in clients and hence cannot compute $\mathcal{K}(\boldsymbol{x}_i, \boldsymbol{x}_j)$ using (1). However, if the central server can learn an effective approximation (denoted by $\widehat{\phi(\boldsymbol{x}_i)}$) for each $\phi(\boldsymbol{x}_i)$ without accessing $\boldsymbol{x}_i$, $\mathcal{K}(\boldsymbol{x}_i, \boldsymbol{x}_j)$ can be estimated, i.e.,

$$\mathcal{K}(\boldsymbol{x}_i, \boldsymbol{x}_j) \simeq \widehat{\phi(\boldsymbol{x}_i)}^T \widehat{\phi(\boldsymbol{x}_j)}. \tag{2}$$

Thus, inspired by [Fan and Udell, 2019; Fan *et al.*, 2021], we propose to approximate each $\phi(\boldsymbol{x}_i)$ by

$$\widehat{\phi(\boldsymbol{x}_i)} = \phi(\boldsymbol{Z})\boldsymbol{c}_i, \tag{3}$$

where $\boldsymbol{Z} = [\boldsymbol{z}_1, \boldsymbol{z}_2, \ldots, \boldsymbol{z}_d] \in \mathbb{R}^{m \times d}$, $\phi(\boldsymbol{Z}) = [\phi(\boldsymbol{z}_1), \phi(\boldsymbol{z}_2) \ldots, \phi(\boldsymbol{z}_d)]$, and $\boldsymbol{c}_i \in \mathbb{R}^d$. Both $\boldsymbol{Z}$ and $\boldsymbol{c}_i$ are learned from individual columns of $\boldsymbol{X}$ and they can be regarded as intermediate variables avoiding the direct access of central server to $\boldsymbol{x}_i$ (details of the learning will be introduced later). It follows from (2) and (3) that

$$\mathcal{K}(\boldsymbol{x}_i, \boldsymbol{x}_j) \simeq \boldsymbol{c}_i^T \phi(\boldsymbol{Z})^\top \phi(\boldsymbol{Z}) \boldsymbol{c}_j. \tag{4}$$

Thus we can reconstruct the similarity between $\boldsymbol{x}_i$ and $\boldsymbol{x}_j$ via (4). For convenience, let $\boldsymbol{K}_{xx} = \mathcal{K}(\boldsymbol{X}, \boldsymbol{X}) = \phi(\boldsymbol{X})^\top \phi(\boldsymbol{X})$, $\boldsymbol{K}_{zz} = \mathcal{K}(\boldsymbol{Z}, \boldsymbol{Z}) = \phi(\boldsymbol{Z})^\top \phi(\boldsymbol{Z}) \in \mathbb{R}^{d \times d}$, and $\boldsymbol{C} = [\boldsymbol{c}_1, \boldsymbol{c}_2, \ldots, \boldsymbol{c}_n] \in \mathbb{R}^{d \times n}$. Then we have

$$\boldsymbol{K}_{xx} \simeq (\phi(\boldsymbol{Z})\boldsymbol{C})^T (\phi(\boldsymbol{Z})\boldsymbol{C}) = \boldsymbol{C}^T \boldsymbol{K}_{zz} \boldsymbol{C} \triangleq \hat{\boldsymbol{K}}_{xx}. \tag{5}$$

Now we use $\hat{\boldsymbol{K}}_{xx}$ as a reconstructed similarity matrix for spectral clustering.

In the form of federated learning, we expand (3) to

$$\phi(\boldsymbol{X}_p) \simeq \phi(\boldsymbol{Z})\boldsymbol{C}_p, \quad p = 1, \ldots, P. \tag{6}$$

It indicates that $\boldsymbol{Z}$ is shared for all $P$ clients and $\boldsymbol{C}_p$ is private for client $c_p$. Note that (6) is a matrix factorization problem in the feature space induced by a kernel on the data in client $c_p$, $p = 1, \ldots, P$. Letting $\boldsymbol{C} = [\boldsymbol{C}_1, \ldots, \boldsymbol{C}_P]$, we solve the following distributed optimization problem[3]

$$\underset{\boldsymbol{Z}, \boldsymbol{C}}{\text{minimize}} \ F(\boldsymbol{Z}, \boldsymbol{C}) \triangleq \sum_{p=1}^{P} \omega_p f_p(\boldsymbol{Z}, \boldsymbol{C}_p). \tag{7}$$

In (7), $f_p$ is a local objective function for client $c_p$ and $\omega_1, \ldots, \omega_P$ are nonnegative weights for the clients. In this work, we let

$$\begin{aligned} f_p(\boldsymbol{Z}, \boldsymbol{C}_p) =& \frac{1}{2} \|\phi(\boldsymbol{X}_p) - \phi(\boldsymbol{Z})\boldsymbol{C}_p\|_F^2 + \frac{\lambda}{2} \|\boldsymbol{C}_p\|_F^2 \\ =& \frac{1}{2} \text{Tr}(\mathcal{K}(\boldsymbol{X}_p, \boldsymbol{X}_p)) - \text{Tr}(\boldsymbol{C}_p^T \mathcal{K}(\boldsymbol{Z}, \boldsymbol{X}_p)) + \frac{1}{2} \text{Tr}(\boldsymbol{C}_p^T \mathcal{K}(\boldsymbol{Z}, \boldsymbol{Z}) \boldsymbol{C}_p) + \frac{\lambda}{2} \|\boldsymbol{C}_p\|_F^2, \end{aligned} \tag{8}$$

where $\lambda \geq 0$ is a penalty parameter. To guarantee the privacy of information, problem (7) shall be solved in the framework of federated learning.

## 2.2 FedSC by Similarity Reconstruction and Model Averaging

In this section, we develop a FedSC algorithm by similarity reconstruction and model averaging. As a classic and popular framework, FederatedAveraging (or FedAvg) is first introduced in [McMahan *et al.*, 2017] for federated learning. In our work, the proposed FedSC is, therefore, built up based on the backbone of FedAvg as in Figure 1. FedSC consists of two stages. The first stage, shown by the left plot of Figure 1, is federated similarity reconstruction, which constructs a similarity matrix in the manner of federated learning. The second stage, shown by the left plot of Figure 1, is using the reconstructed similarity matrix to implement spectral clustering.

### Stage I Federated Similarity Reconstruction

Step ①: As the startup settings for our algorithm, the shared variable $\boldsymbol{Z}$ (*i.e.*, the dictionary matrix $\boldsymbol{Z}$) and each local coefficient matrix $\boldsymbol{C}_p$ for $p = 1, 2, \cdots, P$ are initialized randomly in the central server and each client, respectively.

---

[3]Note that we do not show the data $\boldsymbol{X}_p$ in the objective explicitly since it is absorbed into $f_p$.

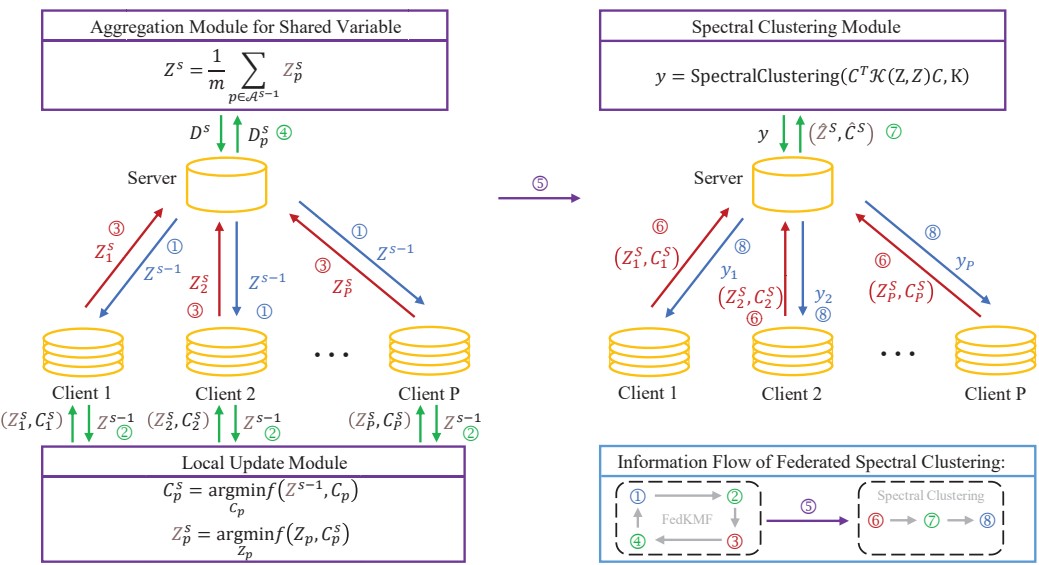

Figure 1: Diagram of the proposed FedSC. Stage I (left plot): Federated Similarity Reconstruction (Steps 1-5). Stage II (right plot): Spectral Clustering (Steps 6-8).

Step ②: For each round $s$, where $1 \leq s \leq S$, the previous shared variable $\boldsymbol{Z}$ will firstly be broadcast to each participated client. After that, every client uses this received dictionary matrix $\boldsymbol{Z}$ to run its own iterative updates of local variables in the Local Update Module (LUM) as:

$$\boldsymbol{C}_p^s = \arg\min_{\boldsymbol{C}_p} f_p(\boldsymbol{Z}^{s-1}, \boldsymbol{C}_p) = \arg\min_{\boldsymbol{C}_p} \frac{1}{2} \left\| \phi(\boldsymbol{X}_p) - \phi(\boldsymbol{Z}^{s-1})\boldsymbol{C}_p \right\|_F^2 + \frac{\lambda}{2} \left\| \boldsymbol{C}_p \right\|_F^2 \qquad (9)$$

$$\boldsymbol{Z}_p^s = \arg\min_{\boldsymbol{Z}_p} f_p(\boldsymbol{Z}_p, \boldsymbol{C}_p^s) = \arg\min_{\boldsymbol{Z}_p} \frac{1}{2} \left\| \phi(\boldsymbol{X}_p) - \phi(\boldsymbol{Z}_p)\boldsymbol{C}_p^s \right\|_F^2 + \frac{\lambda}{2} \left\| \boldsymbol{C}_p^s \right\|_F^2 \qquad (10)$$

Step ③: Each client sends back its own dictionary matrix $\boldsymbol{Z}_p^s$, $p = 1, 2, \ldots, P$, to the central server.

Step ④: The central server collects all (or a subset $\mathcal{A}^{s-1}$ of) these uploaded matrices $\{\boldsymbol{Z}_p^s\}_{p=1}^P$ and averages them into one new matrix $\boldsymbol{Z}^s$ in Aggregation Module (AM), i.e.,

$$\boldsymbol{Z}^s = \frac{1}{|\mathcal{A}^{s-1}|} \sum_{p \in \mathcal{A}^{s-1}} \boldsymbol{Z}_p^s \qquad (11)$$

where $|\mathcal{A}^{s-1}|$ is the number of participated clients. In our study, we fix the number of participating clients for each round $s$. Therefore, we use the notation $\bar{P}$ instead of $|\mathcal{A}^{s-1}|$ in the sequel. This aggregated dictionary matrix $\boldsymbol{Z}^s$ will then be used to push the next round of federated iteration until the tolerance condition is broken.

Step ⑤: When Stage I comes to an end, the spectral clustering will start.

**Stage II   Spectral Clustering**

Step ⑥: Each client sends $(\boldsymbol{Z}_p^S, \boldsymbol{C}_p^S)$ back to the central server for the final aggregation of information.

Step ⑦: The required similarity matrix is then constructed based on the obtained dictionary matrix $\boldsymbol{Z}^S$ and coefficient matrix $\boldsymbol{C}^S$ in Spectral Clustering Module (SCM). Based on this approximated similarity matrix, the standard spectral clustering is implemented as usual:

$$\mathbf{y} = \text{SpectralClustering}(\boldsymbol{C}^T \mathcal{K}(\boldsymbol{Z}, \boldsymbol{Z})\boldsymbol{C}, \mathcal{K}). \qquad (12)$$

Step ⑧: The central server broadcasts its clustering results to every corresponding client.

## 2.3 Optimization Algorithm for Federated Similarity Reconstruction

As described in the above section, alternate updating of local variables is a key to solving the proposed FedSC problem. In the following two parts, we discuss the optimization for $\boldsymbol{Z}$ and $\boldsymbol{C}$, respectively.

For a client $c_p$, consider the corresponding local optimization problem

$$\underset{\boldsymbol{Z},\boldsymbol{C}}{\text{minimize}} \, f_p(\boldsymbol{Z},\boldsymbol{C}) \tag{13}$$

where $f_p(\boldsymbol{Z},\boldsymbol{C}) = \frac{1}{2} \|\phi(\boldsymbol{X}_p) - \phi(\boldsymbol{Z})\boldsymbol{C}\|_F^2 + \frac{\lambda}{2} \|\boldsymbol{C}\|_F^2 = \frac{1}{2}\text{Tr}(\mathcal{K}(\boldsymbol{X}_p,\boldsymbol{X}_p)) - \text{Tr}(\boldsymbol{C}^T\mathcal{K}(\boldsymbol{Z},\boldsymbol{X}_p)) + \frac{1}{2}\text{Tr}(\boldsymbol{C}^T\mathcal{K}(\boldsymbol{Z},\boldsymbol{Z})\boldsymbol{C}) + \frac{\lambda}{2} \|\boldsymbol{C}\|_F^2$. Let the derivative of $f_p(\boldsymbol{Z},\boldsymbol{C})$ w.r.t. $\boldsymbol{C}$ be zero, we get the following one-step update for $\boldsymbol{C}$:

$$\boldsymbol{C}_p^s = (\mathcal{K}(\boldsymbol{Z}^{s-1}, \boldsymbol{Z}^{s-1}) + \lambda\boldsymbol{I}_d)^{-1}\mathcal{K}(\boldsymbol{Z}^{s-1}, \boldsymbol{X}_p), \quad p = 1, 2, \ldots, P. \tag{14}$$

The derivative of $f_p(\boldsymbol{Z},\boldsymbol{C})$ w.r.t. $\boldsymbol{Z}$ is

$$\frac{\partial\mathcal{L}}{\partial\boldsymbol{Z}} = \frac{1}{\sigma^2}(\boldsymbol{X}_p\boldsymbol{W}_Z - \boldsymbol{Z}\bar{\boldsymbol{W}}_Z) + \frac{2}{\sigma^2}(\boldsymbol{Z}\boldsymbol{Q}_Z - \boldsymbol{Z}\bar{\boldsymbol{Q}}_Z), \tag{15}$$

where the intermediate variables are detailed as

$$\boldsymbol{W}_Z = -\boldsymbol{C}^T \odot \mathcal{K}(\boldsymbol{X}_p, \boldsymbol{Z}) \qquad \bar{\boldsymbol{W}}_Z = \text{diag}(\boldsymbol{1}_n^T\boldsymbol{W}_Z)$$
$$\boldsymbol{Q}_Z = (0.5\boldsymbol{C}\boldsymbol{C}^T) \odot \mathcal{K}(\boldsymbol{Z}, \boldsymbol{Z}) \qquad \bar{\boldsymbol{Q}}_Z = \text{diag}(\boldsymbol{1}_d^T\boldsymbol{Q}_Z).$$

Here $\boldsymbol{1}_n$ and $\boldsymbol{1}_d$ are the column vectors with all elements of 1. Because of the kernel function, $\boldsymbol{Z}$ cannot be updated like $\boldsymbol{C}_p$. Here, we use the gradient method to update it. At local iteration $t$, by setting $\boldsymbol{Z}_p^{s,0} = \boldsymbol{Z}^{s-1}$, the update scheme of $\boldsymbol{Z}$ is

$$\boldsymbol{Z}_p^{s,t} = \boldsymbol{Z}_p^{s,t-1} - \eta_t\frac{\partial f_p}{\partial\boldsymbol{Z}}(\boldsymbol{Z}_p^{s,t-1}). \tag{16}$$

where $\eta_t$ is the step size and can be set as the reverse of the Lipschitz constant of gradient if possible.

We summarize the optimization details in Algorithm 1 (shown in Appendix A).

## 2.4 Convergence Analysis of The Proposed Algorithm

First of all, it is obvious that all local objective functions $f_p(\cdot, \cdot)$ for $p = 1, \ldots, P$ are lower bounded. To analyze the convergence of Algorithm 1, we make two assumptions. The first one is the Lipschitz continuity of the gradient of the local objective functions.

**Assumption 2.1.** The gradients of all local objective functions $f_p(\cdot, \cdot)$ for $p = 1, \ldots, P$ are $L_{Z_p}^s$-Lipschitz continuous in $\boldsymbol{Z}$, that is

$$\left\|\nabla_Z f_p(\boldsymbol{Z}^{s,t}, \boldsymbol{C}_p^s) - \nabla_Z f_p(\boldsymbol{Z}^{s,t-1}, \boldsymbol{C}_p^s)\right\|_F \le L_{Z_p}^s \left\|\boldsymbol{Z}^{s,t} - \boldsymbol{Z}^{s,t-1}\right\|_F. \tag{17}$$

In addition, there exist some lower and upper bounds for $L_{Z_p}^s$, i.e., $0 < \underline{L}_Z \le L_{Z_p}^s \le \overline{L}_Z$ hold for all $p = 1, \ldots, P$ and $s = 1, \ldots, S$.

The second assumption, similar to [Li *et al.*, 2019; Lian *et al.*, 2017], is as follows.

**Assumption 2.2.** The difference between the local gradient and the global gradient is bounded as

$$\left\|\nabla_Z f_p(\boldsymbol{Z}, \boldsymbol{C}_p) - \nabla_Z F(\boldsymbol{Z}, \boldsymbol{C})\right\|_F \le \zeta, \quad \forall\, p = 1, \ldots, P. \tag{18}$$

To build the convergence condition, we define the following iterative terms of $\boldsymbol{Z}^{s,t}$ and $\boldsymbol{C}^s$ for all $t = 1, \ldots, Q$ and $s = 1, \ldots, S$:

$$T_C(\boldsymbol{Z}^{s,0}, \boldsymbol{C}^s) = \sum_{p=1}^{P} \omega_p \left\|\boldsymbol{C}_p^s - \boldsymbol{C}_p^{s-1}\right\|_F^2$$
$$T_Z(\boldsymbol{Z}^{s,t}, \boldsymbol{C}^s) = \left\|\boldsymbol{Z}^{s,t} - \boldsymbol{Z}^{s,t-1}\right\|_F^2 \tag{19}$$

where the instantaneous average $\boldsymbol{Z}^{s,t}$ is defined as

$$\boldsymbol{Z}^{s,t} = \frac{1}{\bar{P}} \sum_{p\in\mathcal{A}^s} \boldsymbol{Z}_p^{s,t}. \tag{20}$$

Based on the above assumptions, we provide the following convergence guarantee for Algorithm 1.

**Theorem 2.3** (Convergence of Algorithm 1). *Suppose Assumption 2.1 and Assumption 2.2 hold. Let $T = S(1 + Q)$ be the total number of global and local rounds. Then the sequence $\{(\boldsymbol{Z}^{s,t}, \boldsymbol{C}^s)\}$ generated by Algorithm 1 with stepsize $1/L_{Z_p}^s$ and $\omega_p = \frac{N_p}{n}$ satisfies*

$$\frac{1}{T}\left[\sum_{s=1}^{S} T_C(\boldsymbol{Z}^{s,0}, \boldsymbol{C}^s) + \sum_{s=1}^{S}\sum_{t=1}^{Q} T_Z(\boldsymbol{Z}^{s,t}, \boldsymbol{C}^s)\right] \leq \frac{D}{T}[F(\boldsymbol{Z}^{1,0}, \boldsymbol{C}^0) - \underline{f}] + \frac{16\zeta^2\psi D}{\bar{P}\underline{L}_Z} \quad (21)$$

*where $\psi = 1 + \frac{(\bar{P}+8)(Q-1)(2Q-1)}{\bar{P}-4(Q-1)^2(1+\overline{L}_Z^2/\underline{L}_Z^2)}$ and $D = \frac{2}{\gamma_{min}+\lambda} + \frac{4}{\underline{L}_Z}$.*

The proof can be found in Appendix E. We see that when $T \to \infty$, the algorithm converges to a finite value, which is small if $\zeta$ is small and $\overline{L}_Z$ is close to $\underline{L}_Z$.

# 3 Security-Enhanced FedSC

In order to enhance the security of FedSC, we present two noise-augmented variants of the proposed algorithm in this section.

## 3.1 FedSC with Perturbed Data

We add random noise to the data in each client and then perform Algorithm 1 to reconstruct a similarity matrix, which further improves the privacy of data. Specifically, the data $\boldsymbol{X} \in \mathbb{R}^{m \times n}$ is perturbed by a noise matrix $\boldsymbol{E} \in \mathbb{R}^{m \times n}$ to form the noisy data matrix

$$\tilde{\boldsymbol{X}} = \boldsymbol{X} + \boldsymbol{E}, \quad (22)$$

where $E_{ij} \sim \mathcal{N}(0, \sigma^2)$. We then perform Algorithm 1 with a Gaussian kernel of parameter $r$ on $\tilde{\boldsymbol{X}}$ and obtain $\boldsymbol{Z}$, $\boldsymbol{C} = [\boldsymbol{C}_1, \boldsymbol{C}_2, \ldots, \boldsymbol{C}_P]$, and

$$\hat{K}_{\tilde{x}\tilde{x}} = \boldsymbol{C}^T \mathcal{K}(\boldsymbol{Z}, \boldsymbol{Z})\boldsymbol{C}. \quad (23)$$

We have the following reconstruction (for the true similarity matrix $\boldsymbol{K}_{xx} = \mathcal{K}(\boldsymbol{X}, \boldsymbol{X})$) error bound[4].

**Theorem 3.1** (Error bound of similarity matrix reconstruction). *Suppose $\|\boldsymbol{X}\|_{2,\infty} = \theta$, $\|\boldsymbol{C}\|_{2,\infty} = \tau_C$, and $\left\|\phi(\boldsymbol{Z})\boldsymbol{C} - \phi(\tilde{\boldsymbol{X}})\right\|_{2,\infty} \leq \gamma$, where $\theta$, $\tau_C$, and $\gamma$ are some nonnegative constants. Then with the probability at least $1 - n(n-1)e^{-t}$, the reconstructed similarity matrix $\hat{\boldsymbol{K}}_{\tilde{x}\tilde{x}}$ satisfies*

$$\left\|\hat{\boldsymbol{K}}_{\tilde{x}\tilde{x}} - \boldsymbol{K}_{xx}\right\|_{\infty} \leq \frac{1}{r^2}\left[(\sigma\xi + \sqrt{2}\theta)^2 - 2\theta^2\right] + (\sqrt{d}\tau_C + 1)\gamma \quad (24)$$

*where $\xi = \sqrt{(m + 2\sqrt{mt} + 2t)}$.*

Note that $r$ is the hyperparameter of the Gaussian kernel. In our experiment, $r$ was automatically estimated as the mean of all pairwise distances between data points, *i.e.*, $r = \frac{1}{n^2}\sum_{i,j}\|\boldsymbol{x}_i - \boldsymbol{x}_j\|_2$. Assume $|x_{ik} - x_{jk}| = \mathcal{O}(\varepsilon)$ for all $i, j \in [n], k \in [m]$, then $\|\boldsymbol{x}_i - \boldsymbol{x}_j\| = \mathcal{O}(\sqrt{m}\varepsilon)$, which means $r^2$ is linear with $m\varepsilon^2$. Thus, the reconstruction error $\|\hat{\boldsymbol{K}}_{\tilde{x}\tilde{x}} - \boldsymbol{K}_{xx}\|_{\infty}$ is upper bounded by $\mathcal{O}(\sigma^2/\varepsilon^2 + \sqrt{d}\gamma\tau_C)$, where $\epsilon^2/\sigma^2$ can be regarded as a signal-noise ratio. Therefore, the bound is useful. In general, Theorem 3.1 indicates that when the added noise is small and the optimization makes $\gamma$ small, the reconstruction error for the true similarity matrix is less than a small constant with high probability. This verified the effectiveness of our similarity reconstruction method.

It should be pointed out that $\hat{\boldsymbol{K}}_{\tilde{x}\tilde{x}}$ is not guaranteed to be a sparse matrix and hence the corresponding graph may not contain multiple connected components. We therefore use an extra KNN-based operation to get a sparse similarity matrix, which may also reduce the computational cost of eigenvalue decomposition when $n$ is very large. Specifically, we let

$$\hat{\boldsymbol{K}}_{\tilde{x}\tilde{x}} = \text{getSparseMatrixbyKNN}(\hat{\boldsymbol{K}}_{\tilde{x}\tilde{x}}, k) \quad (25)$$

---

[4]We defer the proof for all theoretical results to the supplementary material.

which only keeps the largest $k$ connections from each point to other points. Finally, we perform spectral clustering using $\hat{K}_{\tilde{x}\tilde{x}}$. The central server broadcasts the clustering results to each participating client.

As mentioned before, one can choose to inject some noise into its raw data to avoid privacy leakage. However, a question is how much noise we can add to the data to the largest extent for the guarantee of correct clustering. We first present the following definitions.

**Definition 3.2** (Local neighbor set). Suppose $\boldsymbol{x}_i$ and $\boldsymbol{x}_j$ are data points of $\boldsymbol{X} \in \mathbb{R}^{m \times n}$ with class labels $L_i$ and $L_j$ respectively, and let KNN($\boldsymbol{x}_i$) be the set of the k-nearest neighbors of $\boldsymbol{x}_i$. We define

$$\mathcal{N}_i^{k,intra} := \{\boldsymbol{x}_j \in \boldsymbol{X} | L_i = L_j \text{ and } \boldsymbol{x}_j \in \text{KNN}(\boldsymbol{x}_i)\}. \tag{26}$$

**Definition 3.3** (Global neighbor set). Suppose $\boldsymbol{x}_i$ and $\boldsymbol{x}_j$ are data points of $\boldsymbol{X} \in \mathbb{R}^{m \times n}$ with class labels $L_i$ and $L_j$ respectively, and let KNN($\boldsymbol{x}_i$) be the point set of k-nearest neighbors of $\boldsymbol{x}_i$. We define

$$\mathcal{N}_i^{k,global} := \{\boldsymbol{x}_j \in \boldsymbol{X} | \boldsymbol{x}_j \in \text{KNN}(\boldsymbol{x}_i)\}. \tag{27}$$

If we call the local neighbor of data point $\boldsymbol{x}_i$ the *intra-class neighbor* of $\boldsymbol{x}_i$, another definition called *inter-class neighbor* of $\boldsymbol{x}_i$ can be further introduced as follows.

**Definition 3.4** (Inter-class neighbor set). Suppose $\boldsymbol{x}_i$ and $\boldsymbol{x}_j$ are data points of data matrix $\boldsymbol{X} \in \mathbb{R}^{m \times n}$ with class labels, $L_i$ and $L_j$, respectively, and let KNN($\boldsymbol{x}_i$) be the point set of k-nearest neighbors of $\boldsymbol{x}_i$. We define

$$\mathcal{N}_i^{k,inter} := \{x_j \in \boldsymbol{X} | L_i \neq L_j \text{ and } \boldsymbol{x}_j \in \text{KNN}(\boldsymbol{x}_i)\}. \tag{28}$$

Based on the above definitions, the following definition is presented to determine whether a data point can be correctly clustered or not.

**Definition 3.5** (Correct clustering). Suppose $\boldsymbol{x}_i \in \mathbb{R}^m$ and $\boldsymbol{x}_j \in \mathbb{R}^m$ are data points of data matrix $\boldsymbol{X} \in \mathbb{R}^{m \times n}$, $\boldsymbol{x}_i$ is said to be correctly clustered with a tolerance of $\epsilon$ if

a. $\hat{\boldsymbol{K}}_{ij} \geq \max_k(\hat{\boldsymbol{K}}_{ik}^{inter}) - \epsilon$ for any of $\boldsymbol{x}_j \in \mathcal{N}_i^{k,intra}$;

b. $\hat{\boldsymbol{K}}_{ij} \leq \min_k(\hat{\boldsymbol{K}}_{ik}^{intra}) + \epsilon$ for any of $\boldsymbol{x}_j \in \mathcal{N}_i^{k,inter}$.

Based on Definition 3.5, the following theorem gives the guarantee of our security-enhanced FedSC.

**Theorem 3.6** (Guarantee of noisy spectral clustering). *Let $B(\sigma) = \frac{1}{r^2}\left[(\sigma\xi + \sqrt{2}\theta)^2 - 2\theta^2\right] + (\sqrt{d}\tau_C + 1)\gamma$. Then with the probability of at least $1 - n(n-1)e^{-t}$, performing spectral clustering using $\hat{\boldsymbol{K}}_{\tilde{x}\tilde{x}}$ yields correct clustering results if*

$$B(\sigma) \leq \frac{\epsilon}{2} - \max_i \frac{1}{4}\left[\max_k(\boldsymbol{K}_{ik}^{inter}) - \min_k(\boldsymbol{K}_{ik}^{intra})\right] \tag{29}$$

*where $\boldsymbol{K}_{ik}^{inter} = (\boldsymbol{K}_{xx})_{ik}^{inter}$ and $\boldsymbol{K}_{ik}^{intra} = (\boldsymbol{K}_{xx})_{ik}^{intra}$.*

Based on Theorem 3.1 and Theorem 3.6, we can get a bound on the variance of noise for FedSC with perturbed data:

$$\sigma \leq \frac{1}{\xi}\left[\sqrt{r^2(B_1 - B_2) + 2\theta^2} - \sqrt{2}\theta\right] \tag{30}$$

where $B_1 = \frac{\epsilon}{2} - \max_i \frac{1}{4}\left[\max_k(\boldsymbol{K}_{ik}^{inter}) - \min_k(\boldsymbol{K}_{ik}^{intra})\right]$ and $B_2 = (\sqrt{d}\tau_C + 1)\gamma$. This bound indicates that the intensity of noise should not be too strong otherwise it may seriously affect the performance of federated spectral clustering. But, at least under this bound, one can choose to inject as much noise as possible into the raw data to ensure data security and privacy.

Using Theorem 3.22 in [Dwork *et al.*, 2014] and the post-processing property of differential privacy, we have the following privacy guarantee for this enhanced FedSC algorithm.

**Proposition 3.7.** *FedSC with perturbed data given by (22) is $(\varepsilon, \delta)$-differentially private if $\sigma \geq 2c\tau_X/\varepsilon$, where $c^2 > 2\ln(1.25/\delta)$.*

Based on this proposition and (30), we obtain the following privacy-utility trade-off:

$$2\sqrt{2\ln 1.25/\delta}\tau_X/\varepsilon < \sigma \leq \frac{1}{\xi}\left[\sqrt{r^2(B_1 - B_2) + 2\theta^2} - \sqrt{2}\theta\right]. \quad (31)$$

This ensures both clustering performance and $(\varepsilon, \delta)$-differential privacy. In particular, if we substitute $\sigma$ with the upper bound, we can get a strong level of privacy but the worst utility. By the way, $B_1 - B_2$ is related to the property of the data. A larger $B_1 - B_2$ means a better property for clustering, which further provides a larger upper bound for the noise level $\sigma$, yielding a stronger privacy guarantee.

### 3.2 FedSC with Perturbed Factors

In FedSC with perturbed factor, we added Gaussian noise to $\boldsymbol{Z}$ in every round of the optimization but added Gaussian noise to $\boldsymbol{C}$ in the last round of the optimization. To be more specific, $\tilde{\boldsymbol{C}} = \boldsymbol{C} + \boldsymbol{E}_C$, and $\tilde{\boldsymbol{Z}} = \boldsymbol{Z} + \boldsymbol{E}_Z$, where the entries of $\boldsymbol{E}_C$ and $\boldsymbol{E}_Z$ are drawn from $\mathcal{N}(0, \sigma_C^2)$ and $\mathcal{N}(0, \sigma_Z^2)$ respectively. Then we perform spectral clustering using the following reconstructed kernel matrix:

$$\tilde{\boldsymbol{K}}_{xx} = \tilde{\boldsymbol{C}}^T \mathcal{K}(\tilde{\boldsymbol{Z}}, \tilde{\boldsymbol{Z}})\tilde{\boldsymbol{C}}. \quad (32)$$

The following theorem shows the reconstruction error bound for the ground truth kernel matrix $\boldsymbol{K}_{xx}$.

**Theorem 3.8.** *Assume* $\|\phi(\boldsymbol{Z})\boldsymbol{C} - \phi(\boldsymbol{X})\|_{2,\infty} \leq \gamma$, $\|\boldsymbol{C}\|_{2,\infty} \leq \tau_C$. *Then with probability at least* $1 - (n+d)e^{-t}$, *it holds that*

$$\left\|\tilde{\boldsymbol{K}}_{xx} - \boldsymbol{K}_{xx}\right\|_\infty \leq \gamma_{zc}(\gamma_{zc} + 2) \quad (33)$$

*where* $\gamma_{zc} = \gamma + \sqrt{d}\left(\sigma_C\xi_d + \tau_C\sqrt{2\left(1 - \exp\left(-\frac{\sigma_Z^2\xi_d^2}{2r^2}\right)\right)}\right)$ *and* $\xi_d^2 = d + 2\sqrt{dt} + 2t$.

We see that, given a fixed $\gamma$, the reconstruction error becomes smaller if $\sigma_Z$ and $\sigma_C$ are smaller. Based on Theorem 3.8 and Definitions 3.2-3.5, we can obtain a bound similar to that in Theorem 3.6 to guarantee correct clustering, which will not be detailed here.

**Theorem 3.9.** *In FedSC, assume* $\max_{(p,j)}\{\|\boldsymbol{x}_{p_j}\|, \|\boldsymbol{x}'_{p_j}\|\} \leq \tau_X$, $\max_{(i,j)}\|\boldsymbol{z}_i - \boldsymbol{x}_j\|_\infty = \Upsilon$, $\|\boldsymbol{Z}_p^s\|_{sp} \leq \tau_Z$ $\forall s$, *and* $\|\boldsymbol{C}^S\|_{2,\infty} \leq \tau_C$, *we perturb* $\{\boldsymbol{Z}_p^s\}_{p=1}^P$, $\forall s = 1, 2, \ldots, S$ *with noise drawn from* $\mathcal{N}(0, \sigma_Z^2)$ *with the parameter* $\sigma_Z \geq \sqrt{(8S\Delta^2(g_Z)\log(e + (\varepsilon_Z/\delta_Z))/\varepsilon_Z^2}$ *where* $\Delta(g_Z) = \frac{2\sqrt{d}\tau_C\tau_X\eta_k}{r^2}\left\{1 + (\tau_X + \tau_Z)\frac{(\tau_X + \Upsilon)}{r^2}\right\}$ *and perturb* $\{\boldsymbol{C}_p^S\}_{p=1}^P$ *with noise drawn from* $\mathcal{N}(0, \sigma_C^2)$ *with the parameter* $\sigma_C \geq 2c\lambda^{-1}\sqrt{d}\tau_X(\tau_X + \Upsilon)/(r^2\varepsilon_C)$ *for* $c^2 > 2\ln(1.25/\delta_C)$. *Then, FedSC is* $(\varepsilon_C + \varepsilon_Z, \delta_C + \delta_Z)$-*differentially private.*

Theorem 3.9 shows that our FedSC with perturbed factors can protect the data privacy provided that the noises added to $\boldsymbol{C}$ and $\boldsymbol{Z}$ are sufficiently large. Similarly to (31), we can also get a privacy-utility trade-off using Theorem 3.8 and Theorem 3.9, which is detailed in Appendix B.

## 4 Related Work

It should be pointed out that the study on federated spectral clustering in literature is very limited. Besides our work, the only work that aims to address the problem is [Hernández-Pereira *et al.*, 2021]. More introduction and discussion about the related work (federated matrix factorization/clustering [Yang *et al.*, 2021; Ghosh *et al.*, 2020; Dennis *et al.*, 2021; Wang and Chang, 2022] and spectral clustering [Von Luxburg, 2007; Hernández-Pereira *et al.*, 2021]) are in the supplementary material.

## 5 Experiments

### 5.1 Performance on similarity reconstruction

Taking the COIL20 dataset [Nene *et al.*, 1996] as an example, we first obtain the similarity matrix from vanilla spectral clustering based on the same kernel function. Then, we use the proposed method to derive the estimated similarity matrix $\hat{\boldsymbol{K}}_{\tilde{x}\tilde{x}}$ which is actually an approximation of ground truth. To

make it clearer, we also give the corresponding sparse similarity matrices by KNN sparsification (25). Figure 2 shows the similarity matrices constructed by different methods. We see that the proposed method can be able to successfully reconstruct the similarity matrix in the federated scenarios. The reconstruction errors on synthetic data, iris, banknote authentication, and COIL20 are in the supplementary material.

## 5.2 Clustering performance of FedSC

In this subsection, we check the clustering performance of the proposed security-enhanced FedSC method on both synthetic and real-world datasets. The synthetic dataset is generated from concentric circles. The details are in the supplementary. This synthetic dataset is visualized in Figure 3(a). Here, we continue to adopt the aforementioned COIL20 as an example of real-world datasets.

Taking the synthetic dataset as an example, the first group of cases helps illustrate the effectiveness of the proposed FedSC method. we first apply the vanilla spectral clustering method to the clean data. The predictive result is shown in Figure 3(b). It is clear that the vanilla spectral clustering method correctly clusters the data points lying in concentric circles. We then use the proposed FedSC method to cluster the data. One can find in Figure 3(c) that almost all of the data points also have been grouped correctly. However, when we inject some volume of noise

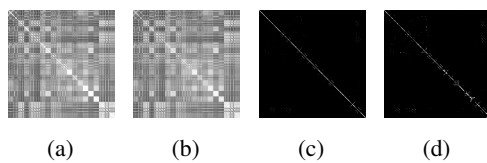

(a)   (b)   (c)   (d)

Figure 2: Visualization of similarity matrices: (a) similarity matrix of vanilla spectral clustering; (b) approximated similarity matrix of the proposed method; (c)(d) the corresponding sparse similarity matrices generated by KNN sparsification.

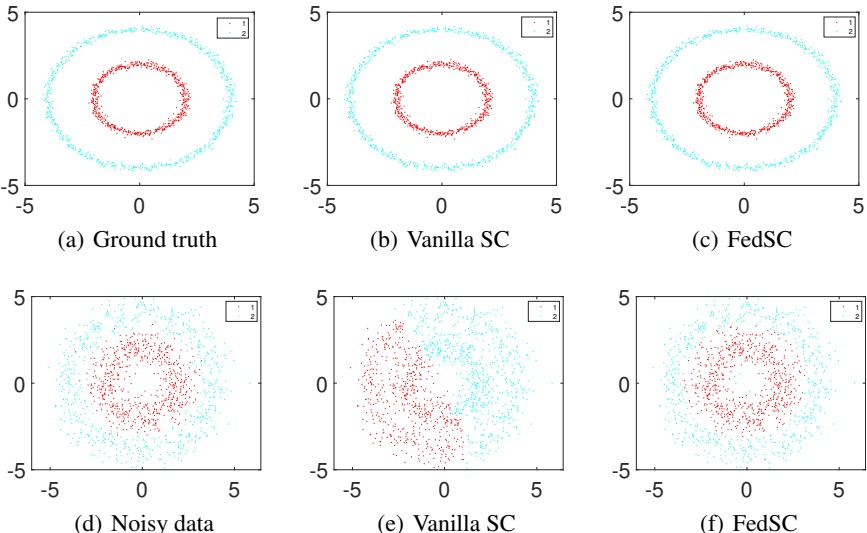

(a) Ground truth   (b) Vanilla SC   (c) FedSC

(d) Noisy data   (e) Vanilla SC   (f) FedSC

Figure 3: FedSC on concentric circles: (a) ground truth; (b) cluster assignment generated by vanilla SC; (c) cluster assignment generated by FedSC; (d) Noisy ground truth; (e) cluster assignment generated by vanilla SC on noisy data; (f) cluster assignment generated by FedSC on noisy data.

into the raw data, things may change a lot. Figure 3(d) is actually the ground truth Figure 3(a) adding some Gaussian noise. When focusing on this noisy data of concentric circles, we see from Figure 3(e) that the vanilla SC failed to cluster the data points while the proposed FedSC method is still able to correctly cluster the data to some extent as in Figure 3(f). As we know, the similarity graph directly constructed from raw data could be very sensitive to each data point. When we add too much noise, the similarity graph may fail to model the local neighborhood relationships which may be the reason why data points in Figure 3(e) are not separable for vanilla SC. Instead, FedSC is based on matrix factorization in the high-dimensional feature space and has a potential denoising effect. Therefore, it is possible for our method to achieve a better performance.The visualization of COIL20 can be found in the supplementary material.

## 5.3 Comparison with baselines

We compare our method with the clustering method DSC proposed by [Hernández-Pereira et al., 2021]. Because the existing literature on federated spectral clustering is rare, we here select both classic K-means and spectral clustering as the baselines. Two metrics including accuracy and NMI are adopted to evaluate the clustering results on four datasets including iris [Dua and Graff, 2017], COIL20 [Nene et al., 1996], banknote authentication [Dua and Graff, 2017], and USPS [Hull, 1994]. The details are in the supplementary material. Besides the clean data, we also consider adding noise to them to test the performance of methods under the condition of privacy protection. We directly inject Gaussian noise with zero mean and variance $\sigma^2$ to the raw matrix $\boldsymbol{X} \in \mathbb{R}^{m \times n}$ as $\tilde{\boldsymbol{X}} = \boldsymbol{X} + \boldsymbol{E}$ where $\boldsymbol{E}$ is a Gaussian noise matrix, each element $\boldsymbol{E}_{i,j}$ of which is i.i.d. with $\mathcal{N}(0, \sigma^2)$.

Table 1 shows the clustering accuracy. Our FedSC almost always achieves comparable clustering results to vanilla SC. It even outperformed vanilla SC in some cases and K-means in most cases since FedSC has a potential denoising effect by approximating a similarity matrix. More importantly, FedSC significantly outperformed DSC in almost all cases. The reason is that DSC performs spectral clustering on each local dataset, which may lead to very unstable and inaccurate results.

Table 1: Comparison of clustering accuracy ($\boldsymbol{X}$ and $\tilde{\boldsymbol{X}}$ denote the raw data and corrupted data respectively). The results of NMI are in Section D.4.

|  |  | Kmeans | SC | DSC | FedSC |
|---|---|---|---|---|---|
| $\boldsymbol{X}$ | Iris | $0.8933 \pm 0.0000$ | $0.9000 \pm 0.0000$ | $0.5480 \pm 0.0679$ | $0.9000 \pm 0.0031$ |
|  | COIL20 | $0.6113 \pm 0.0534$ | $0.8025 \pm 0.0009$ | $0.1009 \pm 0.0100$ | $0.7828 \pm 0.0231$ |
|  | Bank | $0.6122 \pm 0.0000$ | $0.5918 \pm 0.0000$ | $0.5582 \pm 0.0045$ | $0.7672 \pm 0.1457$ |
|  | USPS | $0.6704 \pm 0.0047$ | $0.6635 \pm 0.0000$ | $0.1686 \pm 0.0014$ | $0.6596 \pm 0.0021$ |
|  | ORL | $0.6325 \pm 0.0270$ | $0.7865 \pm 0.0106$ | $0.1653 \pm 0.0073$ | $0.7235 \pm 0.0170$ |
| $\tilde{\boldsymbol{X}}$ with $0.1\sigma$ | Iris | $0.8940 \pm 0.0152$ | $0.9120 \pm 0.0332$ | $0.4533 \pm 0.0658$ | $0.8993 \pm 0.0299$ |
|  | COIL20 | $0.6283 \pm 0.0484$ | $0.8024 \pm 0.0020$ | $0.0995 \pm 0.0122$ | $0.7790 \pm 0.0213$ |
|  | Bank | $0.6067 \pm 0.0022$ | $0.6067 \pm 0.0944$ | $0.5558 \pm 0.0023$ | $0.7168 \pm 0.1068$ |
|  | USPS | $0.6732 \pm 0.0035$ | $0.6647 \pm 0.0016$ | $0.1690 \pm 0.0007$ | $0.6643 \pm 0.0022$ |
|  | ORL | $0.6195 \pm 0.0329$ | $0.7810 \pm 0.0065$ | $0.1600 \pm 0.0089$ | $0.7323 \pm 0.0250$ |
| $\tilde{\boldsymbol{X}}$ with $0.3\sigma$ | Iris | $0.8420 \pm 0.0274$ | $0.8327 \pm 0.0267$ | $0.4533 \pm 0.0674$ | $0.8427 \pm 0.0404$ |
|  | COIL20 | $0.6422 \pm 0.0366$ | $0.7997 \pm 0.0029$ | $0.0981 \pm 0.0084$ | $0.7793 \pm 0.0240$ |
|  | Bank | $0.6020 \pm 0.0038$ | $0.5859 \pm 0.0105$ | $0.5588 \pm 0.0074$ | $0.6046 \pm 0.0064$ |
|  | USPS | $0.6704 \pm 0.0063$ | $0.6720 \pm 0.0044$ | $0.1673 \pm 0.0019$ | $0.6884 \pm 0.0509$ |
|  | ORL | $0.6098 \pm 0.0167$ | $0.7885 \pm 0.0047$ | $0.1665 \pm 0.0057$ | $0.7417 \pm 0.0280$ |
| $\tilde{\boldsymbol{X}}$ with $0.5\sigma$ | Iris | $0.7740 \pm 0.0252$ | $0.7313 \pm 0.0494$ | $0.3833 \pm 0.0204$ | $0.7540 \pm 0.0336$ |
|  | COIL20 | $0.6389 \pm 0.0296$ | $0.7950 \pm 0.0080$ | $0.1033 \pm 0.0167$ | $0.7403 \pm 0.0294$ |
|  | Bank | $0.6051 \pm 0.0076$ | $0.5923 \pm 0.0094$ | $0.5566 \pm 0.0030$ | $0.6086 \pm 0.0073$ |
|  | USPS | $0.6699 \pm 0.0031$ | $0.7843 \pm 0.0030$ | $0.1683 \pm 0.0017$ | $0.7778 \pm 0.0062$ |
|  | ORL | $0.5983 \pm 0.0295$ | $0.7930 \pm 0.0172$ | $0.1615 \pm 0.0096$ | $0.7107 \pm 0.0345$ |
| $\tilde{\boldsymbol{X}}$ with $0.7\sigma$ | Iris | $0.6500 \pm 0.0420$ | $0.6087 \pm 0.0468$ | $0.3927 \pm 0.0349$ | $0.6120 \pm 0.0455$ |
|  | COIL20 | $0.6220 \pm 0.0627$ | $0.7662 \pm 0.0172$ | $0.0893 \pm 0.0055$ | $0.6803 \pm 0.0198$ |
|  | Bank | $0.6100 \pm 0.0112$ | $0.6046 \pm 0.0144$ | $0.5566 \pm 0.0035$ | $0.6106 \pm 0.0107$ |
|  | USPS | $0.6638 \pm 0.0044$ | $0.7747 \pm 0.0040$ | $0.1675 \pm 0.0004$ | $0.7587 \pm 0.0117$ |
|  | ORL | $0.5723 \pm 0.0360$ | $0.7860 \pm 0.0093$ | $0.1613 \pm 0.0066$ | $0.6910 \pm 0.0232$ |

## 5.4 More numerical result

The tSNE visualization, clustering results in terms of NMI, the performance of FedSC with perturbed factors, etc, are in the supplementary material.

## 6 Conclusion

This paper has proposed a secure kernelized factorization method for federated spectral clustering on distributed data. We provide theoretical guarantees for optimization convergence, correct clustering, and differential privacy. The numerical experiments on synthetic and real image datasets verified the effectiveness of our method. To the best knowledge of the authors, this is the work that successfully addresses the problem of federated spectral clustering. One limitation of this work is that we haven't tested our FedSC on very large datasets, though the moderate-size datasets are sufficient to justify the effectiveness of our FedSC. Note that for large-scale datasets, the bottleneck of clustering is the eigenvalue decomposition of the Laplacian matrix, not our FedSC algorithm.

## Acknowledgments

This work was partially supported by the Youth program 62106211 of the National Natural Science Foundation of China, the General Program JCYJ20210324130208022 of Shenzhen Fundamental Research, the research funding T00120210002 of Shenzhen Research Institute of Big Data, the Guangdong Key Lab of Mathematical Foundations for Artificial Intelligence, and the funding UDF01001770 of The Chinese University of Hong Kong, Shenzhen.

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

## A  Framework of the Proposed Algorithm

In the beginning, $\boldsymbol{Z}$ and $\boldsymbol{C}_p$ for $p = 1, 2, \ldots, P$ are initialized randomly in the central server and clients, respectively. Then, $\boldsymbol{Z}$ is broadcast to every participating client and helps do the alternate updating of $\boldsymbol{Z}_p$ and $\boldsymbol{C}_p$ based on the update schemes (16) and (14). Afterward, the obtained matrix $\boldsymbol{Z}_p$ is sent back to the central server and aggregated for the next round of training. When the tolerance condition is broken, both $\boldsymbol{Z}_p^S$ and $\boldsymbol{C}_p^S$ are sent back to the central server for the subsequent clustering task.

---

**Algorithm 1** Proposed Federated Similarity Reconstruction

---

**Input:** Distributed data $\{\boldsymbol{X}_p, : p \in \mathcal{P} := \{1, 2, \ldots, P\}\}$, clients weights $\{\omega_p : p \in \mathcal{P}\}$.
 1: Initialize $\boldsymbol{Z}^0$ at server side and $\{\boldsymbol{C}_p^0\}_{p=1}^P$ at client sides.
 2: Randomly choose $\mathcal{A}^0 \subseteq \mathcal{P}$ with $|\mathcal{A}^s| = \bar{P}$.
 3: **for** round $s = 1$ to $S$ **do**
 4:      Server side: compute $\boldsymbol{Z}^{s-1} = \frac{1}{\bar{P}} \sum_{p \in \mathcal{A}^{s-1}} \boldsymbol{Z}_p^{s-1}$.
 5:      Broadcast $\boldsymbol{Z}^{s-1}$ to clients $c_p, p \in \mathcal{A}^s$.
 6:      Client side:
 7:      **for** client $p = 1$ to $\bar{P}$ in parallel **do**
 8:          set $\boldsymbol{Z}_p^{s,0} = \boldsymbol{Z}^{s-1}$
 9:          update local variable $\boldsymbol{C}_p^s$:
10:              $\boldsymbol{C}_p^s = (\mathcal{K}(\boldsymbol{Z}^{s,0}, \boldsymbol{Z}^{s,0}) + \lambda \boldsymbol{I}_d)^{-1} \mathcal{K}(\boldsymbol{Z}^{s,0}, \boldsymbol{X}_p)$
11:          update local variable $\boldsymbol{Z}_p^s$:
12:          **for** $t = 1$ to $Q$ **do**
13:              $\boldsymbol{Z}_p^{s,t} = \boldsymbol{Z}_p^{s,t-1} - \eta_s \nabla_{\boldsymbol{Z}} f_p(\boldsymbol{Z}_p^{s,t-1})$
14:          **end for**
15:          denote $\boldsymbol{Z}_p^s = \boldsymbol{Z}_p^{s,Q}$
16:          **if** client $p \in \mathcal{A}^{s-1}$ **then**
17:              upload $\boldsymbol{Z}_p^s$ to the server.
18:          **end if**
19:      **end for**
20:      Randomly choose $\mathcal{A}^s \subseteq \mathcal{P}$ with $|\mathcal{A}^s| = \bar{P}$.
21: **end for**
**Output:** $\boldsymbol{Z}, \boldsymbol{C}_p, p = 1, 2, \ldots, P$

---

## B  More theoretical results about FedSC with perturbed factors

Based on Theorem 3.8 and Theorem 3.6, we can get a bound on the variance of noise for FedSC with perturbed factors:

$$\gamma_{zc}(\sigma_Z, \sigma_C) \le -1 + 2\sqrt{1 + B_1} \tag{34}$$

where $B_1 = \frac{\epsilon}{2} - \max_i \frac{1}{4} \left[ \max_k(\boldsymbol{K}_{ik}^{inter}) - \min_k(\boldsymbol{K}_{ik}^{intra}) \right]$ and $\gamma_{zc}(\sigma_Z, \sigma_C)$ is exactly the $\gamma_{zc}$ in Theorem 3.8 and is clearly a non-decreasing function with respect to $\sigma_Z$ and $\sigma_C$, respectively. Therefore, it is a valid upper bound on both $\sigma_Z$ and $\sigma_C$.

*Proof.* By Theorems 3.8 and 3.6, we have

$$\gamma_{zc}(\gamma_{zc} + 2) \le B_1 \tag{35}$$

That is, we need to solve a quadratic equation $\gamma_{zc}^2 + 2\gamma_{zc} - B_1 = 0$ with both $\gamma_{zc} \ge 0$ and $B_1 \ge 0$. Since the discriminant $\Delta = 4 + 4B_1 \ge 0$, we have two roots $-1 \pm 2\sqrt{1 + B_1}$ for this equation. Thus, it has $0 \le \gamma_{zc} \le -1 + 2\sqrt{1 + B_1}$ as desired. $\qquad\qquad\square$

This bound indicates that the intensity of noise should not be too strong otherwise it may seriously affect the performance of federated spectral clustering. But, at least under this bound, one can choose to inject as much noise as possible into the raw data to ensure data security and privacy.

Based on Theorem 3.9 and (34), we obtain the following privacy-utility trade-off:

$$\begin{cases} \gamma_{zc}(\sigma_Z, \sigma_C) \leq -1 + 2\sqrt{1 + B_1} \\ \sigma_Z \geq \sqrt{(8S\Delta^2(g_Z)\log(e + (\varepsilon_Z/\delta_Z))/\varepsilon_Z^2)} \\ \sigma_C \geq 2c\lambda^{-1}\sqrt{d}\tau_X(\tau_X + \Upsilon)/(r^2\varepsilon_C) \end{cases} \tag{36}$$

This ensures both clustering performance and $(\varepsilon, \delta)$-differential privacy. In particular, if we increase the intensity of either $\sigma_Z$ or $\sigma_C$ to reach the upper bound of $\gamma_{zc}$, we can get a strong level of privacy but the worst utility. By the way, $B_1$ is related to the property of the data. A larger $B_1$ means a better property for clustering, which further provides a larger upper bound for the noise level, yielding a stronger privacy guarantee.

## C  More discussion on the privacy-utility trade-off of FedSC

Although theoretical results are presented in our study to ensure the security of FedSC, we still provide here some insight into methods of using Secure Aggregation or other cryptographic techniques to handle pair-wise client functions (*i.e.*, kernel function in our study). Even though the communication cost will be high, it might be an alternative to reduce the DP noise levels. Specifically, it can be performed as follows.

- Step 1: $\boldsymbol{Z}_p^S$ is posted to the central server;

- Step 2: Central server aggregates $\boldsymbol{Z}_p^S$ to get the global $\boldsymbol{Z}^S$;

- Step 3: Central server computes $\mathcal{K}_{ZZ} = \mathcal{K}(\boldsymbol{Z}^S, \boldsymbol{Z}^S)$ and broadcast it to clients;

- Step 4: For client $p$ and $\boldsymbol{c}_i \in \boldsymbol{C}_p$, if $\boldsymbol{c}_j \in \boldsymbol{C}_p$, then client $p$ directly calculate $\hat{\mathcal{K}}_{ij} = \boldsymbol{c}_i \mathcal{K}_{ZZ}\boldsymbol{c}_j$; if $\boldsymbol{c}_j \in \boldsymbol{C}_{p'}$, then client $p$ firstly encrypts and transfers its $\boldsymbol{c}_i$ to client $p'$, and then client $p'$ also encrypts its $\boldsymbol{c}_j$ and use the cipher text to compute $enc(\hat{\mathcal{K}}_{ij}) = enc(\boldsymbol{c}_i^T)enc(\mathcal{K}_{ZZ})enc(\boldsymbol{c}_j)$; Client $p'$ transfers the result $enc(\hat{\mathcal{K}}_{ij})$ back to client $p$; Client $p$ decrypts $enc(\hat{\mathcal{K}}_{ij})$ to get $\hat{\mathcal{K}}_{ij}$.

- Step 5: Each client $p$ sends its estimated results $\hat{\mathcal{K}}_{ij}$ back to the central server without sending its own $\boldsymbol{C}_p$;

- Step 6: Central Server checks whether these posted results from clients are compatible with each other in case of injection attacks and then performs spectral clustering.

This alternative does not send clients' $\boldsymbol{C}$ to the central server and gives an extra cross-validation process in the central server which may be useful to enhance security.

## D  More details and results of the experiments

It should be pointed out that in Table 2 of the main paper, the signal-noise ratio is as high as 12dB, which means the noise is tiny. The parameter $d$ in our FedSC was automatically determined and has a much larger value than that in the noiseless case. That is why the performance of FedSC in the noisy case is even better than that in the noiseless case of some datasets. In this appendix, we increase the noise level ($\sigma_e = \beta\sigma$, where $\sigma$ denotes the standard deviation of the clean data) and consider one more real dataset.

### D.1  Dataset description

**Synthetic data**  The synthetic data is generated from concentric circles. For $\theta_i \in [0, 2\pi]$, $i = 1, 2, \ldots, 1258$, all the points of this synthetic dataset $\boldsymbol{X} \in \mathbb{R}^{2 \times 1258}$ are generated by

$$\boldsymbol{x}_i = (x_{i1}, x_{i2}) : \begin{cases} x_{i1} = r\cos(\theta_i) + e_{i1} \\ x_{i2} = r\sin(\theta_i) + e_{i2} \end{cases} \tag{37}$$

where $\theta_0 = 0$, $\theta_{1258} = 2\pi$, and the remaining $\theta_i$ are evenly spaced points between $\theta_0$ and $\theta_{1258}$. The additive noise $e_{i1}$ and $e_{i2}$ are drawn from $\mathcal{N}(0, \sigma_e^2)$. We let $\sigma_e = 0.1\sigma_x$, where $\sigma_x$ denotes the standard deviation of the data without the additive noise $\boldsymbol{e}$. In our experiment, we set the hyperparameter $r$ in (37) to 2 and 4.

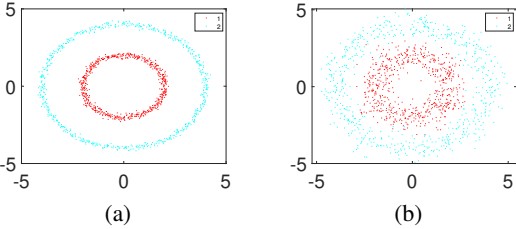

(a)            (b)

Figure 4: Synthetic dataset of 2 concentric circles: (a) ground truth; (b) noisy data perturbed by Gaussian noise with mean zero and standard deviation $0.2\text{std}(\boldsymbol{X})$.

**Real-world data** Both iris and banknote authentication are from the UCI machine learning library. COIL20 is an image dataset from Columbia Imaging and Vision Laboratory. USPS is a dataset for handwritten text recognition research. The details of the mentioned datasets are shown in Table 2.

Table 2: Summary of four real-world datasets

|        | # of clusters | # of attributes | # of instances |
|--------|---------------|-----------------|----------------|
| Iris   | 3             | 4               | 150            |
| COIL20 | 20            | 20×20           | 1440           |
| Bank   | 2             | 5               | 1372           |
| USPS   | 10            | 16×16           | 9298           |
| ORL    | 40            | 92×112          | 400            |

The visualizations (by t-SNE [Van der Maaten and Hinton, 2008]) of three real-world datasets are shown in Figure 5.

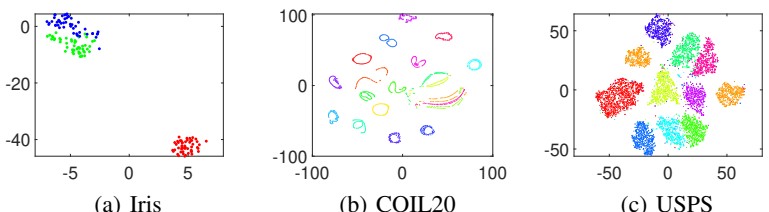

(a) Iris         (b) COIL20         (c) USPS

Figure 5: t-SNE visualization of some real-world datasets

## D.2 Evaluation metrics

We use two metrics to evaluate the clustering performance of our method: accuracy and normalized mutual information (NMI). Between them, accuracy is affected by the misclassification rate of cluster assignment. The smaller the misclassification rate, the greater the accuracy. In this study, given data matrix $\boldsymbol{X} \in \mathbb{R}^{m \times n}$ with $n$ sample points of $m$ features, let $L_i$ and $Lr_i$ for $i = 1, 2, \ldots, n$ be the true labels and predictive labels, respectively, the accuracy of predictive cluster assignment $Lr$ is computed by

$$\text{accuracy} = \frac{1}{n}\text{card}(\{i | L_i = Lr_i, \ i = 1, \ldots, n\}). \tag{38}$$

NMI is a normalized version of mutual information that measures the agreement of two cluster assignments without considering their permutations.

$$\text{NMI} = \frac{2I(L; Lr)}{H(L) + H(Lr)} \tag{39}$$

where $I(L; Lr)$ is the mutual information between $L$ and $Lr$, and $H(L)$ and $H(Lr)$ are the entropy of $L$ and $Lr$, respectively.

### D.3  Parameter settings

In our experiments, we set some hyperparameters including $\lambda_C$, $d$, $r$, and $k$ for implementing the proposed FedSC. Among them, $\lambda_C$ as the penalty parameter of the regularization term is set to $1e - 2$. $k$ is the hyperparameter of the KNN-based operation on the similarity matrix. We set $k$ to $\max(\text{ceil}(\log(n)), 1)$ based on [Von Luxburg, 2007]. The details of methods to set the remaining two parameters are as follows.

**Setting of hyparameter $r$**    The hyperparameter $r$ controls the smoothness of Gaussian kernel function. Due to the distinct characteristics of the datasets, we adopt the following adaptive method to determine the value of $r$:

$$r = c * \mathbf{Mean}(\mathbf{Re}(\sqrt{\boldsymbol{D}})) \tag{40}$$

where $c = 1$ and $\boldsymbol{D}$ is the distance matrix of data points, of which its element

$$\boldsymbol{D}_{ij} = \|\boldsymbol{x}_i - \boldsymbol{x}_j\|_2^2, \quad \text{for } i, j = 1, 2, \ldots, n. \tag{41}$$

In reality, the global $\boldsymbol{D}$ cannot be obtained because FedSC does not allow data transmission. Then we compute $\boldsymbol{D}_p$ for each client $c_p$, $p = 1, \ldots, P$ and then determine $r$ as follows

$$r = \frac{c}{P} \sum_{p=1}^{P} \mathbf{Mean}(\mathbf{Re}(\sqrt{\boldsymbol{D}_p})), \tag{42}$$

where $c$ can be tuned.

**Setting of hyparameter $d$**    The hyperparameter $d$ controls the complexity of the approximation $\phi(\boldsymbol{X}) \simeq \phi(\boldsymbol{Z})\boldsymbol{C}$. We adaptively determine the value of $d$ for each dataset:

$$d = \inf_k \left\{ k \in \mathbb{N} | \sum_{i=1}^{k} s_i \geq \text{tol for } k \leq n \right\} \tag{43}$$

where $s_i$ denotes the $i$-th largest eigenvalue of the kernel matrix $\boldsymbol{K}_{xx}$, $i = 1, \ldots, n$. In our experiment, we set the threshold tol to $0.99$.

It is worth noting that we use the same $r$ in the vanilla SC and our FedSC for a fair comparison, though $\boldsymbol{D}$ cannot be obtained in our FedSC. In addition, the setting of $d$ relies on $\boldsymbol{K}_{xx}$ that is also not obtainable in our FedSC. We use this setting for convenience and in reality we need to determine $d$ by other methods such as letting $d = \gamma m$, where $\gamma$ is a hyperparameter.

### D.4  Clustering results

**NMI results**  The average results of 10 repeated trials are reported in Table 1 (ACC) and Table 3 (NMI). Note that for the USPS dataset, only 5 trials are performed to save time due to its large number of sample points. We see that our FedSC outperformed Kmeans and DSC significantly in almost all cases and has at least comparable performance as SC in most cases.

**Influence of $d$**  The clustering accuracies with different $d$ are reported in Table 4.

**Results of FedSC with perturbed factors**  The results on COIL20 are reported in Table 5, where $\sigma_Z = \alpha_z \text{std}(\boldsymbol{Z}_p)$ and $\sigma_C = \alpha_c \text{std}(\boldsymbol{C})$. Through this experiment, it can be observed that perturbing factors can have a more significant impact on the accuracy of clustering results than perturbing raw data. That is, perturbing factors are more sensitive and can achieve a specified level of differential privacy with weaker noise. Furthermore, it can be seen from Table 5 that the clustering performance is more sensitive to $\sigma_C$ than $\sigma_Z$.

**Malicious attack on $\boldsymbol{C}$**  Due to the kernel trick, the optimization problem is nonlinear and nonconvex. Hence, it is very difficult for potential attackers to recover the data $\boldsymbol{X}$ from the uploaded factors $\boldsymbol{Z}, \boldsymbol{C}$, especially when $\boldsymbol{X}$ or $\boldsymbol{Z}, \boldsymbol{C}$ are perturbed by noise. Nevertheless, the attacker may perform K-means on the $\{\boldsymbol{C}_p\}_{p=1}^{P}$ to obtain clustering results. However, we find that the clustering accuracy

Table 3: Comparison with existing clustering methods (NMI)

| | | NMI | | | |
|---|---|---|---|---|---|
| | | Kmeans | SC | DSC | FedSC |
| $\boldsymbol{X}_0$ | Iris | $0.6356 \pm 0.0000$ | $0.6647 \pm 0.0000$ | $0.2516 \pm 0.0707$ | $0.6708 \pm 0.0139$ |
| | COIL20 | $0.7658 \pm 0.0143$ | $0.8809 \pm 0.0006$ | $0.1259 \pm 0.0225$ | $0.8613 \pm 0.0144$ |
| | Bank | $0.1480 \pm 0.0000$ | $0.1228 \pm 0.0000$ | $0.0122 \pm 0.0160$ | $0.4901 \pm 0.3050$ |
| | USPS | $0.6125 \pm 0.0026$ | $0.8083 \pm 0.0000$ | $0.0064 \pm 0.0016$ | $0.7972 \pm 0.0121$ |
| | ORL | $0.8343 \pm 0.0136$ | $0.8980 \pm 0.0045$ | $0.3970 \pm 0.0045$ | $0.8525 \pm 0.0097$ |
| $\boldsymbol{X}_n$ with $0.1\sigma$ | Iris | $0.6267 \pm 0.0305$ | $0.6878 \pm 0.0764$ | $0.1458 \pm 0.0794$ | $0.6666 \pm 0.0600$ |
| | COIL20 | $0.7721 \pm 0.0122$ | $0.8805 \pm 0.0012$ | $0.1214 \pm 0.0254$ | $0.8529 \pm 0.0209$ |
| | Bank | $0.1383 \pm 0.0039$ | $0.1314 \pm 0.2010$ | $0.0039 \pm 0.0081$ | $0.3853 \pm 0.2390$ |
| | USPS | $0.6147 \pm 0.0022$ | $0.8097 \pm 0.0016$ | $0.0076 \pm 0.0010$ | $0.7920 \pm 0.0133$ |
| | ORL | $0.8280 \pm 0.0153$ | $0.8948 \pm 0.0019$ | $0.3944 \pm 0.0074$ | $0.8554 \pm 0.0104$ |
| $\boldsymbol{X}_n$ with $0.3\sigma$ | Iris | $0.5292 \pm 0.0528$ | $0.5072 \pm 0.0442$ | $0.1403 \pm 0.0792$ | $0.5385 \pm 0.0594$ |
| | COIL20 | $0.7745 \pm 0.0185$ | $0.8767 \pm 0.0014$ | $0.1174 \pm 0.0224$ | $0.8477 \pm 0.0188$ |
| | Bank | $0.1327 \pm 0.0062$ | $0.1031 \pm 0.0211$ | $0.0135 \pm 0.0253$ | $0.1489 \pm 0.0151$ |
| | USPS | $0.6112 \pm 0.0052$ | $0.7927 \pm 0.0123$ | $0.0070 \pm 0.0016$ | $0.7837 \pm 0.0091$ |
| | ORL | $0.8136 \pm 0.0067$ | $0.8974 \pm 0.0047$ | $0.3987 \pm 0.0069$ | $0.8578 \pm 0.0147$ |
| $\boldsymbol{X}_n$ with $0.5\sigma$ | Iris | $0.4316 \pm 0.0310$ | $0.3912 \pm 0.0426$ | $0.0710 \pm 0.0421$ | $0.4046 \pm 0.0388$ |
| | COIL20 | $0.7676 \pm 0.0148$ | $0.8721 \pm 0.0061$ | $0.1238 \pm 0.0286$ | $0.8293 \pm 0.0168$ |
| | Bank | $0.1408 \pm 0.0133$ | $0.1189 \pm 0.0175$ | $0.0059 \pm 0.0109$ | $0.1521 \pm 0.0113$ |
| | USPS | $0.6088 \pm 0.0036$ | $0.8049 \pm 0.0164$ | $0.0059 \pm 0.0028$ | $0.7951 \pm 0.0127$ |
| | ORL | $0.8058 \pm 0.0150$ | $0.8970 \pm 0.0063$ | $0.3977 \pm 0.0095$ | $0.8381 \pm 0.0133$ |
| $\boldsymbol{X}_n$ with $0.7\sigma$ | Iris | $0.3025 \pm 0.0407$ | $0.2755 \pm 0.0381$ | $0.0715 \pm 0.0459$ | $0.2773 \pm 0.0408$ |
| | COIL20 | $0.7589 \pm 0.0234$ | $0.8564 \pm 0.0141$ | $0.1032 \pm 0.0127$ | $0.7772 \pm 0.0196$ |
| | Bank | $0.1506 \pm 0.0208$ | $0.1420 \pm 0.0262$ | $0.0074 \pm 0.0114$ | $0.1577 \pm 0.0196$ |
| | USPS | $0.6007 \pm 0.0051$ | $0.8004 \pm 0.0110$ | $0.0042 \pm 0.0009$ | $0.7636 \pm 0.0184$ |
| | ORL | $0.7877 \pm 0.0165$ | $0.8944 \pm 0.0043$ | $0.3946 \pm 0.0056$ | $0.8236 \pm 0.0102$ |

Table 4: Accurcy of the proposed algorithm on COIL20 (K = 20)

| | d | 7K | 8K | 9K | 194 (SVD) | 10K | 11K | 12K |
|---|---|---|---|---|---|---|---|---|
| Clean data | Trial 1 | 0.7806 | 0.7882 | 0.7951 | 0.7944 | 0.7778 | 0.7979 | 0.7694 |
| | Trial 2 | 0.8035 | 0.7431 | 0.7819 | 0.8007 | 0.7812 | 0.8049 | 0.7792 |
| | Trial 3 | 0.7562 | 0.7896 | 0.7569 | 0.8097 | 0.7708 | 0.7569 | 0.7764 |
| | Trial 4 | 0.7444 | 0.7771 | 0.7771 | 0.7889 | 0.7854 | 0.7903 | 0.7958 |
| | Trial 5 | 0.7965 | 0.7361 | 0.8014 | 0.7833 | 0.7632 | 0.7264 | 0.7694 |
| | Mean | 0.7762 | 0.7668 | 0.7825 | 0.7954 | 0.7757 | 0.7753 | 0.7781 |
| | d | 10K | 20K | 30K | 749 (SVD) | 40K | 50K | 60K |
| Noisy data | Trial 1 | 0.7708 | 0.7722 | 0.7764 | 0.8201 | 0.7000 | 0.7306 | 0.8139 |
| | Trial 2 | 0.7743 | 0.7889 | 0.7646 | 0.7882 | 0.7694 | 0.7451 | 0.7299 |
| | Trial 3 | 0.7375 | 0.7743 | 0.7167 | 0.7493 | 0.7792 | 0.7965 | 0.7299 |
| | Trial 4 | 0.7965 | 0.7688 | 0.8014 | 0.7736 | 0.7903 | 0.7382 | 0.7674 |
| | Trial 5 | 0.7493 | 0.7507 | 0.7576 | 0.7000 | 0.7792 | 0.8076 | 0.8194 |
| | Mean | 0.7657 | 0.7710 | 0.7633 | 0.7662 | 0.7636 | 0.7636 | 0.7721 |

on $\boldsymbol{C}$ is lower than those of Kmeans, SC, and FedSC reported in Table 1. For instance, the clustering accuracy on Iris is reported in Table.

**Influence of $P$** Table 7 compares the clustering performance between using a single client ($P = 1$) and using multiple ones (P = 8). It is clear that the operation of splitting data across multiple clients may lead to an accuracy loss of clustering, which implies that our method is valid.

**More results on MNIST and CIFAR10** We have already included datasets of high-dimensional images in Table 1 like USPS and COIL20 with sizes $16 \times 16$ and $20 \times 20$, respectively. Nevertheless, to further improve the experiment, we added the results of MNIST ($28 \times 28$) and CIFAR10($32 \times 32$) in Table 8.

Table 5: Clustering accuracy (average over 10 trials) of FedSC with perturbed factors on COIL20.

| | | $\alpha_c$ | | | |
|---|---|---|---|---|---|
| | | 0 | 0.05 | 0.1 | 0.15 |
| | | Mean±Std | Mean±Std | Mean±Std | Mean±Std |
| | 0 | 0.7831±0.0268 | 0.6573±0.0291 | 0.6012±0.0391 | 0.4835±0.0526 |
| | 0.05 | 0.7824±0.0126 | 0.6573±0.0243 | 0.6165± 0.0286 | 0.4998±0.0567 |
| $\alpha_z$ | 0.1 | 0.7817±0.0299 | 0.6625±0.0258 | 0.6453±0.0186 | 0.5508±0.0415 |
| | 0.15 | 0.7881±0.0223 | 0.6526±0.0258 | 0.6219±0.0467 | 0.5901±0.0326 |

Table 6: Clustering accuracy among different ways

| | Kmeans | SC | DSC | FedSC | Attack on $C$ |
|---|---|---|---|---|---|
| Iris | $0.8920 \pm 0.0028$ | $0.9000 \pm 0.0000$ | $0.5493 \pm 0.1263$ | $0.9027 \pm 0.0064$ | $0.6360 \pm 0.1557$ |

# E   Proof for theorem on the convergence of FedSC algorithm

We derive the objective descent with respect to $\boldsymbol{C}^s$ and $\boldsymbol{Z}^{s,t}$, respectively.

**Objective descent with w.r.t. $C$:**   Based on the update scheme (14) of $\boldsymbol{C}$, we have

$$\nabla_C f_p(\boldsymbol{Z}^{s,0}, \boldsymbol{C}_p^s) = (\mathcal{K}(\boldsymbol{Z}^{s,0}, \boldsymbol{Z}^{s,0}) + \lambda \boldsymbol{I}_d)\boldsymbol{C}_p^s - \mathcal{K}(\boldsymbol{Z}^{s,0}, \boldsymbol{X}_p) = 0 \qquad (44)$$

where $\boldsymbol{Z}^{s,0} = \boldsymbol{Z}^{s-1} = \frac{1}{P} \sum_{p \in \mathcal{A}^{s-1}} \boldsymbol{Z}_p^{s-1,Q}$.

According to the proposed FedSC problem 7, we have

$$\begin{aligned}
&f_p(\boldsymbol{Z}^{s,0}, \boldsymbol{C}_p^s) - f_p(\boldsymbol{Z}^{s,0}, \boldsymbol{C}_p^{s-1}) \\
&= \left[\frac{1}{2} \left\|\phi(\boldsymbol{X}_p) - \phi(\boldsymbol{Z}^{s,0})\boldsymbol{C}_p^s\right\|_F^2 + \frac{\lambda}{2} \left\|\boldsymbol{C}_p^s\right\|_F^2\right] \\
&\quad - \left[\frac{1}{2} \left\|\phi(\boldsymbol{X}_p) - \phi(\boldsymbol{Z}^{s,0})\boldsymbol{C}_p^{s-1}\right\|_F^2 + \frac{\lambda}{2} \left\|\boldsymbol{C}_p^{s-1}\right\|_F^2\right] \\
&= - \text{Tr}((\boldsymbol{C}_p^s - \boldsymbol{C}_p^{s-1})^T \mathcal{K}(\boldsymbol{Z}^{s,0}, \boldsymbol{X}_p)) + \frac{\lambda}{2}\text{Tr}(\boldsymbol{C}_p^s(\boldsymbol{C}_p^s)^T - \boldsymbol{C}_p^{s-1}(\boldsymbol{C}_p^{s-1})^T) \\
&\quad + \frac{1}{2}\text{Tr}(\left[\boldsymbol{C}_p^s(\boldsymbol{C}_p^s)^T - \boldsymbol{C}_p^{s-1}(\boldsymbol{C}_p^{s-1})^T\right] \mathcal{K}(\boldsymbol{Z}^{s,0}, \boldsymbol{Z}^{s,0})) \\
&= - \text{Tr}((\boldsymbol{C}_p^s - \boldsymbol{C}_p^{s-1})^T (\mathcal{K}(\boldsymbol{Z}^{s,0}, \boldsymbol{Z}^{s,0}) + \lambda \boldsymbol{I}_d)\boldsymbol{C}_p^s) \\
&\quad + \frac{1}{2}\text{Tr}(\left[\boldsymbol{C}_p^s(\boldsymbol{C}_p^s)^T - \boldsymbol{C}_p^{s-1}(\boldsymbol{C}_p^{s-1})^T\right] \left[\mathcal{K}(\boldsymbol{Z}^{s,0}, \boldsymbol{Z}^{s,0}) + \lambda \boldsymbol{I}_d\right]) \\
&= - \frac{1}{2} \underbrace{\text{Tr}([\boldsymbol{C}_p^s - \boldsymbol{C}_p^{s-1}]^T [\mathcal{K}(\boldsymbol{Z}^{s,0}, \boldsymbol{Z}^{s,0}) + \lambda \boldsymbol{I}_d][\boldsymbol{C}_p^s - \boldsymbol{C}_p^{s-1}])}_{\text{T.1}} \\
&\leq - \frac{1}{2}(\gamma_{min}^s + \lambda) \left\|\boldsymbol{C}_p^s - \boldsymbol{C}_p^{s-1}\right\|_F^2
\end{aligned} \qquad (45)$$

where $\gamma_{min}^s = \gamma_{min}(\mathcal{K}(\boldsymbol{Z}^{s,0}, \boldsymbol{Z}^{s,0}))$.

Summing it up from $p = 1$ to $P$, we have

$$F(\boldsymbol{Z}^{s,0}, \boldsymbol{C}^s) - F(\boldsymbol{Z}^{s,0}, \boldsymbol{C}^{s-1}) \leq -\frac{1}{2}(\gamma_{min}^s + \lambda) \sum_{p=1}^P \omega_p \left\|\boldsymbol{C}_p^s - \boldsymbol{C}_p^{s-1}\right\|_F^2 \qquad (46)$$

**Objective descent with w.r.t. $Z$:**   According to Assumption 2.1, it implies

$$\begin{aligned}
F(\boldsymbol{Z}^{s,t}, \boldsymbol{C}^s) - F(\boldsymbol{Z}^{s,t-1}, \boldsymbol{C}^s) &\leq \underbrace{\langle \nabla_Z F(\boldsymbol{Z}^{s,t-1}, \boldsymbol{C}^s), \boldsymbol{Z}^{s,t} - \boldsymbol{Z}^{s,t-1}\rangle}_{\text{T.2}} \\
&\quad + \frac{L_Z^s}{2} \left\|\boldsymbol{Z}^{s,t} - \boldsymbol{Z}^{s,t-1}\right\|_F^2
\end{aligned} \qquad (47)$$

Table 7: Clustering accuracy between using a single client and using multiple clients

|  | Iris | COIL20 | Bank | ORL |
|---|---|---|---|---|
| $P = 1$ | $0.9007 \pm 0.0086$ | $0.8073 \pm 0.0089$ | $0.7536 \pm 0.1271$ | $0.7937 \pm 0.0084$ |
| $P = 8$ | $0.8993 \pm 0.0165$ | $0.8003 \pm 0.0049$ | $0.6821 \pm 0.1405$ | $0.7112 \pm 0.0258$ |

Table 8: Clustering accuracy on MNIST and CIFAR10

|  |  | Kmeans | SC | DSC | FedSC |
|---|---|---|---|---|---|
| **X** | MNIST | $0.5448 \pm 0.0257$ | $0.6265 \pm 0.0439$ | $0.1292 \pm 0.0144$ | $0.6139 \pm 0.0464$ |
|  | CIFAR10 | $0.2171 \pm 0.0132$ | $0.2182 \pm 0.0133$ | $0.1235 \pm 0.0062$ | $0.2134 \pm 0.0131$ |
| **X** with $0.3\sigma$ | MNIST | $0.5402 \pm 0.0225$ | $0.5755 \pm 0.0366$ | $0.1337 \pm 0.0191$ | $0.5606 \pm 0.0457$ |
|  | CIFAR10 | $0.2209 \pm 0.0154$ | $0.2198 \pm 0.0172$ | $0.1194 \pm 0.0051$ | $0.2187 \pm 0.0125$ |
| **X** with $0.7\sigma$ | MNIST | $0.5374 \pm 0.0555$ | $0.5711 \pm 0.0329$ | $0.1340 \pm 0.0135$ | $0.5029 \pm 0.0264$ |
|  | CIFAR10 | $0.2202 \pm 0.0147$ | $0.2205 \pm 0.0084$ | $0.1209 \pm 0.0045$ | $0.2134 \pm 0.0152$ |

Now, we give a bound on T.2.

**Lemma E.1.** *For any $s$ and $t$, it holds that*

$$
\begin{aligned}
\langle \nabla_Z F(\boldsymbol{Z}^{s,t-1}, \boldsymbol{C}^s), \boldsymbol{Z}^{s,t} - \boldsymbol{Z}^{s,t-1} \rangle = & -L_Z^s \left\| \boldsymbol{Z}^{s,t} - \boldsymbol{Z}^{s,t-1} \right\|_F^2 \\
& + \langle \nabla_Z F(\boldsymbol{Z}^{s,t-1}, \boldsymbol{C}^s), \boldsymbol{Z}^{s,t} - \boldsymbol{Z}^{s,t-1} \rangle
\end{aligned}
\tag{48}
$$

*Proof.* Based on the update schemes of $\boldsymbol{Z}$, we have

$$
\boldsymbol{Z}_p^{s,t} = \boldsymbol{Z}_p^{s,t-1} - \frac{1}{L_Z^s} \nabla_Z f_p(\boldsymbol{Z}_p^{s,t-1}, \boldsymbol{C}_p^s)
$$

$$
\Longleftrightarrow \boldsymbol{Z}^{s,t} = \boldsymbol{Z}^{s,t-1} - \frac{1}{\bar{P} L_Z^s} \sum_{p \in \mathcal{A}^s} \nabla_Z f_p(\boldsymbol{Z}_p^{s,t-1}, \boldsymbol{C}_p^s)
\tag{49}
$$

$$
\Longleftrightarrow 0 = \frac{1}{\bar{P}} \sum_{p \in \mathcal{A}^s} \nabla_Z f_p(\boldsymbol{Z}_p^{s,t-1}, \boldsymbol{C}_p^s) + L_Z^s(\boldsymbol{Z}^{s,t} - \boldsymbol{Z}^{s,t-1})
$$

Then, consider the following terms

$$
\begin{aligned}
& \langle \nabla_Z F(\boldsymbol{Z}^{s,t-1}, \boldsymbol{C}^s), \boldsymbol{Z}^{s,t} - \boldsymbol{Z}^{s,t-1} \rangle \\
= & \langle \nabla_Z F(\boldsymbol{Z}^{s,t-1}, \boldsymbol{C}^s) - \frac{1}{\bar{P}} \sum_{p \in \mathcal{A}^s} \nabla_Z f_p(\boldsymbol{Z}_p^{s,t-1}, \boldsymbol{C}_p^s) - L_Z^s(\boldsymbol{Z}^{s,t} - \boldsymbol{Z}^{s,t-1}), \boldsymbol{Z}^{s,t} - \boldsymbol{Z}^{s,t-1} \rangle \\
= & -L_Z^s \left\| \boldsymbol{Z}^{s,t} - \boldsymbol{Z}^{s,t-1} \right\|_F^2 \\
& + \langle \nabla_Z F(\boldsymbol{Z}^{s,t-1}, \boldsymbol{C}^s) - \frac{1}{\bar{P}} \sum_{p \in \mathcal{A}^s} \nabla_Z f_p(\boldsymbol{Z}_p^{s,t-1}, \boldsymbol{C}_p^s), \boldsymbol{Z}^{s,t} - \boldsymbol{Z}^{s,t-1} \rangle
\end{aligned}
\tag{50}
$$

$\square$

Based on Lemma 1, we continue to do the following derivation.

$$
\begin{aligned}
& F(\boldsymbol{Z}^{s,t}, \boldsymbol{C}^s) - F(\boldsymbol{Z}^{s,t-1}, \boldsymbol{C}^s) \\
\leq & -\frac{L_Z^s}{2} \left\| \boldsymbol{Z}^{s,t} - \boldsymbol{Z}^{s,t-1} \right\|_F^2 \\
& + \langle \nabla_Z F(\boldsymbol{Z}^{s,t-1}, \boldsymbol{C}^s) - \frac{1}{\bar{P}} \sum_{p \in \mathcal{A}^s} \nabla_Z f_p(\boldsymbol{Z}_p^{s,t-1}, \boldsymbol{C}_p^s), \boldsymbol{Z}^{s,t} - \boldsymbol{Z}^{s,t-1} \rangle \\
\leq & -\frac{L_Z^s}{4} \left\| \boldsymbol{Z}^{s,t} - \boldsymbol{Z}^{s,t-1} \right\|_F^2 \\
& + \frac{1}{L_Z^s} \underbrace{\left\| \nabla_Z F(\boldsymbol{Z}^{s,t-1}, \boldsymbol{C}^s) - \frac{1}{\bar{P}} \sum_{p \in \mathcal{A}^s} \nabla_Z f_p(\boldsymbol{Z}_p^{s,t-1}, \boldsymbol{C}_p^s) \right\|_F^2}_{\text{T.3}}
\end{aligned}
\tag{51}
$$

where we used the fact that $\langle \boldsymbol{x}, \boldsymbol{y} \rangle \le \|\boldsymbol{x}\|_2^2 + \|\boldsymbol{y}\|_2^2, \forall c > 0$.

Now, we give a bound on T.3.

**Lemma E.2.** *For any $s$ and $t$, it holds that*

$$
\left\| \nabla_Z F(\boldsymbol{Z}^{s,t-1}, \boldsymbol{C}^s) - \frac{1}{\bar{P}} \sum_{p \in \mathcal{A}^s} \nabla_Z f_p(\boldsymbol{Z}_p^{s,t-1}, \boldsymbol{C}_p^s) \right\|_F^2
$$
$$
\le \frac{16\zeta^2}{\bar{P}} + 2(1 + \frac{8}{\bar{P}}) \sum_{p=1}^P \omega_p (L_{Z_p}^s)^2 \left\| \boldsymbol{Z}^{s,t-1} - \boldsymbol{Z}_p^{s,t-1} \right\|_F^2
$$
(52)

*Proof.* For any $s$ and $t$, it holds that

$$
\left\| \nabla_Z F(\boldsymbol{Z}^{s,t-1}, \boldsymbol{C}^s) - \frac{1}{\bar{P}} \sum_{p \in \mathcal{A}^s} \nabla_Z f_p(\boldsymbol{Z}_p^{s,t-1}, \boldsymbol{C}_p^s) \right\|_F^2
$$
$$
= \left\| \nabla_Z F(\boldsymbol{Z}^{s,t-1}, \boldsymbol{C}^s) - \sum_{p=1}^P \omega_p \nabla_Z f_p(\boldsymbol{Z}_p^{s,t-1}, \boldsymbol{C}_p^s) \right.
$$
$$
\left. + \sum_{p=1}^P \omega_p \nabla_Z f_p(\boldsymbol{Z}_p^{s,t-1}, \boldsymbol{C}_p^s) - \frac{1}{\bar{P}} \sum_{p \in \mathcal{A}^s} \nabla_Z f_p(\boldsymbol{Z}_p^{s,t-1}, \boldsymbol{C}_p^s) \right\|_F^2
$$
(53)
$$
\le 2 \underbrace{\left\| \nabla_Z F(\boldsymbol{Z}^{s,t-1}, \boldsymbol{C}^s) - \sum_{p=1}^P \omega_p \nabla_Z f_p(\boldsymbol{Z}_p^{s,t-1}, \boldsymbol{C}_p^s) \right\|_F^2}_{\text{T.4}}
$$
$$
+ 2 \underbrace{\left\| \sum_{p=1}^P \omega_p \nabla_Z f_p(\boldsymbol{Z}_p^{s,t-1}, \boldsymbol{C}_p^s) - \frac{1}{\bar{P}} \sum_{p \in \mathcal{A}^s} \nabla_Z f_p(\boldsymbol{Z}_p^{s,t-1}, \boldsymbol{C}_p^s) \right\|_F^2}_{\text{T.5}}
$$

where we used the fact that $\|\sum_{i=1}^n a_i\|_2^2 \le n \sum_{i=1}^n \|a_i\|_2^2$.

Firstly, we give a bound on T.4:

$$
\left\| \nabla_Z F(\boldsymbol{Z}^{s,t-1}, \boldsymbol{C}^s) - \sum_{p=1}^P \omega_p \nabla_Z f_p(\boldsymbol{Z}_p^{s,t-1}, \boldsymbol{C}_p^s) \right\|_F^2
$$
$$
= \left\| \sum_{p=1}^P \omega_p \left[ \nabla_Z f(\boldsymbol{Z}^{s,t-1}, \boldsymbol{C}_p^s) - \nabla_Z f_p(\boldsymbol{Z}_p^{s,t-1}, \boldsymbol{C}_p^s) \right] \right\|_F^2
$$
(54)
$$
\le \sum_{p=1}^P \omega_p \left\| \nabla_Z F(\boldsymbol{Z}^{s,t-1}, \boldsymbol{C}_p^s) - \nabla_Z f_p(\boldsymbol{Z}_p^{s,t-1}, \boldsymbol{C}_p^s) \right\|_F^2
$$
$$
\le \sum_{p=1}^P \omega_p (L_{Z_p}^s)^2 \left\| \boldsymbol{Z}^{s,t-1} - \boldsymbol{Z}_p^{s,t-1} \right\|_F^2
$$

where we used the fact that $\nabla_Z f_p(\cdot, \boldsymbol{C}_p^s)$ is $L_{Z_p}^s$-Lipschitz continuous.

Secondly, we give a bound on T.5:

$$
\left\| \sum_{p=1}^{P} \omega_p \nabla_Z f_p(\boldsymbol{Z}_p^{s,t-1}, \boldsymbol{C}_p^s) - \frac{1}{\overline{P}} \sum_{p \in \mathcal{A}^s} \nabla_Z f_p(\boldsymbol{Z}_p^{s,t-1}, \boldsymbol{C}_p^s) \right\|_F^2
$$

$$
= \frac{1}{\overline{P}^2} \left\| \sum_{p' \in \mathcal{A}^s} \left[ \sum_{p=1}^{P} \omega_p \nabla_Z f_p(\boldsymbol{Z}_p^{s,t-1}, \boldsymbol{C}_p^s) - \nabla_Z f_{p'}(\boldsymbol{Z}_{p'}^{s,t-1}, \boldsymbol{C}_{p'}^s) \right] \right\|_F^2
$$

$$
\leq \frac{1}{\overline{P}} \sum_{p' \in \mathcal{A}^s} \left\| \sum_{p=1}^{P} \omega_p \nabla_Z f_p(\boldsymbol{Z}_p^{s,t-1}, \boldsymbol{C}_p^s) - \nabla_Z f_{p'}(\boldsymbol{Z}_{p'}^{s,t-1}, \boldsymbol{C}_{p'}^s) \right\|_F^2 \tag{55}
$$

$$
= \frac{1}{\overline{P}} \sum_{p'=1}^{P} \omega_{p'} \left\| \sum_{p=1}^{P} \omega_p \nabla_Z f_p(\boldsymbol{Z}_p^{s,t-1}, \boldsymbol{C}_p^s) - \nabla_Z f_{p'}(\boldsymbol{Z}_{p'}^{s,t-1}, \boldsymbol{C}_{p'}^s) \right\|_F^2
$$

$$
\leq \frac{1}{\overline{P}} \sum_{p'=1}^{P} \omega_{p'} \sum_{p=1}^{P} \omega_p \underbrace{\left\| \nabla_Z f_p(\boldsymbol{Z}_p^{s,t-1}, \boldsymbol{C}_p^s) - \nabla_Z f_{p'}(\boldsymbol{Z}_{p'}^{s,t-1}, \boldsymbol{C}_{p'}^s) \right\|_F^2}_{\text{T.6}}
$$

Thirdly, we give a bound on T.6:

$$
\left\| \nabla_Z f_p(\boldsymbol{Z}_p^{s,t-1}, \boldsymbol{C}_p^s) - \nabla_Z f_{p'}(\boldsymbol{Z}_{p'}^{s,t-1}, \boldsymbol{C}_{p'}^s) \right\|_F^2
$$

$$
= \big\| \left[ \nabla_Z f_p(\boldsymbol{Z}_p^{s,t-1}, \boldsymbol{C}_p^s) - \nabla_Z f_p(\boldsymbol{Z}^{s,t-1}, \boldsymbol{C}_p^s) + \nabla_Z f_p(\boldsymbol{Z}^{s,t-1}, \boldsymbol{C}_p^s) \right.
$$

$$
- \nabla_Z F(\boldsymbol{Z}^{s,t-1}, \boldsymbol{C}^s) + \nabla_Z F(\boldsymbol{Z}^{s,t-1}, \boldsymbol{C}^s) \big]
$$

$$
- \left[ \nabla_Z f_{p'}(\boldsymbol{Z}_{p'}^{s,t-1}, \boldsymbol{C}_{p'}^s) - \nabla_Z f_{p'}(\boldsymbol{Z}^{s,t-1}, \boldsymbol{C}_{p'}^s) + \nabla_Z f_{p'}(\boldsymbol{Z}^{s,t-1}, \boldsymbol{C}_{p'}^s) \right.
$$

$$
\left. - \nabla_Z F(\boldsymbol{Z}^{s,t-1}, \boldsymbol{C}^s) + \nabla_Z F(\boldsymbol{Z}^{s,t-1}, \boldsymbol{C}^s) \right] \big\|_F^2
$$

$$
= \big\| \left[ \nabla_Z f_p(\boldsymbol{Z}_p^{s,t-1}, \boldsymbol{C}_p^s) - \nabla_Z f_p(\boldsymbol{Z}^{s,t-1}, \boldsymbol{C}_p^s) \right]
$$

$$
+ \left[ \nabla_Z f_p(\boldsymbol{Z}^{s,t-1}, \boldsymbol{C}_p^s) - \nabla_Z F(\boldsymbol{Z}^{s,t-1}, \boldsymbol{C}^s) \right]
$$

$$
+ \left[ \nabla_Z f_{p'}(\boldsymbol{Z}_{p'}^{s,t-1}, \boldsymbol{C}_{p'}^s) - \nabla_Z f_{p'}(\boldsymbol{Z}^{s,t-1}, \boldsymbol{C}_{p'}^s) \right] \tag{56}
$$

$$
+ \left[ \nabla_Z f_{p'}(\boldsymbol{Z}^{s,t-1}, \boldsymbol{C}_{p'}^s) - \nabla_Z F(\boldsymbol{Z}^{s,t-1}, \boldsymbol{C}^s) \right] \big\|_F^2
$$

$$
\leq 4 \left\| \nabla_Z f_p(\boldsymbol{Z}_p^{s,t-1}, \boldsymbol{C}_p^s) - \nabla_Z f_p(\boldsymbol{Z}^{s,t-1}, \boldsymbol{C}_p^s) \right\|_F^2
$$

$$
+ 4 \left\| \nabla_Z f_p(\boldsymbol{Z}^{s,t-1}, \boldsymbol{C}_p^s) - \nabla_Z F(\boldsymbol{Z}^{s,t-1}, \boldsymbol{C}^s) \right\|_F^2
$$

$$
+ 4 \left\| \nabla_Z f_{p'}(\boldsymbol{Z}_{p'}^{s,t-1}, \boldsymbol{C}_{p'}^s) - \nabla_Z f_{p'}(\boldsymbol{Z}^{s,t-1}, \boldsymbol{C}_{p'}^s) \right\|_F^2
$$

$$
+ 4 \left\| \nabla_Z f_{p'}(\boldsymbol{Z}^{s,t-1}, \boldsymbol{C}_{p'}^s) - \nabla_Z F(\boldsymbol{Z}^{s,t-1}, \boldsymbol{C}^s) \right\|_F^2
$$

$$
\leq 4(L_{Z_p}^s)^2 \left\| \boldsymbol{Z}^{s,t-1} - \boldsymbol{Z}_p^{s,t-1} \right\|_F^2 + 4(L_{Z_{p'}}^s)^2 \left\| \boldsymbol{Z}^{s,t-1} - \boldsymbol{Z}_{p'}^{s,t-1} \right\|_F^2 + 8\zeta^2
$$

Here, it is the second time that we use the facts that $\| \sum_{i=1}^n a_i \|_2^2 \leq n \sum_{i=1}^n \| a_i \|_2^2$ and that $\nabla_Z f_p(\cdot, \boldsymbol{C}_p^s)$ is $L_{Z_p}^s$-Lipschitz continuous.

Based on the bound of T.6, we derive the bound on $T.5$.

$$
\left\| \sum_{p=1}^{P} \omega_p \nabla_Z f_p(\boldsymbol{Z}_p^{s,t-1}, \boldsymbol{C}_p^s) - \frac{1}{\bar{P}} \sum_{p \in \mathcal{A}^s} \nabla_Z f_p(\boldsymbol{Z}_p^{s,t-1}, \boldsymbol{C}_p^s) \right\|_F^2
$$

$$
\leq \frac{1}{\bar{P}} \sum_{p'=1}^{P} \omega_{p'} \sum_{p=1}^{P} \omega_p \underbrace{\left[ 4(L_{Z_p}^s)^2 \left\| \boldsymbol{Z}^{s,t-1} - \boldsymbol{Z}_p^{s,t-1} \right\|_F^2 + 4(L_{Z_{p'}}^s)^2 \left\| \boldsymbol{Z}^{s,t-1} - \boldsymbol{Z}_{p'}^{s,t-1} \right\|_F^2 + 8\zeta^2 \right]}_{Bound\ of\ T.5}
$$

$$
\leq \frac{8\zeta^2}{\bar{P}} + \frac{8}{\bar{P}} \sum_{p=1}^{P} \omega_p (L_{Z_p}^s)^2 \left\| \boldsymbol{Z}^{s,t-1} - \boldsymbol{Z}_p^{s,t-1} \right\|_F^2
$$

$$
(57)
$$

Fourthly, based on the bounds of $T.4$ and $T.5$, we continue to derive the final required bound.

$$
\left\| \nabla_Z F(\boldsymbol{Z}^{s,t-1}, \boldsymbol{C}^s) - \frac{1}{\bar{P}} \sum_{p \in \mathcal{A}^s} \nabla_Z f_p(\boldsymbol{Z}_p^{s,t-1}, \boldsymbol{C}_p^s) \right\|_F^2
$$

$$
\leq 2 \underbrace{\sum_{p=1}^{P} \omega_p (L_{Z_p}^s)^2 \left\| \boldsymbol{Z}^{s,t-1} - \boldsymbol{Z}_p^{s,t-1} \right\|_F^2}_{Bound\ of\ T.3} + 2 \underbrace{\left[ \frac{8\zeta^2}{\bar{P}} + \frac{8}{\bar{P}} \sum_{p=1}^{P} \omega_p (L_{Z_p}^s)^2 \left\| \boldsymbol{Z}^{s,t-1} - \boldsymbol{Z}_p^{s,t-1} \right\|_F^2 \right]}_{Bound\ of\ T.4}
$$

$$
= \frac{16\zeta^2}{\bar{P}} + 2(1 + \frac{8}{\bar{P}}) \sum_{p=1}^{P} \omega_p (L_{Z_p}^s)^2 \left\| \boldsymbol{Z}^{s,t-1} - \boldsymbol{Z}_p^{s,t-1} \right\|_F^2
$$

$$
(58)
$$

$$\square$$

Based on Lemma 2, we continue to do the following derivation.

$$
F(\boldsymbol{Z}^{s,t}, \boldsymbol{C}^s) - F(\boldsymbol{Z}^{s,t-1}, \boldsymbol{C}^s)
$$

$$
\leq - \frac{L_Z^s}{4} \left\| \boldsymbol{Z}^{s,t} - \boldsymbol{Z}^{s,t-1} \right\|_F^2
$$

$$
+ \frac{1}{L_Z^s} \underbrace{\left[ \frac{16\zeta^2}{\bar{P}} + 2(1 + \frac{8}{\bar{P}}) \sum_{p=1}^{P} \omega_p (L_{Z_p}^s)^2 \left\| \boldsymbol{Z}^{s,t-1} - \boldsymbol{Z}_p^{s,t-1} \right\|_F^2 \right]}_{Bound\ of\ T.3}
$$

$$
(59)
$$

$$
= - \frac{L_Z^s}{4} \left\| \boldsymbol{Z}^{s,t} - \boldsymbol{Z}^{s,t-1} \right\|_F^2 + \frac{16\zeta^2}{\bar{P} L_Z^s}
$$

$$
+ \frac{2}{L_Z^s} (1 + \frac{8}{\bar{P}}) \sum_{p=1}^{P} \omega_p (L_{Z_p}^s)^2 \left\| \boldsymbol{Z}^{s,t-1} - \boldsymbol{Z}_p^{s,t-1} \right\|_F^2
$$

Then, summing it up from $t = 1$ to $Q$ yields

$$
F(\boldsymbol{Z}^{s,Q}, \boldsymbol{C}^s) - F(\boldsymbol{Z}^{s,0}, \boldsymbol{C}^s)
$$

$$
\leq - \frac{L_Z^s}{4} \sum_{t=1}^{Q} \left\| \boldsymbol{Z}^{s,t} - \boldsymbol{Z}^{s,t-1} \right\|_F^2 + \frac{16Q\zeta^2}{\bar{P} L_Z^s}
$$

$$
(60)
$$

$$
+ \frac{2}{L_Z^s} (1 + \frac{8}{\bar{P}}) \underbrace{\sum_{t=1}^{Q} \sum_{p=1}^{P} \omega_p (L_{Z_p}^s)^2 \left\| \boldsymbol{Z}^{s,t-1} - \boldsymbol{Z}_p^{s,t-1} \right\|_F^2}_{T.7}
$$

Now, we give a bound on T.7.

**Lemma E.3.** *For any $s$, it holds that*

$$\left\| \boldsymbol{Z}^{s,t} - \boldsymbol{Z}_p^{s,t} \right\|_F^2$$

$$\leq \frac{4t}{(L_Z^s)^2 \bar{P}} \sum_{j=0}^{t-1} (L_{Z_p}^s)^2 \left\| \boldsymbol{Z}^{s,j} - \boldsymbol{Z}_p^{s,j} \right\|_F^2 + \frac{8t^2 \zeta^2}{(L_Z^s)^2 \bar{P}}$$

$$+ \frac{4t}{(L_Z^s)^2 \bar{P}} \sum_{j=0}^{t-1} \sum_{p'=1}^{P} \omega_{p'} (L_{Z_{p'}}^s)^2 \left\| \boldsymbol{Z}^{s,j} - \boldsymbol{Z}_{p'}^{s,j} \right\|_F^2 \qquad (61)$$

*Proof.* Based on the update schemes of $\boldsymbol{C}$ and $\boldsymbol{Z}$, we have

$$\boldsymbol{Z}_p^{s,t} = \boldsymbol{Z}_p^{s,t-1} - \frac{1}{L_Z^s} \nabla_Z f_p(\boldsymbol{Z}_p^{s,t-1}, \boldsymbol{C}_p^s)$$

$$\iff \boldsymbol{Z}^{s,t} = \boldsymbol{Z}^{s,t-1} - \frac{1}{\bar{P} L_Z^s} \sum_{p \in \mathcal{A}^s} \nabla_Z f_p(\boldsymbol{Z}_p^{s,t-1}, \boldsymbol{C}_p^s) \qquad (62)$$

Consequently, we have

$$\boldsymbol{Z}_p^{s,t} = \boldsymbol{Z}_p^{s,0} - \frac{1}{L_Z^s} \sum_{j=0}^{t-1} \nabla_Z f_p(\boldsymbol{Z}_p^{s,j}, \boldsymbol{C}_p^s)$$

$$\iff \boldsymbol{Z}^{s,t} = \boldsymbol{Z}^{s,0} - \frac{1}{\bar{P} L_Z^s} \sum_{j=0}^{t-1} \sum_{p \in \mathcal{A}^s} \nabla_Z f_p(\boldsymbol{Z}_p^{s,j}, \boldsymbol{C}_p^s) \qquad (63)$$

where $\boldsymbol{Z}_p^{s,0} = \boldsymbol{Z}^{s,0} = \boldsymbol{Z}^{s-1}$.

Based on the above identities,

$$
\begin{aligned}
&\left\| \boldsymbol{Z}^{s,t} - \boldsymbol{Z}_p^{s,t} \right\|_F^2 \\
&= \left\| \left[ \boldsymbol{Z}^{s,0} - \frac{1}{\bar{P}L_Z^s} \sum_{j=0}^{t-1} \sum_{p \in \mathcal{A}^s} \nabla_Z f_p(\boldsymbol{Z}_p^{s,j}, \boldsymbol{C}_p^s) \right] - \left[ \boldsymbol{Z}_p^{s,0} - \frac{1}{L_Z^s} \sum_{j=0}^{t-1} \nabla_Z f_p(\boldsymbol{Z}_p^{s,j}, \boldsymbol{C}_p^s) \right] \right\|_F^2 \\
&= \frac{1}{(L_Z^s)^2} \left\| \frac{1}{\bar{P}} \sum_{j=0}^{t-1} \sum_{p \in \mathcal{A}^s} \nabla_Z f_p(\boldsymbol{Z}_p^{s,j}, \boldsymbol{C}_p^s) - \sum_{j=0}^{t-1} \nabla_Z f_p(\boldsymbol{Z}_p^{s,j}, \boldsymbol{C}_p^s) \right\|_F^2 \\
&\leq \frac{t}{(L_Z^s)^2} \sum_{j=0}^{t-1} \left\| \frac{1}{\bar{P}} \sum_{p \in \mathcal{A}^s} \nabla_Z f_p(\boldsymbol{Z}_p^{s,j}, \boldsymbol{C}_p^s) - \nabla_Z f_p(\boldsymbol{Z}_p^{s,j}, \boldsymbol{C}_p^s) \right\|_F^2 \\
&= \frac{t}{(L_Z^s)^2 \bar{P}^2} \sum_{j=0}^{t-1} \left\| \sum_{p' \in \mathcal{A}^s} \left[ \nabla_Z f_{p'}(\boldsymbol{Z}_{p'}^{s,j}, \boldsymbol{C}_{p'}^s) - \nabla_Z f_p(\boldsymbol{Z}_p^{s,j}, \boldsymbol{C}_p^s) \right] \right\|_F^2 \\
&\leq \frac{t}{(L_Z^s)^2 \bar{P}} \sum_{j=0}^{t-1} \sum_{p' \in \mathcal{A}^s} \left\| \nabla_Z f_{p'}(\boldsymbol{Z}_{p'}^{s,j}, \boldsymbol{C}_{p'}^s) - \nabla_Z f_p(\boldsymbol{Z}_p^{s,j}, \boldsymbol{C}_p^s) \right\|_F^2 \\
&= \frac{t}{(L_Z^s)^2 \bar{P}} \sum_{j=0}^{t-1} \sum_{p'=1}^{P} \omega_{p'} \underbrace{\left\| \nabla_Z f_{p'}(\boldsymbol{Z}_{p'}^{s,j}, \boldsymbol{C}_{p'}^s) - \nabla_Z f_p(\boldsymbol{Z}_p^{s,j}, \boldsymbol{C}_p^s) \right\|_F^2}_{T.6} \\
&\leq \frac{t}{(L_Z^s)^2 \bar{P}} \sum_{j=0}^{t-1} \sum_{p'=1}^{P} \omega_{p'} \underbrace{\left[ 4(L_{Z_p}^s)^2 \left\| \boldsymbol{Z}^{s,j} - \boldsymbol{Z}_p^{s,j} \right\|_F^2 + 4(L_{Z_{p'}}^s)^2 \left\| \boldsymbol{Z}^{s,j} - \boldsymbol{Z}_{p'}^{s,j} \right\|_F^2 + 8\zeta^2 \right]}_{Bound\ of\ T.6} \\
&= \frac{4t}{(L_Z^s)^2 \bar{P}} \sum_{j=0}^{t-1} (L_{Z_p}^s)^2 \left\| \boldsymbol{Z}^{s,j} - \boldsymbol{Z}_p^{s,j} \right\|_F^2 + \frac{8t^2 \zeta^2}{(L_Z^s)^2 m} \\
&\quad + \frac{4t}{(L_Z^s)^2 \bar{P}} \sum_{j=0}^{t-1} \sum_{p'=1}^{P} \omega_{p'} (L_{Z_{p'}}^s)^2 \left\| \boldsymbol{Z}^{s,j} - \boldsymbol{Z}_{p'}^{s,j} \right\|_F^2
\end{aligned} \tag{64}
$$

$\square$

Based on Lemma 3, we give a bound on T.7.

**Lemma E.4.** *For any s, it holds that*

$$
\sum_{t=1}^{Q} \sum_{p=1}^{P} \omega_p (L_{Z_p}^s)^2 \left\| \boldsymbol{Z}^{s,t-1} - \boldsymbol{Z}_p^{s,t-1} \right\|_F^2 \leq \frac{8Q(Q-1)(2Q-1)\zeta^2}{\bar{P} - 4(Q-1)^2(1 + \bar{L}_Z^2 / \underline{L}_Z^2)} \tag{65}
$$

*Proof.* Based on Lemma 3, we have

$$\sum_{t=1}^{Q}\sum_{p=1}^{P}\omega_p(L_{Z_p}^s)^2\left\|\boldsymbol{Z}^{s,t-1}-\boldsymbol{Z}_p^{s,t-1}\right\|_F^2$$

$$\leq\sum_{t=1}^{Q}\sum_{p=1}^{P}\omega_p(L_{Z_p}^s)^2\left[\frac{4(t-1)}{(L_Z^s)^2\bar{P}}\sum_{j=0}^{t-2}(L_{Z_p}^s)^2\left\|\boldsymbol{Z}^{s,j}-\boldsymbol{Z}_p^{s,j}\right\|_F^2+\frac{8(t-1)^2\zeta^2}{(L_Z^s)^2\bar{P}}\right.$$

$$\left.+\frac{4(t-1)}{(L_Z^s)^2\bar{P}}\sum_{j=0}^{t-2}\sum_{p'=1}^{P}\omega_{p'}(L_{Z_{p'}}^s)^2\left\|\boldsymbol{Z}^{s,j}-\boldsymbol{Z}_{p'}^{s,j}\right\|_F^2\right]$$

$$=\sum_{t=1}^{Q}\frac{4(t-1)}{\bar{P}}\sum_{p=1}^{P}\omega_p(L_{Z_p}^s)^2\left(\frac{L_{Z_p}^s}{L_Z^s}\right)^2\sum_{j=0}^{t-2}\left\|\boldsymbol{Z}^{s,j}-\boldsymbol{Z}_p^{s,j}\right\|_F^2+\sum_{t=1}^{Q}\frac{8(t-1)^2\zeta^2}{\bar{P}}$$

$$+\sum_{t=1}^{Q}\frac{4(t-1)}{\bar{P}}\sum_{j=0}^{t-2}\sum_{p'=1}^{P}\omega_{p'}(L_{Z_{p'}}^s)^2\left\|\boldsymbol{Z}^{s,j}-\boldsymbol{Z}_{p'}^{s,j}\right\|_F^2 \qquad (66)$$

$$\leq\sum_{t=1}^{Q}\frac{4(t-1)}{\bar{P}}\sum_{p=1}^{P}\omega_p(L_{Z_p}^s)^2\left(\frac{\overline{L}_Z}{\underline{L}_Z}\right)^2\sum_{j=0}^{t-2}\left\|\boldsymbol{Z}^{s,j}-\boldsymbol{Z}_p^{s,j}\right\|_F^2+\sum_{t=1}^{Q}\frac{8(t-1)^2\zeta^2}{\bar{P}}$$

$$+\sum_{t=1}^{Q}\frac{4(t-1)}{\bar{P}}\sum_{j=0}^{t-2}\sum_{p'=1}^{P}\omega_{p'}(L_{Z_{p'}}^s)^2\left\|\boldsymbol{Z}^{s,j}-\boldsymbol{Z}_{p'}^{s,j}\right\|_F^2$$

$$=\sum_{t=1}^{Q}\frac{4(t-1)}{\bar{P}}(1+\frac{\overline{L}_Z^2}{\underline{L}_Z^2})\sum_{j=0}^{t-2}\sum_{p=1}^{P}\omega_p(L_{Z_p}^s)^2\left\|\boldsymbol{Z}^{s,j}-\boldsymbol{Z}_p^{s,j}\right\|_F^2+\frac{8Q(Q-1)(2Q-1)\zeta^2}{\bar{P}}$$

$$\leq\frac{4(Q-1)^2}{\bar{P}}(1+\frac{\overline{L}_Z^2}{\underline{L}_Z^2})\sum_{t=1}^{Q}\sum_{p=1}^{P}\omega_p(L_{Z_p}^s)^2\left\|\boldsymbol{Z}^{s,t-1}-\boldsymbol{Z}_p^{s,t-1}\right\|_F^2+\frac{8Q(Q-1)(2Q-1)\zeta^2}{\bar{P}}$$

Here, we used the inequality as

$$\sum_{t=1}^{Q}\frac{4(t-1)}{\bar{P}}\sum_{j=0}^{t-2}\left\|\boldsymbol{Z}^{s,j}-\boldsymbol{Z}_p^{s,j}\right\|_F^2\leq\frac{4(Q-1)^2}{\bar{P}}\sum_{t=1}^{Q}\left\|\boldsymbol{Z}^{s,t-1}-\boldsymbol{Z}_p^{s,t-1}\right\|_F^2 \qquad (67)$$

Rearranging the above inequality, we have

$$\sum_{t=1}^{Q}\sum_{p=1}^{P}\omega_p(L_{Z_p}^s)^2\left\|\boldsymbol{Z}^{s,t-1}-\boldsymbol{Z}_p^{s,t-1}\right\|_F^2\leq\frac{8Q(Q-1)(2Q-1)\zeta^2}{\bar{P}-4(Q-1)^2(1+\overline{L}_Z^2/\underline{L}_Z^2)} \qquad (68)$$

$\square$

Based on Lemma 4, we continue to do the following derivation:

$$F(\boldsymbol{Z}^{s,Q}, \boldsymbol{C}^s) - F(\boldsymbol{Z}^{s,0}, \boldsymbol{C}^s)$$

$$\leq -\frac{L_Z^s}{4} \sum_{t=1}^{Q} \left\| \boldsymbol{Z}^{s,t} - \boldsymbol{Z}^{s,t-1} \right\|_F^2 + \frac{16Q\zeta^2}{\bar{P}L_Z^s}$$

$$+ \frac{2}{L_Z^s}(1 + \frac{8}{\bar{P}}) \underbrace{\sum_{t=1}^{Q} \sum_{p=1}^{P} \omega_p (L_{Z_p}^s)^2 \left\| \boldsymbol{Z}^{s,t-1} - \boldsymbol{Z}_p^{s,t-1} \right\|_F^2}_{T.7}$$

$$\leq -\frac{L_Z^s}{4} \sum_{t=1}^{Q} \left\| \boldsymbol{Z}^{s,t} - \boldsymbol{Z}^{s,t-1} \right\|_F^2 + \frac{16Q\zeta^2}{\bar{P}L_Z^s} \qquad (69)$$

$$+ \frac{2}{L_Z^s}(1 + \frac{8}{\bar{P}}) \underbrace{\frac{8Q(Q-1)(2Q-1)\zeta^2}{\bar{P} - 4(Q-1)^2(1 + \overline{L}_Z^2/\underline{L}_Z^2)}}_{Bound\ of\ T.7}$$

$$\leq -\frac{L_Z^s}{4} \sum_{t=1}^{Q} \left\| \boldsymbol{Z}^{s,t} - \boldsymbol{Z}^{s,t-1} \right\|_F^2 + \frac{16Q\zeta^2\psi}{\bar{P}L_Z^s}$$

where $\psi = 1 + \frac{(\bar{P}+8)(Q-1)(2Q-1)}{\bar{P} - 4(Q-1)^2(1+\overline{L}_Z^2/\underline{L}_Z^2)}$.

Then, combining (46) and (69) yields

$$\frac{1}{2}(\gamma_{min}^s + \lambda) \sum_{p=1}^{P} \omega_p \left\| \boldsymbol{C}_p^s - \boldsymbol{C}_p^{s-1} \right\|_F^2 + \frac{L_Z^s}{4} \sum_{t=1}^{Q} \left\| \boldsymbol{Z}^{s,t} - \boldsymbol{Z}^{s,t-1} \right\|_F^2$$

$$\leq [F(\boldsymbol{Z}^{s,0}, \boldsymbol{C}^{s-1}) - F(\boldsymbol{Z}^{s,Q}, \boldsymbol{C}^s)] + \frac{16Q\zeta^2\psi}{\bar{P}L_Z^s} \qquad (70)$$

**Derivation of the main result:** Based on (70), we derive the convergence in terms of the iterative terms, $T_C(\boldsymbol{Z}^{s,0}, \boldsymbol{C}^s)$ and $T_Z(\boldsymbol{Z}^{s,t}, \boldsymbol{C}^s)$ for $s = 1, 2, \ldots, S$.

$$T_C(\boldsymbol{Z}^{s,0}, \boldsymbol{C}^s) \leq \frac{2}{\gamma_{min}^s + \lambda}[F(\boldsymbol{Z}^{s,0}, \boldsymbol{C}^{s-1}) - F(\boldsymbol{Z}^{s,Q}, \boldsymbol{C}^s)] + \frac{32Q\zeta^2\psi}{(\gamma_{min}^s + \lambda)\bar{P}L_Z^s} \qquad (71)$$

Then, summing it up from $s = 1$ to $S$ yields

$$\sum_{s=1}^{S} T_C(\boldsymbol{Z}^{s,0}, \boldsymbol{C}^s) \leq \frac{2}{\underline{\gamma}_{min} + \lambda}[F(\boldsymbol{Z}^{1,0}, \boldsymbol{C}^0) - \underline{f}] + \frac{32SQ\zeta^2\psi}{(\underline{\gamma}_{min} + \lambda)\bar{P}\underline{L}_Z} \qquad (72)$$

Similarly, we have

$$\sum_{t=1}^{Q} T_Z(\boldsymbol{Z}^{s,t}, \boldsymbol{C}^s) \leq \frac{4}{L_Z^s}[F(\boldsymbol{Z}^{s,0}, \boldsymbol{C}^{s-1}) - F(\boldsymbol{Z}^{s,Q}, \boldsymbol{C}^s)] + \frac{64Q\zeta^2\psi}{\bar{P}(L_Z^s)^2} \qquad (73)$$

Then, summing it up from $s = 1$ to $S$ yields

$$\sum_{s=1}^{S} \sum_{t=1}^{Q} T_Z(\boldsymbol{Z}^{s,t}, \boldsymbol{C}^s) \leq \frac{4}{\underline{L}_Z}[F(\boldsymbol{Z}^{1,0}, \boldsymbol{C}^0) - \underline{f}] + \frac{64SQ\zeta^2\psi}{\bar{P}\underline{L}_Z^2} \qquad (74)$$

Lastly, combining (72) and (74) and dividing two sides of it by $T = S(1 + Q)$ yields

$$\frac{1}{T}\left[ \sum_{s=1}^{S} T_C(\boldsymbol{Z}^{s,0}, \boldsymbol{C}^s) + \sum_{s=1}^{S} \sum_{t=1}^{Q} T_Z(\boldsymbol{Z}^{s,t}, \boldsymbol{C}^s) \right] \leq \frac{D}{T}[F(\boldsymbol{Z}^{1,0}, \boldsymbol{C}^0) - \underline{f}] + \frac{16\zeta^2\psi D}{\bar{P}\underline{L}_Z} \qquad (75)$$

where $D = \frac{2}{\underline{\gamma}_{min} + \lambda} + \frac{4}{\underline{L}_Z}$.

# F  Proof for theorem on error bound on noisy similarity matrix

*Proof.* It follows from the triangle inequality of matrix norm that

$$\left\| \hat{K}_{\tilde{x}\tilde{x}} - K_{xx} \right\|_\infty = \left\| \hat{K}_{\tilde{x}\tilde{x}} - K_{\tilde{x}\tilde{x}} + K_{\tilde{x}\tilde{x}} - K_{xx} \right\|_\infty$$
$$\leq \left\| \hat{K}_{\tilde{x}\tilde{x}} - K_{\tilde{x}\tilde{x}} \right\|_\infty + \left\| K_{\tilde{x}\tilde{x}} - K_{xx} \right\|_\infty . \tag{76}$$

Since $\tilde{X} = X + E$ and $\phi(\tilde{X}) = \phi(Z)C$, we have

$$\begin{cases} \hat{K}_{\tilde{x}\tilde{x}} = (\phi(Z)C)^T(\phi(Z)C) = C^T\mathcal{K}(Z,Z)C \\ K_{\tilde{x}\tilde{x}} = \phi(\tilde{X})^T\phi(\tilde{X}) = \mathcal{K}(\tilde{X},\tilde{X}) \\ K_{xx} = \phi(X)^T\phi(X) = \mathcal{K}(X,X) \end{cases} \tag{77}$$

where $E$ is a Gaussian noise matrix, of which $E_{i,j} \sim \mathcal{N}(0,\sigma^2)$. Hence, we obtain

$$\left\| \hat{K}_{\tilde{x}\tilde{x}} - K_{xx} \right\|_\infty \leq \left\| C^T\mathcal{K}(Z,Z)C - \mathcal{K}(\tilde{X},\tilde{X}) \right\|_\infty + \left\| \mathcal{K}(\tilde{X},\tilde{X}) - \mathcal{K}(X,X) \right\|_\infty . \tag{78}$$

Denote by $x_i$ (or $e_i$) the $i$-th column of $X \in \mathbb{R}^{m \times n}$ (or $E \in \mathbb{R}^{m \times n}$), $i = 1, \ldots, n$, we first derive the upper bound of the second term of the RHS of (78) as follows:

$$\begin{aligned} \|K_{\tilde{x}\tilde{x}} - K_{xx}\|_\infty &= \left\| \mathcal{K}(\tilde{X},\tilde{X}) - \mathcal{K}(X,X) \right\|_\infty \\ &= \max_{i,j} |\mathcal{K}(\tilde{x}_i,\tilde{x}_j) - \mathcal{K}(x_i,x_j)| \\ &= \max_{i,j} \left| \exp\left( -\frac{\|(x_i+e_i)-(x_j+e_j)\|_2^2}{2r^2} \right) - \exp\left( -\frac{\|x_i-x_j\|_2^2}{2r^2} \right) \right| \\ &\leq \max_{i,j} \frac{1}{2r^2} \left| -\|(x_i+e_i)-(x_j+e_j)\|_2^2 + \|x_i-x_j\|_2^2 \right| \\ &= \max_{i,j} \frac{1}{2r^2} \left| \|(x_i-x_j)+(e_i-e_j)\|_2^2 - \|x_i-x_j\|_2^2 \right| \\ &= \max_{i,j} \frac{1}{2r^2} \left| \|e_i-e_j\|_2^2 + 2\langle x_i-x_j, e_i-e_j \rangle \right| \\ &\leq \max_{i,j} \frac{1}{2r^2} \left( \|e_i-e_j\|_2^2 + 2|\langle x_i-x_j, e_i-e_j \rangle| \right) \\ &\leq \max_{i,j} \frac{1}{2r^2} \left( \|e_i-e_j\|_2^2 + 2\|x_i-x_j\|_2 \|e_i-e_j\|_2 \right), \end{aligned} \tag{79}$$

where for the first inequality we have used the fact that the exponential function is locally Lipschitz continuous, i.e.,

$$|e^x - e^y| < |x - y| \text{ for } x, y < 0.$$

Now, let us figure out the upper bound of $\|e_i - e_j\|_2^2$. Note that

$$\|e_i - e_j\|_2^2 = \sum_{l=1}^m (e_{li} - e_{lj})^2 = 2\sigma^2 \sum_{l=1}^m \left( \frac{e_{li} - e_{lj}}{\sqrt{2}\sigma} \right)^2 \tag{80}$$

where $e_{li}$ represents the $l$-th element of the column vector $e_i$, $k = 1, \ldots, m$. It is clear that for $l = 1, \ldots, m$,

$$\mathbb{E}[e_{li} - e_{lj}] = 0,$$
$$\text{var}[e_{li} - e_{lj}] = 2\sigma^2. \tag{81}$$

Hence, $\frac{e_{li}-e_{lj}}{\sqrt{2}\sigma}$ is a standard Gaussian random variable drawn from $\mathcal{N}(0,1)$. Based on this, we can define a random variable as

$$Q = \sum_{l=1}^m \left( \frac{e_{li} - e_{lj}}{\sqrt{2}\sigma} \right)^2, \tag{82}$$

which is distributed according to the Chi-squared distribution with $m$ degrees of freedom.

From Laurent and Massart [2000], we know that for any positive $t$, the Chi-squared variable $Q$ with $m$ degrees of freedom satisfies

$$\mathbb{P}(Q > m + 2\sqrt{mt} + 2t) \leq 1 - e^{-t}. \tag{83}$$

Hence, a bound on $\|\boldsymbol{e}_i - \boldsymbol{e}_j\|_2^2$ with probability $1 - e^{-t}$ is

$$\|\boldsymbol{e}_i - \boldsymbol{e}_j\|_2^2 = 2\sigma^2 Q \leq 2\sigma^2(m + 2\sqrt{mt} + 2t). \tag{84}$$

Using union bound for (84), we have

$$\max_{i,j} \|\boldsymbol{e}_i - \boldsymbol{e}_j\|_2^2 = 2\sigma^2 Q \leq 2\sigma^2(m + 2\sqrt{mt} + 2t). \tag{85}$$

holds with probability at least $1 - n(n-1)e^{-t}$. Assume $\|\boldsymbol{x}_i\|_2 \leq \theta$, then $\|\boldsymbol{x}_i - \boldsymbol{x}_i\|_2 \leq 2\theta$. For convenience, let $\xi = \sqrt{m + 2\sqrt{mt} + 2t}$. It follows from (79) and (85) that, with probability at least $1 - n(n-1)e^{-t}$,

$$\begin{aligned}
\|\boldsymbol{K}_{\tilde{x}\tilde{x}} - \boldsymbol{K}_{xx}\|_\infty &\leq \frac{1}{2r^2}\left[2\sigma^2\xi^2 + 2\|\boldsymbol{x}_i - \boldsymbol{x}_j\|_2\sqrt{2}\sigma\xi\right] \\
&\leq \frac{1}{r^2}\left[\sigma^2\xi^2 + 2\sqrt{2}\sigma\xi\theta\right] \\
&= \frac{1}{r^2}\left[(\sigma\xi + \sqrt{2}\theta)^2 - 2\theta^2\right].
\end{aligned} \tag{86}$$

Now, we figure out the upper bound of the first term of the RHS of (78). We have

$$\begin{aligned}
\left\|\hat{\boldsymbol{K}}_{\tilde{x}\tilde{x}} - \boldsymbol{K}_{\tilde{x}\tilde{x}}\right\|_\infty &= \left\|\boldsymbol{C}^T\mathcal{K}(\boldsymbol{Z}, \boldsymbol{Z})\boldsymbol{C} - \mathcal{K}(\tilde{\boldsymbol{X}}, \tilde{\boldsymbol{X}})\right\|_\infty \\
&= \left\|(\phi(\boldsymbol{Z})\boldsymbol{C})^T(\phi(\boldsymbol{Z})\boldsymbol{C}) - \phi(\tilde{\boldsymbol{X}})^T\phi(\tilde{\boldsymbol{X}})\right\|_\infty \\
&= \left\|(\phi(\boldsymbol{Z})\boldsymbol{C})^T(\phi(\boldsymbol{Z})\boldsymbol{C}) - (\phi(\boldsymbol{Z})\boldsymbol{C})^T\phi(\tilde{\boldsymbol{X}}) + (\phi(\boldsymbol{Z})\boldsymbol{C})^T\phi(\tilde{\boldsymbol{X}}) - \phi(\tilde{\boldsymbol{X}})^T\phi(\tilde{\boldsymbol{X}})\right\|_\infty \\
&= \left\|(\phi(\boldsymbol{Z})\boldsymbol{C})^T(\phi(\boldsymbol{Z})\boldsymbol{C} - \phi(\tilde{\boldsymbol{X}})) + (\phi(\boldsymbol{Z})\boldsymbol{C} - \phi(\tilde{\boldsymbol{X}}))^T\phi(\tilde{\boldsymbol{X}})\right\|_\infty \\
&\leq \left\|(\phi(\boldsymbol{Z})\boldsymbol{C})^T(\phi(\boldsymbol{Z})\boldsymbol{C} - \phi(\tilde{\boldsymbol{X}}))\right\|_\infty + \left\|(\phi(\boldsymbol{Z})\boldsymbol{C} - \phi(\tilde{\boldsymbol{X}}))^T\phi(\tilde{\boldsymbol{X}})\right\|_\infty \\
&\leq \|(\phi(\boldsymbol{Z})\boldsymbol{C})\|_{2,\infty} \cdot \left\|\phi(\boldsymbol{Z})\boldsymbol{C} - \phi(\tilde{\boldsymbol{X}})\right\|_{2,\infty} + \left\|\phi(\boldsymbol{Z})\boldsymbol{C} - \phi(\tilde{\boldsymbol{X}})\right\|_{2,\infty} \cdot \left\|\phi(\tilde{\boldsymbol{X}})\right\|_{2,\infty} \\
&= (\|(\phi(\boldsymbol{Z})\boldsymbol{C})\|_{2,\infty} + \left\|\phi(\tilde{\boldsymbol{X}})\right\|_{2,\infty}) \left\|\phi(\boldsymbol{Z})\boldsymbol{C} - \phi(\tilde{\boldsymbol{X}})\right\|_{2,\infty}
\end{aligned} \tag{87}$$

where $\|\cdot\|_{2,\infty}$ is a norm such that $\|\boldsymbol{X}\|_{2,\infty} = \max_i \|\boldsymbol{X}_{:,i}\|_2$ for a real matrix $\boldsymbol{X}$. Here we can just assume that $\left\|\phi(\boldsymbol{Z})\boldsymbol{C} - \phi(\tilde{\boldsymbol{X}})\right\|_{2,\infty} \leq \gamma$, $\gamma$ is some constant; this relies on the optimization.

Moreover, assume $\|\boldsymbol{C}\|_2 \leq \tau_C$, we have

$$\begin{aligned}
\left\|\phi(\tilde{\boldsymbol{X}})\right\|_{2,\infty} &= 1 \\
\|\phi(\boldsymbol{Z})\boldsymbol{C}\|_{2,\infty} &\leq \|\phi(\boldsymbol{Z})\|_F \max_j \|\boldsymbol{C}_{:,j}\| \leq \sqrt{d}\tau_C
\end{aligned} \tag{88}$$

Hence, we can continue to do the derivation of the preceding inequality.

$$\begin{aligned}
\left\|\hat{\boldsymbol{K}}_{\tilde{x}\tilde{x}} - \boldsymbol{K}_{\tilde{x}\tilde{x}}\right\|_\infty &\leq \left(\|\phi(\boldsymbol{Z})\boldsymbol{C}\|_{2,\infty} + \left\|\phi(\tilde{\boldsymbol{X}})\right\|_{2,\infty}\right) \left\|\phi(\boldsymbol{Z})\boldsymbol{C} - \phi(\tilde{\boldsymbol{X}})\right\|_{2,\infty} \\
&\leq (\sqrt{d}\tau_C + 1)\gamma
\end{aligned} \tag{89}$$

As a result, the overall bound is

$$\left\| \hat{\boldsymbol{K}}_{\tilde{x}\tilde{x}} - \boldsymbol{K}_{xx} \right\|_\infty \leq \left\| \hat{\boldsymbol{K}}_{\tilde{x}\tilde{x}} - \boldsymbol{K}_{\tilde{x}\tilde{x}} \right\|_\infty + \left\| \boldsymbol{K}_{\tilde{x}\tilde{x}} - \boldsymbol{K}_{xx} \right\|_\infty$$
$$\leq \frac{1}{r^2}\left[ (\sigma\xi + \sqrt{2}\theta)^2 - 2\theta^2 \right] + (\sqrt{d}\tau_C + 1)\gamma \tag{90}$$

where $\xi = \sqrt{(m + 2\sqrt{mt} + 2t)}$.

$\square$

## G  Proof for Theorem 3.6

*Proof.* For a data matrix $\boldsymbol{X} \in \mathbb{R}^{m \times n}$, it is perturbed by a Gaussian noise matrix $\boldsymbol{E} \in \mathbb{R}^{m \times n}$ with $\mathcal{N}(0, \sigma^2)$ to form the noisy data matrix $\tilde{\boldsymbol{X}} = \boldsymbol{X} + \boldsymbol{E}$. Let $\boldsymbol{K}_{xx} = \mathcal{K}(\boldsymbol{X}, \boldsymbol{X})$ be the ground truth and $\hat{\boldsymbol{K}}_{\tilde{x}\tilde{x}} = \boldsymbol{C}^T \mathcal{K}(\boldsymbol{Z}, \boldsymbol{Z})\boldsymbol{C}$ be the approximated similarity matrix by FedKMF algorithm 1. Then, consider any two data points in $\boldsymbol{X}$, $\boldsymbol{x}_i$ and $\boldsymbol{x}_j$, with the identical label, we know that

$$\boldsymbol{K}_{iu} \leq \boldsymbol{K}_{ij} \tag{91}$$

where $\boldsymbol{K}_{iu} = \max_k((\boldsymbol{K}_{xx})_{ik}^{inter})$ and $\boldsymbol{K}_{ij} = (\boldsymbol{K}_{xx})_{ij}$.

After running FedKMF (Algorithm 1) on the noisy matrix $\tilde{\boldsymbol{X}}$, if the approximated similarity matrix satisfies $\left\| \hat{\boldsymbol{K}}_{\tilde{x}\tilde{x}} - \boldsymbol{K}_{xx} \right\| < B(\sigma)$, then consider two points in $\tilde{\boldsymbol{X}}$, $\tilde{\boldsymbol{x}}_i$ and $\tilde{\boldsymbol{x}}_j$, that are actually $\boldsymbol{x}_i$ and $\boldsymbol{x}_j$ perturbed by some noise, we have

$$\hat{\boldsymbol{K}}_{iu} - \hat{\boldsymbol{K}}_{ij} = \hat{\boldsymbol{K}}_{iu} - \boldsymbol{K}_{iu} + \boldsymbol{K}_{iu} - \hat{\boldsymbol{K}}_{ij} + \boldsymbol{K}_{ij} - \boldsymbol{K}_{ij}$$
$$= (\hat{\boldsymbol{K}}_{iu} - \boldsymbol{K}_{iu}) + (\boldsymbol{K}_{ij} - \hat{\boldsymbol{K}}_{ij}) + (\boldsymbol{K}_{iu} - \boldsymbol{K}_{ij})$$
$$\leq |\hat{\boldsymbol{K}}_{iu} - \boldsymbol{K}_{iu}| + |\boldsymbol{K}_{ij} - \hat{\boldsymbol{K}}_{ij}| + (\boldsymbol{K}_{iu} - \boldsymbol{K}_{ij}) \tag{92}$$
$$\leq B(\sigma) + B(\sigma) + (\boldsymbol{K}_{iu} - \boldsymbol{K}_{ij})$$
$$= 2B(\sigma) + (\boldsymbol{K}_{iu} - \boldsymbol{K}_{ij})$$

where $\hat{\boldsymbol{K}}_{iu} = \max_k((\hat{\boldsymbol{K}}_{\tilde{x}\tilde{x}})_{ik}^{inter})$ and $\hat{\boldsymbol{K}}_{ij} = (\hat{\boldsymbol{K}}_{\tilde{x}\tilde{x}})_{ij}$.

Based on Definition 3.5, $\tilde{\boldsymbol{x}}_i$ and $\tilde{\boldsymbol{x}}_j$ can be correctly clustered only if the following inequality holds with some tolerance $\epsilon > 0$.

$$\hat{\boldsymbol{K}}_{iu} - \hat{\boldsymbol{K}}_{ij} \leq \epsilon \tag{93}$$

Thus, combining inequalities (92) and (93), the bound function $B(\sigma)$ satisfies

$$B(\sigma) \leq \frac{1}{2}[\epsilon - (\boldsymbol{K}_{iu} - \boldsymbol{K}_{ij})] \tag{94}$$

Similarly, if we consider any two data points in $\boldsymbol{X}$, $\boldsymbol{x}_i$ and $\boldsymbol{x}_j$, with different labels, we know that

$$\boldsymbol{K}_{ij} \leq \boldsymbol{K}_{iv} \tag{95}$$

where $\boldsymbol{K}_{iv} = \min_k((\boldsymbol{K}_{xx})_{ik}^{intra})$.

After running FedKMF algorithm 1 on the noisy matrix $\tilde{\boldsymbol{X}}$, we have

$$\hat{\boldsymbol{K}}_{ij} - \hat{\boldsymbol{K}}_{iv} = \hat{\boldsymbol{K}}_{ij} - \boldsymbol{K}_{ij} + \boldsymbol{K}_{ij} - \hat{\boldsymbol{K}}_{iv} + \boldsymbol{K}_{iv} - \boldsymbol{K}_{iv}$$
$$= (\hat{\boldsymbol{K}}_{ij} - \boldsymbol{K}_{ij}) + (\boldsymbol{K}_{iv} - \hat{\boldsymbol{K}}_{iv}) + (\boldsymbol{K}_{ij} - \boldsymbol{K}_{iv})$$
$$\leq |\hat{\boldsymbol{K}}_{ij} - \boldsymbol{K}_{ij}| + |\boldsymbol{K}_{iv} - \hat{\boldsymbol{K}}_{iv}| + (\boldsymbol{K}_{ij} - \boldsymbol{K}_{iv}) \tag{96}$$
$$\leq B(\sigma) + B(\sigma) + (\boldsymbol{K}_{ij} - \boldsymbol{K}_{iv})$$
$$= 2B(\sigma) + (\boldsymbol{K}_{ij} - \boldsymbol{K}_{iv})$$

where $\hat{\boldsymbol{K}}_{iu} = \max_k((\hat{\boldsymbol{K}}_{\tilde{x}\tilde{x}})_{ik}^{intra})$.

Based on Definition 3.5, $\tilde{x}_i$ and $\tilde{x}_j$ can be correctly clustered only if the following inequality holds with some tolerance $\epsilon > 0$.

$$\hat{K}_{ij} - \hat{K}_{iv} \le \epsilon \tag{97}$$

Thus, combining inequalities (96) and (97), the bound function $B(\sigma)$ satisfies

$$B(\sigma) \le \frac{1}{2}[\epsilon - (K_{ij} - K_{iv})] \tag{98}$$

Then, with two upper bounds on $B(\sigma)$, (94) and (98), we have

$$B(\sigma) \le \frac{1}{2} \min_i \{\epsilon - (K_{iu} - K_{ij}), \epsilon - (K_{ij} - K_{iv})\}. \tag{99}$$

where $\epsilon$ is the parameter of tolerance.

Alternatively, a slightly looser version is like

$$B(\sigma) \le \min_i \frac{1}{4}[2\epsilon - (K_{iu} - K_{iv})]$$
$$\text{or} \quad B(\sigma) \le \frac{\epsilon}{2} - \max_i \frac{1}{4}(K_{iu} - K_{iv}). \tag{100}$$

where $K_{iu} = \max_k((K_{xx})_{ik}^{inter})$ and $K_{iv} = \min_k((K_{xx})_{ik}^{intra})$.

$\square$

# H    Proof for Proposition 3.7

*Proof.* The $\ell_2$-sensitivity [Dwork *et al.*, 2014] of a function $f : \mathbb{N}^{|\mathcal{X}|} \to \mathbb{R}^k$ is:
$$\Delta_2(f) = \max_{x \sim y} \|f(x) - f(y)\|_2,$$

where $x \sim y$ denotes that $x$ and $y$ are neighboring datasets. In our case, the function is $f(x) = x$. Then
$$\|f(x) - f(y)\|_2 = \|x - y\|_2 \le 2\tau_X.$$
It means $\Delta_2(f) \le 2\tau_X$. Now using Theorem 3.22 in [Dwork *et al.*, 2014] and Lemma H.1, we get the desired result.

**Lemma H.1** (Post-Processing [Dwork *et al.*, 2014])**.** *Let $\mathcal{M} : \mathbb{N}^{|\mathcal{X}|} \to R$ be a randomized algorithm that is $(\varepsilon, \delta)$-differentially private. Let $h : R \to R'$ be an arbitrary randomized mapping. Then $h \circ \mathcal{M} : \mathbb{N}^{|\mathcal{X}|} \to R'$ is $(\varepsilon, \delta)-$ differentially private.*

$\square$

# I    Proof for Theorem 3.8

*Proof.* Based on the assumptions $\|C\|_{2,\infty} \le \tau_C$, $\|\phi(Z)C - \phi(X)\|_{2,\infty} \le \gamma$, $\tilde{C} = C + E_C$ for the entry $(E_C)_{ij} \sim \mathcal{N}(0, \sigma_C^2)$, and $\tilde{Z} = Z + E_Z$ for the entry $(E_Z)_{ij} \sim \mathcal{N}(0, \sigma_Z^2)$, we obtain

$$\left\| \tilde{K}_{xx} - K_{xx} \right\|_\infty = \left\| \tilde{C}^T \mathcal{K}(\tilde{Z}, \tilde{Z})\tilde{C} - \mathcal{K}(X, X) \right\|_\infty$$
$$= \left\| (\phi(\tilde{Z})\tilde{C})^T (\phi(\tilde{Z})\tilde{C}) - \phi^T(X)\phi(X) \right\|_\infty$$
$$= \left\| (\phi(\tilde{Z})\tilde{C})^T (\phi(\tilde{Z})\tilde{C}) - (\phi(\tilde{Z})\tilde{C})^T \phi(X) + (\phi(\tilde{Z})\tilde{C})^T \phi(X) - \phi^T(X)\phi(X) \right\|_\infty$$
$$= \left\| (\phi(\tilde{Z})\tilde{C})^T (\phi(\tilde{Z})\tilde{C}) - \phi(X)) + (\phi(\tilde{Z})\tilde{C} - \phi(X))^T \phi(X) \right\|_\infty$$
$$\le \left\| \phi(\tilde{Z})\tilde{C} \right\|_{2,\infty} \left\| \phi(\tilde{Z})\tilde{C} - \phi(X) \right\|_{2,\infty} + \left\| \phi(\tilde{Z})\tilde{C} - \phi(X) \right\|_{2,\infty} \|\phi(X)\|_{2,\infty}$$
$$\le \left( \left\| \phi(\tilde{Z})\tilde{C} \right\|_{2,\infty} + \|\phi(X)\|_{2,\infty} \right) \underbrace{\left\| \phi(\tilde{Z})\tilde{C} - \phi(X) \right\|_{2,\infty}}_{\text{T.1}}$$

$$\tag{101}$$

where $\|\phi(\boldsymbol{X})\|_{2,\infty} = 1$. An upper bound on $\left\|\phi(\tilde{\boldsymbol{Z}})\tilde{\boldsymbol{C}}\right\|_{2,\infty}$ can be obtained only if we derive the upper bound on $\left\|\phi(\tilde{\boldsymbol{Z}})\tilde{\boldsymbol{C}} - \phi(\boldsymbol{X})\right\|_{2,\infty}$. That is, if $\left\|\phi(\tilde{\boldsymbol{Z}})\tilde{\boldsymbol{C}} - \phi(\boldsymbol{X})\right\|_{2,\infty} \leq \gamma_{zc}$, it implies that

$$\left\|\phi(\tilde{\boldsymbol{Z}})\tilde{\boldsymbol{C}}\right\|_{2,\infty} \leq \left\|\phi(\tilde{\boldsymbol{Z}})\tilde{\boldsymbol{C}} - \phi(\boldsymbol{X})\right\|_{2,\infty} + \|\phi(\boldsymbol{X})\|_{2,\infty} = \gamma_{zc} + 1 \tag{102}$$

which consequently gives

$$\left\|\tilde{\boldsymbol{K}}_{xx} - \boldsymbol{K}_{xx}\right\|_{\infty} \leq \gamma_{zc}(\gamma_{zc} + 2) \tag{103}$$

Hence, we derive the upper bound on $\left\|\phi(\tilde{\boldsymbol{Z}})\tilde{\boldsymbol{C}} - \phi(\boldsymbol{X})\right\|_{2,\infty}$ for the remaining proof.

$$\begin{aligned}
\left\|\phi(\tilde{\boldsymbol{Z}})\tilde{\boldsymbol{C}} - \phi(\boldsymbol{X})\right\|_{2,\infty} &= \left\|\phi(\tilde{\boldsymbol{Z}})\tilde{\boldsymbol{C}} - \phi(\boldsymbol{Z})\boldsymbol{C} + \phi(\boldsymbol{Z})\boldsymbol{C} - \phi(\boldsymbol{X})\right\|_{2,\infty} \\
&\leq \underbrace{\left\|\phi(\tilde{\boldsymbol{Z}})\tilde{\boldsymbol{C}} - \phi(\boldsymbol{Z})\boldsymbol{C}\right\|_{2,\infty}}_{T.2} + \|\phi(\boldsymbol{Z})\boldsymbol{C} - \phi(\boldsymbol{X})\|_{2,\infty}
\end{aligned} \tag{104}$$

where the second term $\|\phi(\boldsymbol{Z})\boldsymbol{C} - \phi(\boldsymbol{X})\|_{2,\infty} \leq \gamma$ is the assumption. Next, we derive the upper bound on $T.2$.

$$\begin{aligned}
\left\|\phi(\tilde{\boldsymbol{Z}})\tilde{\boldsymbol{C}} - \phi(\boldsymbol{Z})\boldsymbol{C}\right\|_{2,\infty} &= \left\|\phi(\tilde{\boldsymbol{Z}})\tilde{\boldsymbol{C}} - \phi(\tilde{\boldsymbol{Z}})\boldsymbol{C} + \phi(\tilde{\boldsymbol{Z}})\boldsymbol{C} - \phi(\boldsymbol{Z})\boldsymbol{C}\right\|_{2,\infty} \\
&= \left\|\phi(\tilde{\boldsymbol{Z}})(\tilde{\boldsymbol{C}} - \boldsymbol{C}) + (\phi(\tilde{\boldsymbol{Z}}) - \phi(\boldsymbol{Z}))\boldsymbol{C}\right\|_{2,\infty} \\
&\leq \left\|\phi(\tilde{\boldsymbol{Z}})(\tilde{\boldsymbol{C}} - \boldsymbol{C})\right\|_{2,\infty} + \left\|(\phi(\tilde{\boldsymbol{Z}}) - \phi(\boldsymbol{Z}))\boldsymbol{C}\right\|_{2,\infty} \\
&\leq \sqrt{d}\,\|\boldsymbol{E}_C\|_{2,\infty} + \left\|\phi(\tilde{\boldsymbol{Z}}) - \phi(\boldsymbol{Z})\right\|_F \|\boldsymbol{C}\|_{2,\infty} \\
&\leq \sigma_C \xi_d \sqrt{d} + \tau_C \underbrace{\left\|\phi(\tilde{\boldsymbol{Z}}) - \phi(\boldsymbol{Z})\right\|_F}_{T.3}
\end{aligned} \tag{105}$$

where we used $\frac{1}{\sigma_C^2}\|\boldsymbol{E}_C\|_{2,\infty}^2 \leq \xi_d^2 = d + 2\sqrt{dt} + 2t$ with probability at least $1 - ne^{-t}$ [Laurent and Massart, 2000] since it is the fact that $\frac{1}{\sigma_C^2}\|\boldsymbol{E}_C\|_{2,\infty}^2 \sim \chi_d^2$ where the entry $(\boldsymbol{E}_C)_{ij} \sim \mathcal{N}(0, \sigma_C^2)$. For $T.3$, we have

$$\begin{aligned}
\left\|\phi(\tilde{\boldsymbol{Z}}) - \phi(\boldsymbol{Z})\right\|_F^2 &= \operatorname{Tr}((\phi(\tilde{\boldsymbol{Z}}) - \phi(\boldsymbol{Z}))^T(\phi(\tilde{\boldsymbol{Z}}) - \phi(\boldsymbol{Z}))) \\
&= \operatorname{Tr}(\phi^T(\tilde{\boldsymbol{Z}})\phi(\tilde{\boldsymbol{Z}}) - \phi^T(\tilde{\boldsymbol{Z}})\phi(\boldsymbol{Z}) - \phi^T(\boldsymbol{Z})\phi(\tilde{\boldsymbol{Z}}) + \phi^T(\boldsymbol{Z})\phi(\boldsymbol{Z})) \\
&= 2d - 2\underbrace{\operatorname{Tr}(\phi^T(\tilde{\boldsymbol{Z}})\phi(\boldsymbol{Z}))}_{T.4}
\end{aligned} \tag{106}$$

For $T.4$, we can obtain

$$\begin{aligned}
\operatorname{Tr}(\phi^T(\tilde{\boldsymbol{Z}})\phi(\boldsymbol{Z})) &= \sum_{j=1}^{d} \phi^T(\tilde{\boldsymbol{z}}_j)\phi(\boldsymbol{z}_j) = \sum_{j=1}^{d} \exp\left(-\frac{\|\boldsymbol{z}_j + (\boldsymbol{E}_Z)_{:,j} - \boldsymbol{z}_j\|_2^2}{2r^2}\right) \\
&= \sum_{j=1}^{d} \exp\left(-\frac{\|(\boldsymbol{E}_Z)_{:,j}\|_2^2}{2r^2}\right) \\
&\geq d\exp\left(-\frac{\sigma_Z^2 \xi_d^2}{2r^2}\right)
\end{aligned} \tag{107}$$

where we use $\frac{1}{\sigma_Z^2}\|(\boldsymbol{E}_Z)_{:,j}\|_2^2 \leq \xi_d^2 = d + 2\sqrt{dt} + 2t$ with probability at least $1 - de^{-t}$ [Laurent and Massart, 2000] since it is the fact that $\frac{1}{\sigma_Z^2}\|(\boldsymbol{E}_Z)_{:,j}\|_2^2 \sim \chi_d^2$ where the entry $(\boldsymbol{E}_Z)_{ij} \sim \mathcal{N}(0, \sigma_Z^2)$.

Hence, we can go back to give an upper bound on $\left\|\phi(\tilde{\boldsymbol{Z}})\tilde{\boldsymbol{C}} - \phi(\boldsymbol{X})\right\|_{2,\infty}$:

$$\left\|\phi(\tilde{\boldsymbol{Z}})\tilde{\boldsymbol{C}} - \phi(\boldsymbol{X})\right\|_{2,\infty} \leq \gamma + \sigma_C \xi_d \sqrt{d} + \tau_C \sqrt{2d\left(1 - \exp\left(-\frac{\sigma_Z^2 \xi_d^2}{2r^2}\right)\right)} \tag{108}$$

Finally, we have

$$\left\|\tilde{\boldsymbol{K}}_{xx} - \boldsymbol{K}_{xx}\right\|_\infty \leq \gamma_{zc}(\gamma_{zc} + 2) \tag{109}$$

where

$$\gamma_{zc} = \gamma + \sqrt{d}\left(\sigma_C \xi_d + \tau_C \sqrt{2\left(1 - \exp\left(-\frac{\sigma_Z^2 \xi_d^2}{2r^2}\right)\right)}\right). \tag{110}$$

$\square$

## J  Proof for Theorem 3.9

**Lemma J.1.** *Assume $\Upsilon = \max_{\boldsymbol{z}_j, \boldsymbol{x}'} \|\boldsymbol{z}_j - \boldsymbol{x}'\|_2$ and $\|\boldsymbol{x} - \boldsymbol{x}'\|_2 \leq 2\tau_X$, let $\{\boldsymbol{C}_p^S\}_{p=1}^P$ be perturbed by noise drawn from $\mathcal{N}(0, \sigma^2)$ with the parameter $\sigma \geq \frac{2c\lambda^{-1}\sqrt{d}\tau_X(\tau_X + \Upsilon)}{r^2\varepsilon}$ for $c^2 > 2\ln(1.25/\delta)$. Then, the Gaussian Mechanism that adds noise to $\{\boldsymbol{C}_p^S\}_{p=1}^P$ is $(\varepsilon, \delta)$-differentially private.*

**Lemma J.2.** *Assume $\max_{(p,j)}\{\|\boldsymbol{x}_{p_j}\|_2, \|\boldsymbol{x}'_{p_j}\|_2\} \leq \tau_X$, $\max_{(i,j)}\|\boldsymbol{z}_i - \boldsymbol{x}_j\|_\infty = \Upsilon$, $\|\boldsymbol{Z}_p^s\|_{sp} \leq \tau_Z \ \forall s$, and $\|\boldsymbol{C}^S\|_{2,\infty} \leq \tau_C$, let $\{\boldsymbol{Z}_p^s\}_{p=1}^P$ for $s = 1, \cdots, S$ be perturbed by noise drawn from $\mathcal{N}(0, \sigma^2)$ with variance $(8S\Delta^2(g_Z)\log(e + (\varepsilon/\delta))/\varepsilon^2)$ where $\Delta(g_Z) = \frac{2\sqrt{d}\tau_C\tau_X\eta_k}{r^2}\left\{1 + (\tau_X + \tau_Z)\frac{(\tau_X + \Upsilon)}{r^2}\right\}$. Then, the Gaussian Mechanism that adds noise to $\{\boldsymbol{Z}_p^s\}_{p=1}^P$ for $s = 1, \cdots, S$ is $(\varepsilon, \delta)$-differentially private.*

By Lemma J.2, the mechanism that adds Gaussian noise to $\boldsymbol{Z}_p^s$ for $s = 1, \cdots, S$ with variance $(8S\Delta^2(g_Z)\log(e + (\varepsilon_Z/\delta_Z))/\varepsilon_Z^2)$ satisfies $(\varepsilon_Z, \delta_Z)$-differential privacy under $S$-fold adaptive composition for any $\varepsilon_Z > 0$ and $\delta_Z \in (0, 1]$. By Lemma J.1, the Gaussian Mechanism that injects noise to $\boldsymbol{C}_p^S$ with parameter $\sigma \geq 2c\lambda^{-1}\sqrt{d}\tau_X(\tau_X + \Upsilon)/(r^2\varepsilon_C)$ is $(\varepsilon_C, \delta_C)$-differentially private. Therefore, by Theorem 3.16 of [Dwork *et al.*, 2014], the proposed algorithm that adds Gaussian noise to $\boldsymbol{Z}_p^s$ for $s = 1, \cdots, S$ and $\boldsymbol{C}_p^S$ is $(\varepsilon_C + \varepsilon_Z, \delta_C + \delta_Z)$-differentially private. This finished the proof.

## K  Proof for Lemma J.1

*Proof.* In our FedSC, for each column of $\boldsymbol{C}$, we have

$$\boldsymbol{c} = g_C(\boldsymbol{x}) = \boldsymbol{G}\mathcal{K}(\boldsymbol{Z}, \boldsymbol{x}) = \boldsymbol{G}\begin{bmatrix} \exp\left(-\frac{\|\boldsymbol{z}_1 - \boldsymbol{x}\|_2^2}{2r^2}\right) \\ \vdots \\ \exp\left(-\frac{\|\boldsymbol{z}_d - \boldsymbol{x}\|_2^2}{2r^2}\right) \end{bmatrix}, \tag{111}$$

where $\boldsymbol{G} = (\mathcal{K}(\boldsymbol{Z}, \boldsymbol{Z}) + \lambda \boldsymbol{I}_d)^{-1}$. We have

$$\|g_C(\boldsymbol{x}) - g_C(\boldsymbol{x}')\|_2 \leq \|\boldsymbol{G}\|_{sp}\|\mathcal{K}(\boldsymbol{Z}, \boldsymbol{x}) - \mathcal{K}(\boldsymbol{Z}, \boldsymbol{x}')\|_2, \tag{112}$$

where $\|\cdot\|_{sp}$ denotes the spectral norm of matrix. Since $\exp(z)$ is locally Lipschitz continuous when $z < 0$, we have

$$
\begin{aligned}
&\left(\exp\left(-\frac{\|\boldsymbol{z}_j - \boldsymbol{x}\|_2^2}{2r^2}\right) - \exp\left(-\frac{\|\boldsymbol{z}_j - \boldsymbol{x}\|_2^2}{2r^2}\right)\right)^2 \\
&\leq \left|\frac{1}{2r^2}\left(\|\boldsymbol{z}_j - \boldsymbol{x}\|^2 - \|\boldsymbol{z}_j - \boldsymbol{x}'\|_2^2\right)\right|^2 \\
&= \left|\frac{1}{2r^2}\left(\|\boldsymbol{x} - \boldsymbol{x}'\|_2^2 + 2\langle \boldsymbol{z}_j - \boldsymbol{x}', \boldsymbol{x}' - \boldsymbol{x}\rangle\right)\right|^2 \\
&\leq \left|\frac{1}{2r^2}\left(\|\boldsymbol{x} - \boldsymbol{x}'\|_2^2 + 2\|\boldsymbol{z}_j - \boldsymbol{x}'\|_2\|\boldsymbol{x} - \boldsymbol{x}'\|_2\right)\right|^2.
\end{aligned}
\tag{113}
$$

Let $\Upsilon = \max_{\boldsymbol{z}_j, \boldsymbol{x}'} \|\boldsymbol{z}_j - \boldsymbol{x}'\|_2$ and $\|\boldsymbol{x} - \boldsymbol{x}'\|_2 \leq 2\tau_X$. Then the $\ell_2$-sensitivity of $g$ is

$$
\begin{aligned}
\triangle_2(g_C) &\leq \sup_{\boldsymbol{x}, \boldsymbol{x}'} \|\boldsymbol{G}\|_\sigma \sqrt{\sum_{j=1}^{d} \left| \frac{1}{2r^2} \left( \|\boldsymbol{x} - \boldsymbol{x}'\|_2^2 + 2\|\boldsymbol{z}_j - \boldsymbol{x}'\|_2 \|\boldsymbol{x}' - \boldsymbol{x}'\|_2 \right) \right|^2} \\
&\leq \|\boldsymbol{G}\|_\sigma \sqrt{\sum_{j=1}^{d} \left| \frac{1}{2r^2} \left( 4\tau_X^2 + 4\Upsilon\tau_X \right) \right|^2} \\
&= \|\boldsymbol{G}\|_\sigma \frac{2\sqrt{d}\tau_X(\tau_X + \Upsilon)}{r^2} \\
&\leq \frac{2\lambda^{-1}\sqrt{d}\tau_X(\tau_X + \Upsilon)}{r^2}.
\end{aligned}
\tag{114}
$$

Then according to Theorem 3.22 in [Dwork *et al.*, 2014], for $c^2 > 2\ln(1.25/\delta)$ the Gaussian Mechanism with parameter $\sigma \geq \frac{2c\lambda^{-1}\sqrt{d}\tau_X(\tau_X+\Upsilon)}{r^2\varepsilon}$ is $(\varepsilon, \delta)$-differentially private. This finished the proof. $\square$

## L  Proof for Lemma J.2

*Proof.* **Proof Sketch** The $(\epsilon, \delta)$-differential privacy of the proposed algorithm can be achieved by injecting noise into $\boldsymbol{Z}$ for each local update and into $\boldsymbol{C}$ at the final round. To prove this, we first compute the sensitivity of $\boldsymbol{Z}$ and $\boldsymbol{C}$ for determining the differential privacy of them. Then, we use the adaptive composition [Kairouz *et al.*, 2015] to get the superposition of them which will give the final theoretical result.

Now, the formal proof is as follows.

In our FedSC, consider one-step update of $\boldsymbol{Z}$ at client of $p$

$$
\boldsymbol{Z}_p^{s,t} = \boldsymbol{Z}_p^{s,t-1} - \eta_t \frac{\partial f}{\partial \boldsymbol{Z}}(\boldsymbol{Z}_p^{s,t-1})
\tag{115}
$$

where the derivative is given by

$$
\frac{\partial f}{\partial \boldsymbol{Z}}(\boldsymbol{Z}_p^{s,t-1}) = \frac{1}{r^2}(\boldsymbol{X}_p \boldsymbol{W}_Z - \boldsymbol{Z}\bar{\boldsymbol{W}}_Z) + \frac{2}{r^2}(\boldsymbol{Z}\boldsymbol{Q}_Z - \boldsymbol{Z}\bar{\boldsymbol{Q}}_Z)
\tag{116}
$$

and $\boldsymbol{X}_p = [\boldsymbol{x}_{p_1}, \cdots, \boldsymbol{x}_{p_j-1}, \boldsymbol{x}_{p_j}, \boldsymbol{x}_{p_j+1}, \cdots, \boldsymbol{x}_{p_{N_p}}]$.

For the simplicity of the proof, we omit the pair of iteration parameters $(s, t)$ and instead denote two adjacent local updates by $k$ and $k-1$ for nonnegative $k \geq 1$. Thus, we have the equivalent version of a one-step update of $\boldsymbol{Z}_p$ at client $p$

$$
\boldsymbol{Z}_p^k = \boldsymbol{Z}_p^{k-1} - \eta_k \left\{ \frac{1}{r^2}(\boldsymbol{X}_p \boldsymbol{W}_Z - \boldsymbol{Z}_p^{k-1}\bar{\boldsymbol{W}}_Z) + \frac{2}{r^2}(\boldsymbol{Z}_p^{k-1}\boldsymbol{Q}_Z - \boldsymbol{Z}_p^{k-1}\bar{\boldsymbol{Q}}_Z) \right\}
\tag{117}
$$

where $\boldsymbol{X}_p = [\boldsymbol{x}_{p_1}, \cdots, \boldsymbol{x}_{p_j-1}, \boldsymbol{x}_{p_j}, \boldsymbol{x}_{p_j+1}, \cdots, \boldsymbol{x}_{p_{N_p}}]$.

To compute the sensitivity of $\boldsymbol{Z}_p$, we give the counterpart of the above update as

$$
(\boldsymbol{Z}_p^k)' = \boldsymbol{Z}_p^{k-1} - \eta_k \left\{ \frac{1}{r^2}(\boldsymbol{X}_p' \boldsymbol{W}_Z' - \boldsymbol{Z}_p^{k-1}\bar{\boldsymbol{W}}_Z') + \frac{2}{r^2}(\boldsymbol{Z}_p^{k-1}\boldsymbol{Q}_Z - \boldsymbol{Z}_p^{k-1}\bar{\boldsymbol{Q}}_Z) \right\}
\tag{118}
$$

where $\boldsymbol{X}_p' = [\boldsymbol{x}_{p_1}, \cdots, \boldsymbol{x}_{p_j-1}, \boldsymbol{x}_{p_j}', \boldsymbol{x}_{p_j+1}, \cdots, \boldsymbol{x}_{p_{N_p}}]$.

Next, let's start to derive the upper bound on $\|\boldsymbol{Z}_p^k - (\boldsymbol{Z}_p^k)'\|_F$ term by term.

$$
\begin{aligned}
\left\| \boldsymbol{Z}_p^k - (\boldsymbol{Z}_p^k)' \right\|_F &= \left\| -\frac{\eta_k}{r^2} \left\{ (\boldsymbol{X}_p \boldsymbol{W}_Z - \boldsymbol{Z}_p^{k-1}\bar{\boldsymbol{W}}_Z) - (\boldsymbol{X}_p' \boldsymbol{W}_Z' - \boldsymbol{Z}_p^{k-1}\bar{\boldsymbol{W}}_Z') \right\} \right\|_F \\
&= \frac{\eta_k}{r^2} \left\| (\boldsymbol{X}_p \boldsymbol{W}_Z - \boldsymbol{X}_p' \boldsymbol{W}_Z') - \boldsymbol{Z}_p^{k-1} \left( \bar{\boldsymbol{W}}_Z - \bar{\boldsymbol{W}}_Z' \right) \right\|_F \\
&\leq \frac{\eta_k}{r^2} \left\{ \underbrace{\left\| \boldsymbol{X}_p \boldsymbol{W}_Z - \boldsymbol{X}_p' \boldsymbol{W}_Z' \right\|_F}_{\text{T.1}} + \underbrace{\left\| \boldsymbol{Z}_p^{k-1} \left( \bar{\boldsymbol{W}}_Z - \bar{\boldsymbol{W}}_Z' \right) \right\|_F}_{\text{T.2}} \right\}
\end{aligned}
\tag{119}
$$

For $T.1$, we have

$$
\begin{aligned}
\left\|\boldsymbol{X}_p \boldsymbol{W}_Z - \boldsymbol{X}'_p \boldsymbol{W}'_Z\right\|_F &= \left\|\boldsymbol{X}_p \boldsymbol{W}_Z - \boldsymbol{X}'_p \boldsymbol{W}_Z + \boldsymbol{X}'_p \boldsymbol{W}_Z - \boldsymbol{X}'_p \boldsymbol{W}'_Z\right\|_F \\
&\leq \underbrace{\left\|\left(\boldsymbol{X}_p - \boldsymbol{X}'_p\right) \boldsymbol{W}_Z\right\|_F}_{T.3} + \underbrace{\left\|\boldsymbol{X}'_p\left(\boldsymbol{W}_Z - \boldsymbol{W}'_Z\right)\right\|_F}_{T.4}
\end{aligned} \tag{120}
$$

For $T.3$, we have

$$
\begin{aligned}
\left\|\left(\boldsymbol{X}_p - \boldsymbol{X}'_p\right) \boldsymbol{W}_Z\right\|_F &= \left\|(\boldsymbol{x}_{p_j} - \boldsymbol{x}'_{p_j})(-\boldsymbol{c}_j^T \odot \mathcal{K}(\boldsymbol{x}_{p_j}, \boldsymbol{Z}_p^{k-1}))\right\|_F \\
&\leq \left\|\boldsymbol{x}_{p_j} - \boldsymbol{x}'_{p_j}\right\|_2 \left\|\boldsymbol{c}_j^T \odot \mathcal{K}(\boldsymbol{x}_{p_j}, \boldsymbol{Z}_p^{k-1})\right\|_2 \\
&= \left\|\boldsymbol{x}_{p_j} - \boldsymbol{x}'_{p_j}\right\|_2 \sqrt{(\boldsymbol{c}_j \odot \boldsymbol{c}_j)^T(\mathcal{K}(\boldsymbol{Z}_p^{k-1}, \boldsymbol{x}_{p_j}) \odot \mathcal{K}(\boldsymbol{Z}_p^{k-1}, \boldsymbol{x}_{p_j}))} \\
&\leq \left\|\boldsymbol{x}_{p_j} - \boldsymbol{x}'_{p_j}\right\|_2 \sqrt{\left\|\boldsymbol{c}_j \odot \boldsymbol{c}_j\right\|_2 \left\|\mathcal{K}(\boldsymbol{Z}_p^{k-1}, \boldsymbol{x}_{p_j}) \odot \mathcal{K}(\boldsymbol{Z}_p^{k-1}, \boldsymbol{x}_{p_j})\right\|_2} \\
&= \left\|\boldsymbol{x}_{p_j} - \boldsymbol{x}'_{p_j}\right\|_2 \left\|\boldsymbol{c}_{p_j}\right\|_4 \left\|\mathcal{K}(\boldsymbol{Z}_p^{k-1}, \boldsymbol{x}_{p_j})\right\|_4 \\
&\leq \left\|\boldsymbol{x}_{p_j} - \boldsymbol{x}'_{p_j}\right\|_2 \left\|\boldsymbol{c}_{p_j}\right\|_2 \left\|\mathcal{K}(\boldsymbol{Z}_p^{k-1}, \boldsymbol{x}_{p_j})\right\|_2 \\
&\leq \left\|\boldsymbol{x}_{p_j} - \boldsymbol{x}'_{p_j}\right\|_2 \left\|\boldsymbol{C}\right\|_{2,\infty} \left\|\mathcal{K}(\boldsymbol{Z}_p^{k-1}, \boldsymbol{x}_{p_j})\right\|_2
\end{aligned} \tag{121}
$$

Here, we use $\left\|\boldsymbol{a} \odot \boldsymbol{b}\right\|_2 = \sqrt{(\boldsymbol{a} \odot \boldsymbol{a})^T (\boldsymbol{b} \odot \boldsymbol{b})}$ for $\boldsymbol{a}, \boldsymbol{b} \in \mathbb{R}^d$ for the second equality; Cauchy-Schwarz inequality for the second inequality; $\sqrt{\left\|\boldsymbol{a} \odot \boldsymbol{a}\right\|_2} = \left\|\boldsymbol{a}\right\|_4$ for $\boldsymbol{a} \in \mathbb{R}^d$ for the third inequality. Let $\Delta_{\boldsymbol{Z},x} = \mathcal{K}(\boldsymbol{Z}, \boldsymbol{x}) - \mathcal{K}(\boldsymbol{Z}, \boldsymbol{x}')$, then we have for $T.4$,

$$
\begin{aligned}
\left\|\boldsymbol{X}'_p\left(\boldsymbol{W}_Z - \boldsymbol{W}'_Z\right)\right\|_F &= \left\|\boldsymbol{x}'_{p_j}\left\{-\boldsymbol{c}_j^T \odot \left(\mathcal{K}(\boldsymbol{x}_{p_j}, \boldsymbol{Z}_p^{k-1}) - \mathcal{K}(\boldsymbol{x}'_{p_j}, \boldsymbol{Z}_p^{k-1})\right)\right\}\right\|_F \\
&\leq \left\|\boldsymbol{x}'_{p_j}\right\|_2 \left\|\boldsymbol{c}_j^T \odot \left(\mathcal{K}(\boldsymbol{x}_{p_j}, \boldsymbol{Z}_p^{k-1}) - \mathcal{K}(\boldsymbol{x}'_{p_j}, \boldsymbol{Z}_p^{k-1})\right)\right\|_2 \\
&= \left\|\boldsymbol{x}'_{p_j}\right\|_2 \sqrt{(\boldsymbol{c}_j \odot \boldsymbol{c}_j)^T \left(\triangle_{\boldsymbol{Z}_p^{k-1}, \boldsymbol{x}_{p_j}} \odot \triangle_{\boldsymbol{Z}_p^{k-1}, \boldsymbol{x}_{p_j}}\right)} \\
&\leq \left\|\boldsymbol{x}'_{p_j}\right\|_2 \left\|\boldsymbol{C}\right\|_{2,\infty} \left\|\triangle_{\boldsymbol{Z}_p^{k-1}, \boldsymbol{x}_{p_j}}\right\|_2
\end{aligned} \tag{122}
$$

Therefore, assume $\max\{\left\|\boldsymbol{x}_{p_j}\right\|_2, \left\|\boldsymbol{x}'_{p_j}\right\|_2\} \leq \tau_X$, which means $\left\|\boldsymbol{x} - \boldsymbol{x}'\right\|_2 \leq 2\tau_X$, we can get an upper bound on $T.1$.

$$
\begin{aligned}
\left\|\boldsymbol{X}_p \boldsymbol{W}_Z - \boldsymbol{X}'_p \boldsymbol{W}'_Z\right\|_F &\leq \underbrace{\left\|\left(\boldsymbol{X}_p - \boldsymbol{X}'_p\right) \boldsymbol{W}_Z\right\|_F}_{T.5} + \underbrace{\left\|\boldsymbol{X}'_p\left(\boldsymbol{W}_Z - \boldsymbol{W}'_Z\right)\right\|_F}_{T.6} \\
&\leq \left\|\boldsymbol{x}_{p_j} - \boldsymbol{x}'_{p_j}\right\|_2 \left\|\boldsymbol{C}\right\|_{2,\infty} \left\|\mathcal{K}(\boldsymbol{Z}_p^{k-1}, \boldsymbol{x}_{p_j})\right\|_2 + \left\|\boldsymbol{x}'_{p_j}\right\|_2 \left\|\boldsymbol{C}\right\|_{2,\infty} \left\|\triangle_{\boldsymbol{Z}_p^{k-1}, \boldsymbol{x}_{p_j}}\right\|_2 \\
&= \left\|\boldsymbol{C}\right\|_{2,\infty} \left(\left\|\boldsymbol{x}_{p_j} - \boldsymbol{x}'_{p_j}\right\|_2 \left\|\mathcal{K}(\boldsymbol{Z}_p^{k-1}, \boldsymbol{x}_{p_j})\right\|_2 + \left\|\boldsymbol{x}'_{p_j}\right\|_2 \left\|\triangle_{\boldsymbol{Z}_p^{k-1}, \boldsymbol{x}_{p_j}}\right\|_2\right) \\
&\leq 2\sqrt{d}\tau_C \tau_X \left(1 + \frac{\tau_X(\tau_X + \Upsilon)}{r^2}\right)
\end{aligned} \tag{123}
$$

Here, we also used the fact that $\left\|\mathcal{K}(\boldsymbol{Z}_p^{k-1}, \boldsymbol{x}_{p_j})\right\|_2 \leq \sqrt{d}$ and the derived bound $\frac{2\sqrt{d}\tau_X(\tau_X + \Upsilon)}{r^2}$ on $\left\|\triangle_{\boldsymbol{Z}_p^{k-1}, \boldsymbol{x}_{p_j}}\right\|_2$ given by K, and $\left\|\boldsymbol{C}\right\|_{2,\infty} \leq \tau_C$ given in Proof F.

Assume $\|\boldsymbol{Z}_p^k\|_{sp} \leq \tau_Z\ \forall k$, we have for $T.2$

$$
\begin{aligned}
\left\|\boldsymbol{Z}_p^{k-1}\left(\bar{\boldsymbol{W}}_Z - \bar{\boldsymbol{W}}_Z'\right)\right\|_F &= \left\|\boldsymbol{Z}_p^{k-1}\left(\mathrm{diag}(\mathbf{1}_n^T \boldsymbol{W}_Z) - \mathrm{diag}(\mathbf{1}_n^T \boldsymbol{W}_Z')\right)\right\|_F \\
&= \left\|\boldsymbol{Z}_p^{k-1}\mathrm{diag}(\mathbf{1}_n^T(\boldsymbol{W}_Z - \boldsymbol{W}_Z'))\right\|_F \\
&\leq \left\|\boldsymbol{Z}_p^{k-1}\right\|_{sp}\left\|\mathrm{diag}(\mathbf{1}_n^T(\boldsymbol{W}_Z - \boldsymbol{W}_Z'))\right\|_F \\
&= \left\|\boldsymbol{Z}_p^{k-1}\right\|_{sp}\left\|\mathbf{1}_n^T(\boldsymbol{W}_Z - \boldsymbol{W}_Z')\right\|_2 \\
&\leq \left\|\boldsymbol{Z}_p^{k-1}\right\|_{sp}\left\|\boldsymbol{c}_j^T \odot \left(\mathcal{K}(\boldsymbol{x}_{p_j}, \boldsymbol{Z}_p^{k-1}) - \mathcal{K}(\boldsymbol{x}_{p_j}', \boldsymbol{Z}_p^{k-1})\right)\right\|_2 \\
&\leq \left\|\boldsymbol{Z}_p^{k-1}\right\|_{sp}\left\|\boldsymbol{C}\right\|_{2,\infty}\left\|\triangle_{\boldsymbol{Z}_p^{k-1}, \boldsymbol{x}_{p_j}}\right\|_2 \\
&\leq \frac{2\sqrt{d}\tau_Z \tau_C \tau_X(\tau_X + \Upsilon)}{r^2}
\end{aligned}
\tag{124}
$$

Thus, we get the upper bounds on $T.1$ and $T.2$, respectively, and finally give an upper bound on $\left\|\boldsymbol{Z}_p^k - (\boldsymbol{Z}_p^k)'\right\|_F$.

$$
\begin{aligned}
\left\|\boldsymbol{Z}_p^k - (\boldsymbol{Z}_p^k)'\right\|_F &\leq \frac{\eta_k}{r^2}\left\{\underbrace{\left\|\boldsymbol{X}_p \boldsymbol{W}_Z - \boldsymbol{X}_p' \boldsymbol{W}_Z'\right\|_F}_{T.1} + \underbrace{\left\|\boldsymbol{Z}_p^{k-1}\left(\bar{\boldsymbol{W}}_Z - \bar{\boldsymbol{W}}_Z'\right)\right\|_F}_{T.2}\right\} \\
&\leq \frac{\eta_k}{r^2}\left\{2\sqrt{d}\tau_C \tau_X\left(1 + \frac{\tau_X(\tau_X + \Upsilon)}{r^2}\right) + \frac{2\sqrt{d}\tau_Z \tau_C \tau_X(\tau_X + \Upsilon)}{r^2}\right\} \\
&= \frac{2\sqrt{d}\tau_C \tau_X \eta_k}{r^2}\left\{1 + (\tau_X + \tau_Z)\frac{(\tau_X + \Upsilon)}{r^2}\right\}
\end{aligned}
\tag{125}
$$

Therefore, if we define $\boldsymbol{Z}_p = g_Z(\boldsymbol{X}_p)$, the $\ell_2$-sensitivity of $g_Z$ is

$$
\begin{aligned}
\triangle_2(g_Z) &= \sup_{\boldsymbol{X}_p, \boldsymbol{X}_p'}\|g_Z(\boldsymbol{X}_p) - g_Z(\boldsymbol{X}_p')\|_2 \\
&= \sup_{\boldsymbol{X}_p, \boldsymbol{X}_p'}\left\|\boldsymbol{Z}_p^k - (\boldsymbol{Z}_p^k)'\right\|_F \\
&\leq \frac{2\sqrt{d}\tau_C \tau_X \eta_k}{r^2}\left\{1 + (\tau_X + \tau_Z)\frac{(\tau_X + \Upsilon)}{r^2}\right\}
\end{aligned}
\tag{126}
$$

By Theorem 4.3 of [Kairouz *et al.*, 2015], the mechanism that adds Gaussian noise to $\boldsymbol{Z}_p^s$ for $s = 1, \cdots, S$ with variance $(8S\Delta^2(g_Z)\log(e + (\varepsilon_Z/\delta_Z))/\varepsilon_Z^2)$ satisfies $(\varepsilon_Z, \delta_Z)$-differential privacy under $S$-fold adaptive composition for any $\varepsilon_Z > 0$ and $\delta_Z \in (0, 1]$. This finished the proof.

$\square$

