# OpenReview forum: "Federated Spectral Clustering via Secure Similarity Reconstruction"
_NeurIPS.cc/2023/Conference — NeurIPS 2023 poster_

### Official Review · Reviewer_Mxhk · 2023-07-01

**Soundness:** 3 good
**Presentation:** 3 good
**Contribution:** 3 good
**Rating:** 7
**Confidence:** 5

**Summary:**

The paper proposes a secure kernelized factorization method for federated spectral clustering on distributed data. It provides convergence guarantee of the optimization, reconstruction error bound of similarity matrix, and sufficient condition of correct clustering. It also presents guarantees of differential privacy. The effectiveness of the proposed method is demonstrated by synthetic and real datasets.

**Strengths:**

1. The paper studies federated spectral clustering, which is an important problem but was rarely studied in literature.

2. The authors propose a federated spectral clustering algorithm and analyzed the convergence of the optimization algorithm.

3. Besides the federated factorization model for similarity matrix reconstruction, the authors propose two approaches to enhancing the privacy of federated spectral clustering, one is to perturb the original data, the other is to perturb the learned factors.

4. The authors provide a lot of theoretical guarantees for the proposed methods, including the reconstruction error (Theorem 3.1) of the true similarity matrix, condition of correct clustering (Theorem 3.6), and differential privacy analysis (Proposition 3.7 and Theorem 3.8).

5. The numerical results show that the proposed FedSC is more effective than the baselines such as DSC.

In sum, this is a strong paper that addressed an important problem. It contains both theoretical analysis and numerical evaluation. The paper is well-written and easy to follow.

**Weaknesses:**

1. Some theoretical result and all the experimental results of FedSC with perturbed factors are in the supplementary material. I suggest the authors put them back to the main paper via adjusting the structure of the paper, e.g., simplifying Table 1 in the main paper.

2. A few points haven’t been sufficiently explained. Please refer to the session of Questions.

**Questions:**

1. In equation (11), why not use a fixed number of participated clients?

2. In Theorem 3.1, was the $\ell_{2,\infty}$ norm defined as the maximum of the $\ell_2$ norms of the matrix columns? If yes, the definition of $\theta$ and $\tau$ can also use the $\ell_{2,\infty}$ norm.

3. Does formula (27) ensure the symmetry?

4. Section 3.1 considered perturbed data while section 3.2 considered perturbed factors. Which strategy is better?

5. In Theorem 3.8, the definition of $\Upsilon$ is related to $\mathbf{z}_i$, which does not include an $s$ or $S$. Is it a typo? What is the connection between the privacy protection and the reconstruction of similarity matrix or the performance of spectral clustering.

6. In Figure 3(e), it is not clear why vanilla SC failed but FedSC still worked well.

7. In Table 1, it seems that the performance of the baseline DSC is not satisfactory in most cases. Can the authors explain more about the result?

---

> ### Author Rebuttal · Authors · 2023-08-09
>
> The authors sincerely appreciate the insightful comments made by the reviewer.
>
> **Response to weakness 1:**
>
> This issue gets involved with the unexpected drop-out of participated clients in each round, which is actually a common phenomenon in network communication. But here, we just want to claim our model adopted the federated framework of model averaging. To concentrate on our topic, we have used a fixed number of participating clients (as claimed in lines 109 - 110.) and ignored the abnormal phenomenon.
>
> **Response to weakness 2:**
>
> Thanks for pointing out. We will correct it.
>
> **Response to weakness 3:**
>
> In general, the affinity matrix for spectral clustering is not necessarily symmetric. However, in (27), we have made $\hat{\boldsymbol{K}}_{\tilde{x} \tilde{x}}$ symmetric, which is to ensure the consistency with the kernel matrix to be reconstructed.
>
> **Response to weakness 4:**
>
> Through our experiments, it can be observed that perturbing factors can have a more significant impact on the accuracy of clustering results than perturbing raw data. That is, perturbing factors are more sensitive and can achieve a specified level of DP with weaker noise. In terms of this, the latter may be better.
>
> **Response to weakness 5:**
>
> Sorry about this typo. Since we only perturbed factor $\\{\mathbf{C}_p^S\\} _{p = 1}^P$, it should be $\mathbf{z}_i^S$. We will correct this. We aim at designing a secure spectral clustering for protecting users' raw data but still need to guarantee that the proposed algorithm has satisfactory accuracy. Specifically, privacy protection gives a lower bound on the variance of noise while reconstruction with good quality requires that the noise cannot break down some upper bound. The intensity of noise should meet the two requirements to help see the privacy-utility tradeoff.
>
> **Response to weakness 6:**
>
> As we know, the similarity graph directly constructed from raw data could be very sensitive to each data point. When we added too much noise to the raw data, the similarity graph may fail to model the local neighborhood relationships which may be the reason why 3(e) is not separable for vanilla SC. Instead, FedSC is just to realize an approximation $\mathbf{C}^T\mathcal{K}(\mathbf{Z}, \mathbf{Z})\mathbf{C}$ of $\mathcal{K}(\mathbf{X},\mathbf{X})$ which could have potential denoising effect on the raw data. Therefore, it is possible for FedSC to achieve a better performance.
>
> **Response to weakness 7:**
>
> As proposed by DSC, it performs spectral clustering on each local dataset, which may lead to very unstable results. Besides, the extra operation includes the second k-means on the local centroids to get the global centroids. This makes it less interpretable compared with the vanilla SC. These may be reasons why DSC did not perform well in our experiments.

---

> > ### Comment · Reviewer_Mxhk · 2023-08-19
> >
> > Thanks for your response to my comments. I have no concerns about the work. By the way, after reading the questions of the other reviewers and the corresponding rebuttals given by authors, I am more confident about my assessment. Thus, I raise my confidence level to 5 and keep my rating as 7. The work is worth of being published in NeurIPS.

---

> > > ### Author Response · Authors · 2023-08-21
> > > **Authors' feedback**
> > >
> > > Dear Reviewer,
> > >
> > > We sincerely appreciate your very positive assessment on our work.
> > >
> > > Best,
> > >
> > > Authors

---

### Official Review · Reviewer_fSDW · 2023-07-05

**Soundness:** 3 good
**Presentation:** 4 excellent
**Contribution:** 3 good
**Rating:** 7
**Confidence:** 4

**Summary:**

This paper solves unsupervised spectral clustering by reconstructing the similarity matrix under a Federated Learning (FL) and Differentially Private (DP) setting. The authors argue that since the similarity matrix requires computation across pairs of clients it might not be directly FL-compatible. Thus, the authors set out to approximate the feature-mapped matrix ($\phi(X_p)$) for some client $p$ by an appropriate dictionary-coefficient product. The eventual similarity matrix is computed by the server by collecting each client's approximate dictionary-coefficient pair. In particular, since the raw data does not directly reach the server, the proposed algorithm FedSC is FL-compatible.

 The rest of the article demonstrates the convergence analysis of the proposed method, DP analysis and results, and empirical results. Especially, the FL results seem to converge well with large enough iterations for moderately sized datasets. However, the reviewer has some confusion regarding the DP results and their efficacy for high-dimensional matrices, the comparison against other traditional FL / distributed approaches, and the provided empirical results.

**Strengths:**

The paper discusses a solution for the often under-studied field of Federated unsupervised learning. In particular, the authors propose a solution for spectral clustering that is FL-compatible while ensuring that the sum over the differences in the consecutive sequence terms generated by spectral clustering converges to a finite value over asymptotic $T$.

1. Primarily, the authors manage to tackle the data pair problem faced in computing the similarity matrix. For simply computing the Gaussian kernel, a cross-term product between terms of different clients is required. Thus vanilla FL training no longer suffices without an assumption that clients are allowed to directly interact with one another. Thus, the general framework provided by the authors can be quite important for not only FL-based clustering but to reduce the communication cost of $O(N^2)$ required by non-private training where each client is allowed to interact with other clients. Here, $N$ is the total number of clients.
2. The authors demonstrate the FL compatibility of FedSC via the proposed two-pronged optimization technique that generates a “close” similarity matrix to the original one while also ensuring that the errors generated due to FL are asymptotically small for a large number of iterations.
3. Empirically, compared to previous FL spectral clustering approaches, the proposed FedSC performs the best for most scenarios. Furthermore, FedSC is Differential Privacy (DP) compatible too.
4. The paper is clearly written and explores multiple avenues such as FL, DP-FL, and adding privacy noise to both the original data matrix as well as the outputs of the proposed algorithm.

**Weaknesses:**

Although the authors provide a unique approach to handling FL-based spectral clustering, some questions about their approach are detailed below.

1. It is unclear why the assumption of $\phi(X) = C\phi(Z)$ is the best choice for the FL-only case. The aim of FL is to ensure that raw data is never transmitted to the server and the artifact transmitted to the server should not be easily invertible back to the raw data. Thus, instead of the proposed solution if we now approximate $\phi(X) = f(Z) - C$ where $C$ is kept private to the client. Then, it might be interesting to understand whether such a transformation would suffice for FL and what benefits might the approach in the paper provide over this other approach (or another similar approach).
2. The paper provides DP-FL guarantees for FedSC. However, the accuracy guarantee is weak for the high number of dimensions $m$ or high value of $d$ (dictionary and coefficient size). Particularly, from Thm. 3.1, the $l_{\infty}$ norm is of the order $m$.
3. Although this paper does not tackle a distributed setting, it will be beneficial to provide some insight into any possibilities of using Secure Aggregation or other cryptographic techniques to handle pair-wise client functions such as the kernel function. Although the communication cost will be high, there might be ways to reduce the DP noise levels.
4. For the empirical evaluations, it is surprising that Vanilla spectral clustering performs worse than FedSC. Providing some intuition for this result will be helpful.
5. Consider including a discussion section that demonstrates the benefits of the proposed approach over distributed learning (in terms of communication costs).

**Questions:**

The paper is well-written and presented. In particular, the authors demonstrate a general framework that can compute pair-wise client functions without relying on a distributed learning assumption. There are no major concerns for the FL-only work. For the DP-FL work point 2 in the weaknesses section is important to address as it might render the DP-FL solution impractical for high dimensions. Point 3 in the same section might provide an alternative solution that could be explored further.

**Limitations:**

The authors generally address most limitations well including the impact of large privacy noise and the impact of large-scale datasets (inherent limitation of spectral clustering).

---

> ### Author Rebuttal · Authors · 2023-08-09
>
> First, the authors sincerely thank you for recognizing our work. We provide the responses to your five questions as follows.
>
> **Response to weakness 1:**
>
> Thanks for the insightful comment. Our goal is to compute a kernel matrix $\mathcal{K}(\mathbf{X}, \mathbf{X})$ without sending the data to the server. To achieve the goal, we consider $\phi(\mathbf{X}) = \phi(\mathbf{Z})\mathbf{C}$ as it can be used to estimate the kernel matrix, i.e., $\mathbf{C}^T\mathcal{K}(\mathbf{Z}, \mathbf{Z})\mathbf{C}\approx \mathcal{K}(\mathbf{X}, \mathbf{X})$, where we send $\mathbf{Z},\mathbf{C}$ instead of $\mathbf{X}$ to the server. If we use $\phi(\mathbf{X}) = f(\mathbf{Z}) - \mathbf{C}$, we may not obtain an estimation for $\mathcal{K}(\mathbf{X}, \mathbf{X})$. In other words, it could be difficult to find $f$ an $\mathbf{C}$ to ensure $(f(\mathbf{Z}) - \mathbf{C})^\top (f(\mathbf{Z}) - \mathbf{C})=f(\mathbf{Z})^\top  f(\mathbf{Z})+\mathbf{C}^\top\mathbf{C}-f(\mathbf{Z})^\top\mathbf{C}-\mathbf{C}^\top f(\mathbf{Z})$ is a good reconstruction of the kernel matrix, unless we replace $\mathbf{C}$ with $\phi(\mathbf{C})$ and $f(\mathbf{Z})$ with $\phi(\mathbf{Z})$. We will provide more explanation in the revised paper.
>
> **Response to weakness 2:**
>
> The accuracy guarantee is actually not that bad since Theorem 3.1 implies that the reconstruction error increases linearly with $\sqrt{d}$ and linear with ${m}$ but there is an $r^2$ in the denominator: $\left\Vert\hat{\boldsymbol{K}} _{\tilde{x} \tilde{x}}-\boldsymbol{K} _{x x}\right\Vert _{\infty} \leq \frac{1}{r^2}\left[(\sigma \xi+\sqrt{2} \theta)^2-2 \theta^2\right]+\left(\sqrt{d} \tau_C+1\right) \gamma$ where $\xi=\sqrt{(m+2 \sqrt{m t}+2 t)}$. Note that $r$ is the hyperparameter of the Gaussian kernel. In our experiment, $r$ was automatically estimated as the mean of all pairwise distance between data points, _i.e._, $r = \frac{1}{n^2}\sum _{i,j}d(\mathbf{x}_i, \mathbf{x}_j)$. Note that $d(\mathbf{x}_i, \mathbf{x}_j)=\Vert \mathbf{x} _i - \mathbf{x} _j \Vert$ is linear with $\sqrt{m}$. More intuitively, if we assume $|\mathbf{x} _{ik} - \mathbf{x} _{jk}| = \mathcal{O}(\varepsilon)$ for all $i, j \in [n], k\in[m]$, then $d(\mathbf{x} _i, \mathbf{x} _j) =  \mathcal{O}(\sqrt{m}\varepsilon)$, which implies that $r=\mathcal{O}(\sqrt{m}\varepsilon)$. To sum up, $r^2$ is linear with $m$, which indicates that the $\ell _{\infty}$ norm of the reconstruction error is only linear with $\sqrt{d}$, not related to $m$. We will add this explanation to the revised paper. Thank you very much for raising the question that has made our theoretical result more meaningful and clearer.
>
> **Response to weakness 3:**
>
> Thanks a lot for the nice suggestion. In our proposed framwork, both $\mathbf{Z} _p^S$ and $\mathbf{C} _p^S$ need to be posted to central server for estimating a similarity graph $\mathbf{C}^T\mathcal{K}(\mathbf{Z}, \mathbf{Z})\mathbf{C}$, which may give more information to the central server even though we have some theoretical guarantees on the security of FedSC. If we we want to further enhance the security of the proposed framework using cryptographic methods, here is an alternative that can be shown.
> 1. Step 1: $\mathbf{Z} _p^S$ is posted to the central server;
> 2. Step 2: Central server aggregates $\mathbf{Z} _p^S$ to get the global $\mathbf{Z}^S$;
> 3. Step 3: Central server computes $\mathcal{K} _{ZZ} = \mathcal{K}(\mathbf{Z}^S, \mathbf{Z}^S)$ and broadcast it to clients;
> 4. Step 4: For client $p$ and $\mathbf{c} _i \in \mathbf{C} _p$, if $\mathbf{c} _j \in \mathbf{C} _p$, then client $p$ directly calculate $\hat{\mathcal{K}} _{ij} = \mathbf{c} _i\mathcal{K} _{ZZ}\mathbf{c} _j$; if $\mathbf{c} _j \in \mathbf{C} _{p'}$, then client $p$ firstly encrypts and transfers its $\mathbf{c} _i$ to client $p'$, and then client $p'$ also encrypts its $\mathbf{c} _j$ and use the cipher text to compute $enc(\hat{\mathcal{K}} _{ij}) = enc(\mathbf{c} _i^T)enc(\mathcal{K} _{ZZ})enc(\mathbf{c} _j)$; Client $p'$ transfers the result $enc(\hat{\mathcal{K}} _{ij})$ back to client $p$; Client $p$ decrypts $enc(\hat{\mathcal{K}} _{ij})$ to get $\hat{\mathcal{K}} _{ij}$.
> 5. Step 5: Each client $p$ sends its estimated results $\hat{\mathcal{K}} _{ij}$ back to the central server without sending its own $\mathbf{C} _p$;
> 6. Step 6: Central Server checks whether these posted results from clients are compatible with each other in case of injection attacks and then performs spectral clustering.
>
> This alternative does not send clients' $\mathbf{C}$ to the central server and gives an extra cross-validation process in the central server which may be useful to enhance the security. We will discuss the possibilities of using Secure Aggregation or other cryptographic techniques to handle pair-wise client functions such as the kernel function.
>
> **Response to weakness 4:**
>
> As we know, the similarity graph is a key ingredient of spectral clustering. As for vanilla spectral clustering, the similarity graph directly constructed from raw data could be very sensitive to each data point. Instead, FedSC is just to realize an approximation $\mathbf{C}^T\mathcal{K}(\mathbf{Z}, \mathbf{Z})\mathbf{C}$ of $\mathcal{K}(\mathbf{X},\mathbf{X})$ which may have potential denoising effect on the raw data. Therefore, it is possible for FedSC to achieve a better performance.
>
> **Response to weakness 5:**
>
> Yes. It is important and interesting to show the benefits of our method over distributed learning since federated learning is closely related to distributed learning.

---

> > ### Comment · Reviewer_fSDW · 2023-08-21
> >
> > Thank you for providing a detailed response to my queries. I believe almost all of my concerns have been alleviated (especially weakness 2). I have no further questions and although I maintain my current paper rating as 7, I have updated its confidence to a 4.

---

> > > ### Author Response · Authors · 2023-08-21
> > > **Authors' feedback**
> > >
> > > Dear Reviewer,
> > >
> > > Thank you so much for recognizing our contribution and updating the confidence score. We will add the above analysis as well as more details to the paper. Your comments have made our paper stronger.
> > >
> > > Sincerely,
> > >
> > > Authors

---

### Official Review · Reviewer_cNYr · 2023-07-06

**Soundness:** 2 fair
**Presentation:** 3 good
**Contribution:** 2 fair
**Rating:** 6
**Confidence:** 3

**Summary:**

This paper focuses on computing spectral clustering in the setting of horizontal federated learning. The main contributions of this paper are: (1) proposing a secure method for computing the adjacency matrix to achieve spectral clustering, (2) providing rigorous theoretical analysis to ensure the algorithm satisfies differential privacy, and (3) conducting experiments on multiple datasets to demonstrate the effectiveness of the proposed method, FedSC.

**Strengths:**

Strengths:
(1) The paper is logically organized and well-written, effectively introducing the work on FedSC.
(2) The proposed method, FedSC, is simple and reasonable, with theoretical guarantees. The paper is comprehensive (although detailed proof examination was not conducted).

**Weaknesses:**

I believe the experimental section of the paper can be further improved:

(1) In the Stage II Spectral Clustering phase of FedSC, when clients upload to the server, is there a significant risk of privacy leakage? Is the solution to add noise (as mentioned in Section 3.2)?

(2) Are both Stage I and Stage II of FedSC updated in each epoch? Can the author evaluate the difference in runtime between the two stages of FedSC and baseline models?


(3) Experimental section:

a. Although the paper has theoretically demonstrated the privacy of FedSC, there is still a lack of direct evidence in the experimental section to prove the privacy preservation of FedSC. It is suggested that the author report the clustering accuracy recovered by attackers from the data uploaded by clients to the server, as this would intuitively reflect the emphasized privacy of FedSC.

b. I would recommend including a natural baseline is to perform FedSC using only one client (e.g., just perform some kind of dimension reduction if Z is not too high-diemensional), which can give us better sense of the loss due to splitting if any and reduce the influence of this potentially denoise step (which is not novel) in the comparison.

c. Why does Table 1 show that FedSC outperforms SC (such as in the Bank dataset)?Can the author provide further explanation? Why does FedSC in the Bank dataset show a large standard deviation and unstable performance in Table 1, while baselines do not have this issue?

d. Is FedSC in Table 1 considering only perturbed data? Why not consider FedSC with perturbed factors? The Iris dataset and other datasets used in experiments are too simplistic. Can higher-dimensional image datasets be considered to demonstrate the effectiveness of the method?

**Questions:**

See weakness.

**Limitations:**

Not applicable.

---

> ### Author Rebuttal · Authors · 2023-08-09
>
> The authors sincerely appreciate the insightful comments made by the reviewer.
>
> **Response to weakness 1:**
>
> As claimed by Chai et al.[1], the general framework of federated matrix factorization may still leak users' raw data. Therefore, extra measure needs to be taken for enhancing the security of algorithms. Our study chose to add noise to give differential privacy guarantees on the proposed framework. According to Proposition 3.7 and Theorem 3.8 in our paper, privacy can be protected when sufficient noise is added.
>
> [1] Chai, D., Wang, L., Chen, K., $\And$ Yang, Q. (2020). Secure federated matrix factorization. IEEE Intelligent Systems, 36(5), 11-20.
>
> **Response to weakness 2:**
>
> Not like that. Stage I is a learning process for $\mathbf{Z}$ and $\\{\mathbf{C}_p\\} _{p=1}^P$ and has a number of epochs of training. After Stage I is completed, namely $\mathbf{Z}$ and $\\{\mathbf{C}_p\\} _{p=1}^P$ are well-learned, we perform Stage II, which just computes the adjacency matrix using $\mathbf{Z}$ and $\\{\mathbf{C}_p\\} _{p=1}^P$ and then conduct spectral clustering. The time complexity of Stage I is $\mathcal{O}((dmn+d^2m+d^3)T)$ and the time complexity of Stage II is $\mathcal{O}(d^2m+d^2n+Kn^2)$, where $T$ denotes the number of iterations in Stage I and $K$ denotes the number of clusters. We see that when $n$ is relatively small, the time cost of Stage I is higher. When $n$ is very large, the time cost of Stage II is higher (caused by eigenvalue decomposition). This is consistent with our numerical result.
>
> **Response to weakness 3(a):**
>
> We are sorry that we cannot fully understand your question. If the following response does not accurately match your question, please do not hesitate to let us know.
>
> First, due to the kernel trick, the optimization problem is nonlinear and nonconvex and hence it is very difficult for the attackers to recover the data $\mathbf{X}$ from the uploaded factors $\mathbf{Z},\mathbf{C}$, especially when $\mathbf{X}$ or $\mathbf{Z},\mathbf{C}$ are perturbed by noise. Thus, the clustering accuracy on the recovered $\mathbf{X}$ by attackers will be low. In addition, we suppose that the attackers use the upload $\\{\mathbf{C}_p\\} _{p=1}^P$ to cluster the data via k-means. We find that the clustering accuracy is lower than those of Kmeans, SC, and FedSC reported in Table 1 in our paper. For instance, the clustering accuracy on Iris is reported in the following table.
>
> |      |       Kmeans      |         SC        |        DSC        |       FedSC       |  FedSC(Attacker)  |
> |:----:|:-----------------:|:-----------------:|:-----------------:|:-----------------:|:-----------------:|
> | Iris | $0.8920\pm0.0028$ | $0.9000\pm0.0000$ | $0.5493\pm0.1263$ | $0.9027\pm0.0064$ | $0.6360\pm0.1557$ |
>
> **Response to weakness 3(b):**
>
> We designed an extra experiment to compare the clustering performance between using a single client ($P=1$) and using multiple ones ($P=8$). The result can be shown here. It is clear that the operation of splitting data across multiple clients may lead to accuracy loss of clustering.
>
> \begin{matrix}
> \hline
> &  Iris                &  Coil20       &Bank        &ORL     \\\\ \hline
> P = 1 & 0.9007\pm 0.0086 &0.8073\pm 0.0089  & 0.7536\pm 0.1271& 0.7937\pm0.0084 \\\\
> P = 8 & 0.8993\pm 0.0165 & 0.8003\pm0.0049 & 0.6821\pm0.1405 & 0.7112\pm0.0258 \\\\ \hline
> \end{matrix}
>
> **Response to weakness 3(c):**
>
> As we know, the similarity graph is a key ingredient of spectral clustering. As for vanilla spectral clustering, the similarity matrix $\mathcal{K}(\mathbf{X},\mathbf{X})$ directly constructed from raw data could be very sensitive to noise and outliers. Instead, FedSC reconstructs $\mathcal{K}(\mathbf{X},\mathbf{X})$ by $\mathbf{C}^T\mathcal{K}(\mathbf{Z}, \mathbf{Z})\mathbf{C}$, which has a potential denoising effect. Therefore, it is possible for FedSC to achieve a better performance in some cases. The unstable phenomenon on the Bank dataset is caused by the random initialization of factor $\mathbf{Z}$ and the number ($m$) of attributes of the data is only $5$ and the adaptively determined $d$ is also small, where the optimization is nonconvex. When we increased $d$, we found that the variance was reduced.
>
> **Response to weakness 3(d):**
>
> We are very sorry to have you confused about this. In fact, we reported the performance of FedSC with perturbed factors in Table 5 of the supplementary material. On the other hand, we have already included datasets of high-dimensional images like USPS and COIL-20 with sizes $16\times 16$ and $20 \times 20$, respectively. Nevertheless, to further improve the experiment, we added the results of MNIST ($28\times28$) and CIFAR-10($32\times32$) here.
>
> |                               |         |       Kmeans      |         SC        |        DSC        |       FedSC       |
> |:-----------------------------:|:-------:|:-----------------:|:-----------------:|:-----------------:|:-----------------:|
> |          $\mathbf{X}$         |  MNIST  | $0.5448\pm0.0257$ | $0.6265\pm0.0439$ | $0.1292\pm0.0144$ | $0.6139\pm0.0464$ |
> |                               | CIFAR10 | $0.2171\pm0.0132$ | $0.2182\pm0.0133$ | $0.1235\pm0.0062$ | $0.2134\pm0.0131$ |
> | $\mathbf{X}$ with $0.3\sigma$ |  MNIST  | $0.5402\pm0.0225$ | $0.5755\pm0.0366$ | $0.1337\pm0.0191$ | $0.5606\pm0.0457$ |
> |                               | CIFAR10 | $0.2209\pm0.0154$ | $0.2198\pm0.0172$ | $0.1194\pm0.0051$ | $0.2187\pm0.0125$ |
> | $\mathbf{X}$ with $0.7\sigma$ |  MNIST  | $0.5374\pm0.0555$ | $0.5711\pm0.0329$ | $0.1340\pm0.0135$ | $0.5029\pm0.0264$ |
> |                               | CIFAR10 | $0.2202\pm0.0147$ | $0.2205\pm0.0084$ | $0.1209\pm0.0045$ | $0.2134\pm0.0152$ |

---

> ### Comment · Reviewer_cNYr · 2023-08-21
>
> Thank you for the clarification, and I have raised to score to 6.

---

> > ### Author Response · Authors · 2023-08-21
> > **Authors' feedback**
> >
> > Dear Reviewer,
> >
> > We sincerely thank you for raising the score. We will include the experimental results you suggested in the revised paper.
> >
> > Best,
> >
> > Authors

---

### Official Review · Reviewer_EUeG · 2023-07-14

**Soundness:** 2 fair
**Presentation:** 3 good
**Contribution:** 3 good
**Rating:** 4
**Confidence:** 3

**Summary:**

This paper introduces a secure kernelized factorization approach of spectral clustering. The proposed method works on distributed datasets while maintaining privacy. The proposed technique approximates the kernel matrix required as in the spectral clustering, in an effort to preserve the privacy. This paper analyses the convergence of the optimization algorithm and offers bounds on the reconstruction error of the kernel matrix. The efficacy of this proposed approach is supported by experiments on both synthetic and real image datasets.

**Strengths:**

This paper studies an interesting problem: federated spectral clustering with privacy considerations. Since clustering requires access to the data, it is non-trivial to do clustering while maintaining privacy. The proposed method seems novel to me.

**Weaknesses:**

The only major concern I have is the privacy claim in this paper, and it is a vital one. The proposed method is only meaningful when it is proven to be secure and private (while keeping good utility), otherwise one could simply use any clustering algorithms out there. However, the evidence regarding its privacy guarantee is not sufficient in my opinion. Therefore, I would suggest the authors put more efforts into grounding their claim on the privacy.

Concretely, there are two (or three if more specifically) cases where security may be considered separately.
1. No noise is injected, i.e., the method detailed in section 2 where no differential privacy is considered. In this case, it is implied that this federated algorithm can help privacy, e.g., see line 89 (To guarantee the privacy of information, problem (7) shall be solved
90 in the framework of federated learning.) However, no evidence shows this federated operation can guarantee privacy. For example, one might apply the following idea to extract the local data $X_p$. Since equation (9) and (10) are quadratic minimization problem, they have explicit solutions. Therefore, by (10) it seems that one can write $Z^s_p$ as an expression of $X_p$ and $C^s_p$. Noting that by (9) $C^s_p$ can be written as an expression of $X_p$ and $Z^{s-1}$. Therefore, we have an equation of $Z^s, Z^{s-1}$ and $X_p$ for each $s$. Since $Z^s$ and $Z^{s-1}$ are known, can we solve $X_p$ from them? One argument for this idea not to work is that the dimension of the feature space may be high. However, there are also multiple such equations, i.e., $s=1,\dots,S$. In general, the choice of the feature map $\phi$, the number of communication rounds may also affect the level of privacy. Therefore, I am expecting to see such discussions regarding the "privacy guarantee" brought by the federated procedure.

2. In addition to learning in a federated way, the author also considers adding noise to enhance privacy, as analyzed in a differential privacy way. I acknowledge that Proposition 3.7 and Theorem 3.8 shows the DP guarantee for the proposed approach. However, DP can be achieved by adding sufficiently large noise if not considering the utility. Therefore, it is important to see the privacy-utility trade-off. For example, theoretically, eq. 32 gives an upper bound on the variance of the noise of the clustering $\sigma$, and proposition 3.7 gives a lower bound on the $\sigma$ for the sake of differential privacy. Can one derive a trade-off relation from the two inequalities? Empirically, Table 1 shows the performance of the proposed method given certain level of noise being injected. Can the level of noise being associated with the level of DP guarantee?


**Questions:**

Please refer to the weakness section for questions.

**Limitations:**

The authors addressed some of the limitations, e.g., the methods are not tested on very large datasets. However, there may be some other potential limitations as discussed in the weakness section.

---

> ### Author Rebuttal · Authors · 2023-08-09
>
> The authors thank the reviewer for the comments. Our responses to your comments and questions are as follows.
>
> **Response to Weakness 1:**
>
> First, due to the kernel mapping, the minimization problem for $\mathbf{Z}_p$ is nonconvex and has no closed-form solution. This implies that it is very difficult or even impossible to achieve its global minimum by common numerical methods when solving $\mathbf{X}_p$ from $\\{\mathbf{Z}^{s}\\} _{s=1}^S$. Thus, peeking at the privacy information from $\\{\mathbf{Z}^{s}\\} _{s=1}^S$ is computationally difficult. On the other hand, to our own knowledge and experience, the existing literature on federated matrix factorization did not claim that federated settings can efficiently protect users' privacy. For example, Chai et al. [1] argued that gradient information may still leak users' raw data and had to adopt homomorphic encryption to enhance the distributed federated matrix framework. Similarly, this is exactly the reason why we choose to add noise to the data or factors for improving the security of the proposed framework.
>
> [1] Chai, D., Wang, L., Chen, K., $\And$ Yang, Q. (2020). Secure federated matrix factorization. IEEE Intelligent Systems, 36(5), 11-20.
>
> **Response to Weakness 2:**
>
> It is easy to derive a range of $\sigma$ between the lower bound given by Proposition 3.7 and the upper bound given by Theorem 3.6. In particular, if we substitute $\sigma$ with the upper bound, we can get a strong level of privacy but the worst utility. Analogously, we can also get such a case for Theorem 3.8.
>
> For instance, Theorem 3.6 implies that, to ensure correct clustering, the noise (with standard deviation $\sigma$) should satisfy the following inequality
> $$\quad \sigma \leq \frac{1}{\xi}\left[\sqrt{r^2\left(B_1-B_2\right)+2 \theta^2}-\sqrt{2} \theta\right]$$
> On the other hand, Proposition 3.7 indicates that, to ensure $(\epsilon, \delta)$ differential privacy, $\sigma$ should obey
> $$\quad \sigma> 2 \sqrt{2 \ln (1.25 / \delta)} \tau_ X / \varepsilon$$
>
> Now combining these two inequalities together, we obtain
> $$2 \sqrt{2 \ln (1.25 / \delta)} \tau_ X / \varepsilon<\sigma\leq \frac{1}{\xi}\left[\sqrt{r^2\left(B_1-B_2\right)+2 \theta^2}-\sqrt{2} \theta\right]$$
> This provides a privacy-utility trade-off.
> When the noise level $\sigma$ is in the above range, we can get correct clustering while protecting privacy ($(\epsilon, \delta)$ differential privacy). By the way, $B_1-B_2$ is related to the property of the data. A larger $B_1-B_2$ means a better property for clustering, which further provide a larger upper bound for the noise level $\sigma$, yielding a stronger privacy guarantee. We will add more discussion in our revised paper. Thank you again for the insightful comments.
>
> The level of noise in Table 1 is directly related to the DP guarantee. For convenience, we show part of Table 1 below.
> The first block shows the noise-free case. The second and third blocks are the cases with noise inject, where $0.3\sigma_x$ and $0.5\sigma_x$ are two specific examples of the $\sigma$ in the DP guarantee.
>
> \begin{matrix}
> \hline & & \text { Kmeans } & \text { SC } & \text { DSC } & \text { FedSC } \\\\
> \hline  & \text { Iris } & 0.8933 \pm 0.0000 & 0.9000 \pm 0.0000 & 0.5480 \pm 0.0679 & 0.9000 \pm 0.0031 \\\\
> & \text { COIL20 } & 0.6113 \pm 0.0534 & 0.8025 \pm 0.0009 & 0.1009 \pm 0.0100 & 0.7828 \pm 0.0231 \\\\
> {\boldsymbol{X}} & \text { Bank } & 0.6122 \pm 0.0000 & 0.5918 \pm 0.0000 & 0.5582 \pm 0.0045 & 0.7672 \pm 0.1457 \\\\
> & \text { USPS } & 0.6704 \pm 0.0047 & 0.6635 \pm 0.0000 & 0.1686 \pm 0.0014 & 0.6596 \pm 0.0021 \\\\
> & \text { ORL } & 0.6325 \pm 0.0270 & 0.7865 \pm 0.0106 & 0.1653 \pm 0.0073 & 0.7235 \pm 0.0170 \\\\
> \hline & \text { Iris } & 0.8420 \pm 0.0274 & 0.8327 \pm 0.0267 & 0.4533 \pm 0.0674 & 0.8427 \pm 0.0404 \\\\
> & \text { COIL20 } & 0.6422 \pm 0.0366 & 0.7997 \pm 0.0029 & 0.0981 \pm 0.0084 & 0.7793 \pm 0.0240 \\\\
> {\tilde{\boldsymbol{X}} \text { with 0.3 } \sigma_x} & \text { Bank } & 0.6020 \pm 0.0038 & 0.5859 \pm 0.0105 & 0.5588 \pm 0.0074 & 0.6046 \pm 0.0064 \\\\
> & \text { USPS } & 0.6704 \pm 0.0063 & 0.6720 \pm 0.0044 & 0.1673 \pm 0.0019 & 0.6884 \pm 0.0509 \\\\
> & \text { ORL } & 0.6098 \pm 0.0167 & 0.7885 \pm 0.0047 & 0.1665 \pm 0.0057 & 0.7417 \pm 0.0280 \\\\
> \hline & \text { Iris } & 0.7740 \pm 0.0252 & 0.7313 \pm 0.0494 & 0.3833 \pm 0.0204 & 0.7540 \pm 0.0336 \\\\
> & \text { COIL20 } & 0.6389 \pm 0.0296 & 0.7950 \pm 0.0080 & 0.1033 \pm 0.0167 & 0.7403 \pm 0.0294 \\\\
> {\tilde{\boldsymbol{X}} \text { with 0.5 } \sigma_x}  & \text { Bank } & 0.6051 \pm 0.0076 & 0.5923 \pm 0.0094 & 0.5566 \pm 0.0030 & 0.6086 \pm 0.0073 \\\\
> & \text { USPS } & 0.6699 \pm 0.0031 & 0.7843 \pm 0.0030 & 0.1683 \pm 0.0017 & 0.7778 \pm 0.0062 \\\\
> & \text { ORL } & 0.5983 \pm 0.0295 & 0.7930 \pm 0.0172 & 0.1615 \pm 0.0096 & 0.7107 \pm 0.0345 \\\\ \hline
> \end{matrix}
>
> By the way, Figure 3 in our paper can also show the trade-off of privacy-utility, where the synthetic data are better posed. We will add more results of the synthetic data to quantify the trade-off of privacy-utility.

---

> > ### Author Response · Authors · 2023-08-21
> > **Any feedback to our rebuttal?**
> >
> > Dear Reviewer,
> >
> > As the period of reviewer-author discussion is ending, we'd like to know whether our rebuttal addressed your concerns or not. We have provided an explicit trade-off of privacy-utility based on the existing results in our paper and clarified the correspondence between the $\sigma$ in the privacy/clustering guarantees and the added noise in the experiments. Please feel free to let us know if you have further comments.
> >
> > Sincerely,
> >
> > Authors

---

### Author Rebuttal · Authors · 2023-08-10

We sincerely thank the area chairs and all reviewers. We have responded to the comments given by the reviewers carefully.
Here we summarize a few important points of our rebuttal or revision.

1. For Reviewer EUeG, we provide a formula to present a privacy-utility trade-off, namely, a trade-off between privacy guarantee and clustering success.

2. For Reviewer cNYr, we added three experiments.
   * Clustering result on the data recovered by an attacker
   * Accuracy loss of splitting data across multiple clients
   * Experiments on two more image datasets, MNIST and CIFAR10

3. For reviewer fSDW, we theoretically showed that the kernel reconstruction error in terms of $\ell_{\infty}$-norm is not related to the data dimension via analyzing the property of hyperparameter $r$. Eventually, the error is linear with $\sqrt{d}$, which is quite tight.

---

### Decision · Program_Chairs · 2023-09-21

**Decision:**

Accept (poster)

**Comment:**

This paper studies the problem of (kernelized) spectral clustering in the federated learning setting. While spectral clustering has been well studied in the past, applying existing results to the federated learning setting is not straightforward. This is particularly due to the fact that computing the kernel matrix requires calculating the inner products between (pairs of) data points (and hence sharing data). This paper proposes an approach for approximately calculating the kernel matrix and uses that for spectral clustering. The convergence of the proposed algorithm is studied. The authors also provided recovery results in the Gaussian kernel setting under certain assumptions. The reviewers had some initial concerns about the experiments section and setup. These concerns are addressed in the rebuttal period and hence I recommend acceptance of the manuscript.